# Conjunction of factors triggering waves of seasonal influenza

**Ishanu Chattopadhyay[1,2], Emre Kiciman[3], Joshua W Elliott[4], Jeffrey L Shaman[5], Andrey Rzhetsky[1,2,4,6]***

[1]Institute of Genomics and Systems Biology, University of Chicago, Chicago, United States; [2]Department of Medicine, University of Chicago, Chicago, United States; [3]Information and Data Science Group, Microsoft Research, Redmond, United States; [4]Computation Institute, University of Chicago, Chicago, United States; [5]Department of Environmental Health Sciences, Mailman School of Public Health, Columbia University, New York, United States; [6]Departments of Human Genetics, University of Chicago, Chicago, United States

**Abstract** Using several longitudinal datasets describing putative factors affecting influenza incidence and clinical data on the disease and health status of over 150 million human subjects observed over a decade, we investigated the source and the mechanistic triggers of influenza epidemics. We conclude that the initiation of a pan-continental influenza wave emerges from the simultaneous realization of a complex set of conditions. The strongest predictor groups are as follows, ranked by importance: (1) the host population's socio- and ethno-demographic properties; (2) weather variables pertaining to specific humidity, temperature, and solar radiation; (3) the virus' antigenic drift over time; (4) the host population'€™s land-based travel habits, and; (5) recent spatio-temporal dynamics, as reflected in the influenza wave auto-correlation. The models we infer are demonstrably predictive (area under the Receiver Operating Characteristic curve 80%) when tested with out-of-sample data, opening the door to the potential formulation of new population-level intervention and mitigation policies.

DOI: https://doi.org/10.7554/eLife.30756.001

*For correspondence: arzhetsky@uchicago.edu

**Competing interests:** The authors declare that no competing interests exist.

## Introduction

Seasonal influenza is a serious threat to public health, claiming tens of thousands of lives every year. A large number of past studies have focused on identifying the likely factors responsible for initiating each seasonal disease wave. Typically, each such study focused on one or a few hypothetical factors. Our study aimed at an integrative, joint analysis of numerous suggested disease triggers, comparing their relative importance and possible cooperation in triggering pan-US waves of seasonal influenza infection. The goal of this study was to identify the most informative combinations of statistical predictors associated with the initiation of pan-US influenza infection waves.

### Recent computational studies of influenza:

Computational study of the dynamics and factors influencing infectious disease spread began with compartmental models, such as the Susceptible-Infected-Resistant (SIR) model, which traces its origins to the beginning of the last century (*Kermack and McKendrick, 1927*). Initially a purely theoretical tool, these SIR-style models were subsequently enhanced with population and geographic data, allowing their application to specific cities and the distances between them (*Keeling and Rohani, 2002*). For instance, one approach, termed 'gravity wave' modeling, used geographic, short- and

**eLife digest** Influenza – or 'the flu' – is a contagious disease which sweeps across the globe like clockwork, claiming tens of thousands of lives. This is known as 'seasonal flu'.

Many scientists have tried to identify the factors that spark these yearly outbreaks. Some past studies have found that seasonal flu occurs when air that is normally humid turns dry, suggesting weather patterns play an important part. Other research has shown that air travel contributes to the flu spreading across the world. However, these studies typically focus on just one or two factors on their own. It is still not clear how exactly these factors combine to drive outbreaks, and then sustain the wave of infection.

To address this, Chattopadhyay et al. analyze the medical histories of 150 million American people over a decade, combining this information with large datasets about the different factors that trigger flu outbreaks. This includes detailed data about air travel and weather patterns, as well as census data that describe features of the population. Patterns of movement are also examined, for example by processing billions of Twitter messages "tagged" with a location. Chattopadhyay et al. used all of these datasets to model outbreaks of the flu in the United States, and see which factors play the biggest role.

It turns out that yearly outbreaks of seasonal flu are a result of a combination of elements. Some factors interact to help trigger the start of the wave, like humid weather in a highly populated area with nearby airports. Other factors, such how people move, encourage the spread of the infection. Finally, certain features of the population, for example how closely knitted a community is, make specific areas of the country more susceptible to the arrival of the disease. Overall, some of the most important elements of the model relate to the characteristics of the populations, the weather, the type of virus, and the number of short-distance journeys (rather than air travel).

Understanding how and why outbreaks occur can help policy-makers design strategies that reduce the spread and impact of seasonal flu, which could potentially save thousands of lives. Ultimately, the model developed by Chattopadhyay et al. could be used to test whether these policies would work before they are implemented in the real world.

DOI: https://doi.org/10.7554/eLife.30756.002

long-range, work-related human movement and demographic data to formulate gravity potentials between US counties in order to infer the dynamics of infection spread (*Viboud et al., 2006*).

Some studies have focused on one specific factor affecting infection, such as air travel, to simulate the spread of influenza (*Colizza et al., 2006*); other studies used SIR models, generalized for a collection of interconnected geographic areas, spatial-network or patch models, to model a number of common infections, including influenza, measles, and foot-and-mouth disease (*Riley, 2007*).

More ambitious network model approaches have simulated the global transmission of infectious disease using high-resolution, worldwide population data and the locations of International Air Transport Association (IATA)-indexed airports (*Balcan et al., 2009*). Similar to (*Viboud et al., 2006*), the authors of the study computed the global infection-pre-disposing 'gravity field' over the network of international airports. This network-based approach was subsequently developed further (*Balcan and Vespignani, 2011*) through the modeling of 'phase transition'–that is the chain-reaction switch of geographic infection status–in complex networks, utilizing approaches introduced in theoretical physics.

Another layer of sophistication was achieved by incorporating rich historical records. For example, Eggo et al. (*Eggo et al., 2011*) modeled the Spanish influenza epidemic of 1918–1919, using mortality documents from both the UK and the US, explicitly accounting for the size and distances between cities. In the same spirit, Brockmann and Helbing (*Brockmann and Helbing, 2013*) represented infection as diffusion on a complex network, estimating arrival times for infection across the globe.

Following the formulation of the hypothesis that absolute humidity modulates influenza survival and transmission (*Shaman and Kohn, 2009*), researchers began incorporating climate variables into SIR-like models (*Chowell et al., 2012*). More recent dynamic models have incorporated a probabilistic description of influenza infection's spatial transitions in space and time, accounting for selected demographic confounders (*Gog et al., 2014*) and (*Charu et al., 2017*).

In this study, rather than following the SIR-style modeling tradition, we used statistical epidemiology- and econometric-like approaches, in addition to a causality-network method presented here for the first time. There are some prior studies that are close to ours in spirit (but not in details). For example, (*Barreca and Shimshack, 2012*) used historical US influenza mortality data (1973–2002) in conjunction with collinear humidity and temperature records to establish county-level statistical associations between variables in the datasets. They concluded that absolute humidity was 'an especially critical determinant of observed human influenza mortality, even after controlling for temperature.' Another study, focusing on historical influenza records in the Netherlands, (*te Beest et al., 2013*) used the number of weekly influenza-like patient visits (transformed into an estimated rate of infection) as a response variable in a regression analysis of climate data. They concluded that the bulk of explained variation (57%) was attributed to the depletion of susceptible hosts during the disease season and non-weather-related 'between-season effects,' with only 3% explained by absolute humidity, represented as a continuous predictor variable. Additionally, this study observed that school holidays did not have a statistically significant effect on influenza transmission.

As all causality detection methods come with dissimilar limitations and are imperfect in unique ways, we designed our study intentionally to attack the same target problem using three different statistical approaches: Approach 1: A non-parametric Granger analysis (*Granger, 1980*) focusing on infection flows' directionalities across the US and whether influenza propagates via long- vs. short-distance travel (we run analysis across all pairs of air- and land-travel county neighbors, respectively). Approach 2: A mixed-effect Poisson regression (*Hedeker and Gibbons, 2006*) explicitly accounting for the auto-correlation of infection waves in time and space, along with the full set of socioeconomic, climate, and geographic predictors. Approach 3: A county-matching, non-parametric analysis to identify the minimum predictive set of factors that distinguish those counties associated with the onset of the influenza season (*Morgan and Winship, 2015*).

Our study became possible through access to several, very large longitudinal datasets: (1) a nine-year collection of insurance records capturing the dynamics of influenza-like illnesses (ILIs) in the United States (Truven MarketScan database, see Materials and methods); (2) temporally collinear, high-resolution weather measurements over every US county; (3) detailed air travel (*The United States Bureau of Transportation Statistics, 2010*) and geographic proximity data (*The United States Census, 2016*) showing connectivity between US counties; (4) billions of geo-located Twitter messages reflecting long- and short-distance human movement patterns, and; (5) US census data accounting for US county and county-equivalent population distribution, demographic, and socioeconomic properties (*HRSA, 2016*). An explicit comparison of the ILI data in the insurance claims to the influenza records provided by the Center for Disease Control and Prevention (*CDC, 2016*) showed that the two sources agree well ($\rho = 0.91, p = 3.5 \times 10^{-201}$), with insurance claims providing higher data resolution, see *Figure 1—figure supplement 1*. Curiously, the relationship between the two sources of ILI observations is not linear: We attribute this to the lower resolution of the CDC data. These three types of analysis produce congruent–albeit not identical–results.

## Results

The logical flow of our analysis is as follows: (1) We first show that our definition of ILIs corresponds well with CDC data, and that our causality coefficients, defined in Approach 1, have similar meanings to coefficients in the regression analysis; (2) We then explain the outcomes of the analysis according to Approach 1; (3) We then analyze the importance of putative casual factors, *Figure 1*, as applied to an initiation of influenza season, *Figure 2*; (4) In Approach 2, we pay special attention to the relative importance of short- and long-distance travel in influenza propagation, Figure 4; (5) We further test the best regression model in terms of predictive accuracy, Figure 5, using disjointed data parts for training and testing, and; (6) We culminate our analysis with a county-matching analysis, Figure 6.

### Approach 1: Causality streamlines from non-parametric granger analysis

Our analysis of health insurance claims covers nine years of influenza cycles (2003 to 2011, inclusively), see *Figure 2*. We visualized weekly, county-level prevalence as a movie (see Supplement); *Figure 2A–H* show a few relevant weekly snapshots from different years. The plates in *Figure 2A–H*, and especially the movie, clearly show that seasonal influenza cycles initiate in the South/

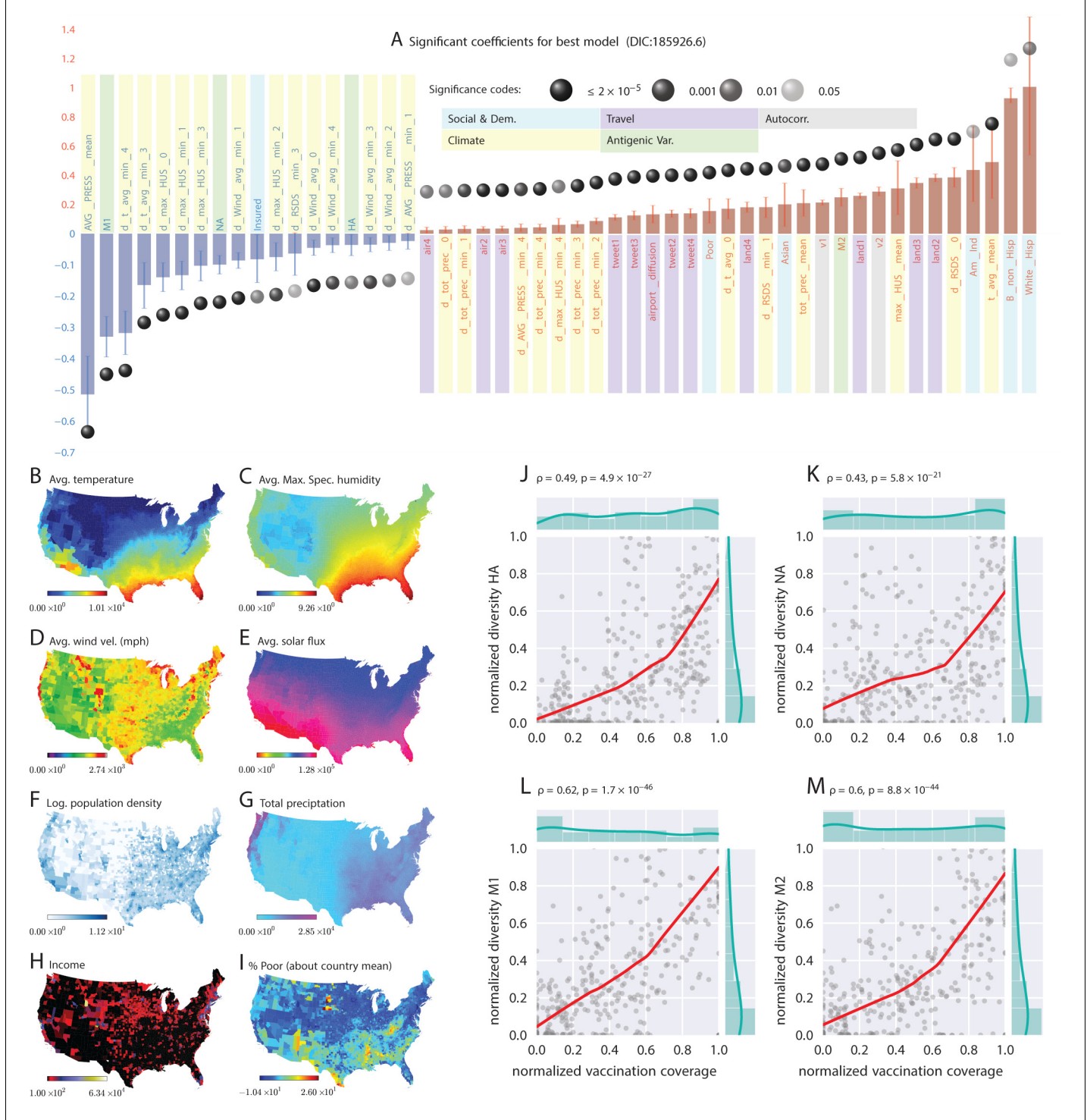

**Figure 1.** Putative determinants of seasonal influenza onset in the continental US and Poisson mixed-effect regression analysis (Approach 2). Plate A shows the significant variables along with their computed influence coefficients from the mixed-effect Poisson regression analysis (the best model chosen from 126 different regression equations with different variable combinations). The statistically significant estimates of fixed effects are grouped into several classes: climate variables, economic and demographic variables, auto-regression variables, variables related to travel, and those related to antigenic diversity (see the last entry in *Table 5* for the detailed regression equation used. The complete list of all models considered is given in Table S-D7). The fixed-effect regression coefficients plotted in Plate A are shown on a logarithmic scale, meaning that the absolute magnitude of predictor-specific effect is obtained by exponentiating the parameter value. A negative coefficient for a predictor variable suggests that the influenza rate falls as this factor increases, while a positive coefficient predicts a growing rate of infection as the parameter value grows. The integrated influence of

*Figure 1 continued on next page*

*Figure 1 continued*

individual predictors, under this model, is additive with respect to the county-specific rate of infection. For example, a coefficient of $-0.6$ for parameter AVG_PRESS_mean tells us that the average atmospheric pressure has a negative association with the influenza rate. As the mean atmospheric pressure for the county grows, the probability that the county would participate in an infection initiation wave falls. As $\exp(-0.6) = 0.54$, the rate of infection drops by 46% when atmospheric pressure increases by one unit of zero-centered and standard-deviation-normalized atmospheric pressure. Similarly, an increase in the share of a white Hispanic population predicts an increase in influenza rate: A coefficient of 1.3 translates into a $\exp(1.3) \times 100\% - 100\% = 267\%$ rate increase, possibly, because of the higher social network connectivity associated with this segment of population. Plates B - I enumerate the average spatial distribution of a few key significant factors considered in Poisson regression: (B) average temperature; (C) average maximum specific humidity; (D) average wind velocity in miles per hour; (E) average solar flux; (F) logarithm of population density (people per square mile); (G) total precipitation; (H) income, and; (I) percent of poor as deviations about the country average. Plates J-M show the strong dependence between our estimated antigenic diversity (normalized, see Definition in text) corresponding to the HA, NA, M1, and M2 viral proteins, and the cumulative fraction of the inoculated population (normalized between 0 and 1), where both sets of variables are geo-spatially and temporally stratified. Pearson's correlation tests shown in Plates J-M were performed under null hypothesis that there the two quantities (plotted along axes X and Y) are statistically independent ($H_0 : \rho = 0$).

DOI: https://doi.org/10.7554/eLife.30756.003

The following figure supplements are available for figure 1:

**Figure supplement 1.** Logical flow and cross-corroboration of conclusions.

DOI: https://doi.org/10.7554/eLife.30756.004

**Figure supplement 2.** Significant influencing variables obtained with mixed effect regression with different models as tabulated in *Table 1* of main text (three more models with DIC larger than that of the best model shown in *Figure 1* plate A).

DOI: https://doi.org/10.7554/eLife.30756.005

**Figure supplement 3.** Additional Cases: Significant influencing variables obtained with mixed effect regression with different models as tabulated in *Table 1* of main text (three more models with DIC larger than that of the best model shown in *Figure 1* plate A).

DOI: https://doi.org/10.7554/eLife.30756.006

**Figure supplement 4.** Violin plots for the coefficients inferred for variables that turn out to be significant in the best model, computed considering the complete set of models we investigated.

DOI: https://doi.org/10.7554/eLife.30756.007

**Figure supplement 5.** Spatial variation in the probability of patient visits corresponding to any ICD9-CM code (plate on left), and for diagnoses corresponding to influenza-like diseases (plate on right).

DOI: https://doi.org/10.7554/eLife.30756.008

**Figure supplement 6.** Informativeness of model vs model complexity as related to the number of terms in the regression equation.

DOI: https://doi.org/10.7554/eLife.30756.009

Southeastern US and sweep the country from south to north. This pattern is repeated, with some variation, each season.

*Figure 3G* shows the country-wide propagation dynamics as represented by our computed causality streamlines. The alignment of causality flow vectors into long, continuous streamlines suggests a stable propagation mechanism across the country; the probability of a long sequence of summary movement vectors accidentally matching in the direction by mere chance is vanishingly small ($p < 10^{-16}$ for longer streamlines).

Do epidemics originate from the same counties season after season? To answer this question, we follow 'causality streamlines' back to their source county. Informally, influenza onset in these source counties has little or no causal dependency on their neighbors. That is, their epidemic states are seemingly caused by factors outside of disease prevalence in other counties. *Figure 2K* presents the county-specific likelihood of streamline initiation across our nine years of data. To verify the near-shores position of these source counties is not a mere manifestation of a boundary effect of shore counties (no neighbors at the side of large water body), we carried out identical causality analyses with two different infections, specifically choosing diseases less likely to share etiologies with influenza: HIV and *Escherichia coli*. The results for both HIV and *Escherichia coli* infections are shown in *Figure 3J and K*, which exhibit flow patterns very different from those obtained for influenza. These streamlines almost never originate from the coasts, thus reducing the likelihood that the pattern observed for influenza is a geo-spatial boundary effect. Combined with the exceedingly low probability ($\sim 10^{-185}$) of chance inference for the streamlines, this strongly supports our conclusion that the epidemics are of coastal origin.

We directly validated our conclusion that influenza waves tend to start in the South by identifying counties which seem to trigger the epidemic. We computed a 'trigger period' of five to six weeks

**Table 1.** Social connectivity: The US Southern region appears to have an unusually high level of social connectivity.
(In GSS survey results, the number of close friends, close friends who are neighbors, and number of friends who all or mostly know each other is higher in the South, especially in the East/South/Central census region, than in the country at large.)

| | WSC (TX, OK, AR, LA) | ESC (MS, AL, TN, KY) | SA (FL, GA, SC, NC, VA, WV, MD, DC) | Country-at-large | WNC (ND, SD, NE, KS, MO, IA, MN) (not in South/Southeast) this is the second most social region following ESC (MS, AL, TN, KY) |
|---|---|---|---|---|---|
| Close friends | 7.22 | 12.76 | 8.20 | 7.57 | 10.56 |
| Close friends who are neighbors | 1.02 | 3.40 | 1.32 | 1.45 | 3.15 |
| % of friends who all or mostly know each other | All:20% Mostly: 43% | All:18% Mostly: 58% | All: 11% Mostly: 52% | All: 12 Mostly: 50% | All: 16% Mostly: 58% |
| How often visit closest friends* | 107 | 151 | 126 | 122 | 129 |

*Survey options are: lives in household, daily, several times a week, once a week, once a month, several times a year, and less often. These are converted to approximate number of visits per year (see Supplement for more information about the GSS analysis).

DOI: https://doi.org/10.7554/eLife.30756.011

for each season, defined as the period immediately preceding an exponential increase in influenza dispersion. To calculate this weekly dispersion, we treated each county as a node in an undirected graph, each with an edge connecting two geographically adjacent counties–only if they had both reported at least one influenza case in the specified week. We defined dispersion as the size of the largest, connected component in this undirected graph. Thus, a trigger period describes the period in which the size of the giant component of the infection graph rises above 250 counties from being under 100 as shown in *Figure 2I*, and then proceeds to the seasonal peak. *Figure 2J* presents the likelihood of a county being part of this largest, connected component during the trigger period. In the second approach, we followed causality streamlines back to their source county. *Figure 2K* presents the county-specific likelihood of streamline initiation across nine years.

These approaches produced qualitatively similar results (*Figure 2J and K*). While epidemics seem to start in many places around the country (see the origins of streamlines in *Figure 3J and I*), they successfully gain traction near large bodies of water (as evidenced by the most likely places of epidemic onset, see *Figure 2J and K*). Otherwise, they fizzle out before triggering an actual epidemic cycle (see *Figure 2J*). Seasonal initiation is neither spatially uniform nor simply a reflection of county-specific population density.

Our analysis of the Twitter movement matrix indicates that people most frequently travel between neighboring counties, preferentially towards higher-population-density areas, which shows that the maximum-probability movement patterns follow the local gradient of increasing population density (see *Figure 4—figure supplement 1*). In contrast, the geo-spatially-averaged movement vectors for each county reveal global flows in the movement patterns (see *Figure 3H*, along with Methods for the calculation of spatial averages).

*Figure 3H–I* suggest that average movement patterns largely agree with the influenza streamlines: Both patterns, especially in the South/Southeast of the country, are associated with flow pointing away from large bodies of water.

In addition to looking at the direction of short-range travel, we used our non-parametric Granger analysis to investigate the comparative strength of short- vs. long-range influenza propagation. In the first case, we considered the neighborhood map shown in *Figure 4A* (for a detailed definition of "neighbors,' see Materials and methods), and the in the second case, we considered associations between major, airport-bearing counties (see *Figure 4B*). We then plotted the distribution of the maximum pairwise coefficient of causality, where the maximization is carried out by fixing the source and the target and varying the delay in weeks, after which we attempt to predict the target stream.

Conclusions associated with Approach 1: The inferred causality streamlines computed from the infection time series in all counties (*Figure 3*) show that epidemics are mostly triggered near large water bodies and flow inland and away. They also illustrate that the US continental Southern states act as 'sinks' to a large proportion of these streamlines. ('Sinks,' in our definition here, are geographic areas that multiple streamlines converge *towards*; sinks are especially obvious when we look

**Table 2.** Fisher's exact test results on matched treatment combinations

| YR | $dh_0^\star$ | $dh_1^\star$ | $dt_0^\star$ | $h^\star$ | $t^\star$ | $u^\star$ | $M1^\star$ | $M2^\star$ | $V_0^\star$ | $V_1^\star$ | $a^\star$ | p-value | Odds ratio | Lower 99% cnf. bnds. | Upper 99% cnf. bnds. |
|---|---|---|---|---|---|---|---|---|---|---|---|---|---|---|---|
| 2003 | X | X | Y | Y | Y | Y | X | X | X | X | X | $1.9 \times 10^{-8}$ | 2.83 | 1.73 | 4.66 |
| 2004 | X | X | Y | Y | Y | Y | X | X | Y | X | X | $6.5 \times 10^{-3}$ | 6.22 | 1.08 | 132.03 |
| 2005 | X | X | Y | Y | Y | Y | X | Y | X | X | X | $3.4 \times 10^{-6}$ | 8.31 | 2.16 | 54.93 |
| 2006 | X | X | Y | Y | Y | Y | X | Y | X | X | X | $5.3 \times 10^{-7}$ | 4.56 | 1.96 | 12.0 |
| 2007 | Y | Y | X | Y | Y | Y | X | Y | X | X | X | $2.1 \times 10^{-2}$ | 3.85 | 0.82 | 28.16 |
| 2008 | Y | Y | Y | Y | Y | X | X | X | X | X | X | $1.9 \times 10^{-3}$ | 5.26 | 1.23 | 50.2 |
| 2009 | Y | Y | X | Y | Y | Y | X | X | X | X | X | $3.1 \times 10^{-10}$ | 4.78 | 2.38 | 10.34 |
| 2010 | X | X | Y | Y | Y | Y | X | X | X | Y | X | $1.4 \times 10^{-2}$ | 3.64 | 0.93 | 24.27 |
| 2011 | X | X | Y | Y | Y | Y | X | X | Y | X | X | $4.9 \times 10^{-11}$ | 4.91 | 2.51 | 10.05 |
| All Years | Y | Y | Y | Y | Y | Y | Y | Y | Y | Y | X | $7.2 \times 10^{-9}$ | 3.88 | 2.10 | 7.89 |
| All Years | Y | Y | Y | Y | Y | Y | Y | Y | Y | Y | Y | 1.0 | 1.0 | 0.48 | 2.15 |

DOI: https://doi.org/10.7554/eLife.30756.012

at the vector representation of causality direction. The opposite of a 'sink' is a 'source,' defined as an area at which at least one streamline starts.) This might explain the increased prevalence in the designated region. Additionally, the analysis shows that human travel is a very important driver of emergent epidemiological patterns, and that short-range, land-based travel is more important than air-travel. This result is cross-corroborated by our Poisson regression analysis (described next in Approach 2).

Approaches 2 and 3 are motivated by the 'why' questions: (1) Why do epidemics initiate where and when they do? and; (2) Why do some disease initiations become epidemics while others do not?

## Approach 2: Importance of factors from poisson regression

We focused on a subset of weeks associated with the initial rise of influenza waves (indicated by the gray bars in *Figure 2I*, and calculated as discussed earlier). The results from our best-fit model are illustrated in *Figure 1A*. We selected this particular model out of a total of 126 compared in the Bayesian analysis, a few of which we list in *Table 5*, ranked by their decreasing goodness-of-fit, measured with the Deviance Information Criterion, DIC (see Supplement). From the values of the inferred coefficients corresponding to the different factors, and taking into account their significance levels and credible intervals, we concluded that the roles played by weather variables, particularly humidity, appear to be substantially more complicated compared to what has been suggested in the literature.

### The surprisingly unimportant factors

School schedule was not predictive of influenza onset in our analysis: We ended up with a *p*-value of 0.84 and an odds ratio of 0.8403, strongly suggesting that school opening dates are not a significant factor in triggering the seasonal epidemic.

We are not claiming here that closing down schools during the seasonal peak, or during an initial phase of a seasonal epidemic, would not have a beneficial effect on maximum incidence. Rather, the observed epidemiological patterns over the time period we analyzed (2003–2011) do not seem to name 'school opening times' as a significant predictive factor–at least in the continental US.

We factored in the effect of vaccination coverage by estimating the cumulative fraction of the population that received the current influenza vaccine stratified by geo-spatial location and time of inoculation within each influenza season. Our analysis indicated that vaccination coverage is not a significant predictor of influenza onset/triggering period. It could be a reflection of overall vaccine ineffectiveness, or the choice of outcome predicted (i.e. vaccination might effect overall infection numbers over the entire outbreak, but not the timing of the trigger). It could also reflect the fact that different influenza type/subtypes have different virulence–so a vaccine against H3N2 during an H3N2 year, may be more effective, but due to that fact that H3N2 is more virulent, more people still wind up seeking medical care.

**Table 3.** Fisher's exact test results on matched treatment combinations

(a)

| YR p | -value 99% | Conf. Bnd. |
|---|---|---|
| max hus avg | | |
| 2003 | 0.003603 | 1.0, 4.21 |
| 2004 | 0.6919 | 0.16, 17.63 |
| 2005 0. | 1948 | 0.61, 7.89 |
| 2006 0. | 6525 0.28, 3.06 | |
| 2007 0. | 3574 0.49, 18.85 | |
| 2008 0. | 103 | 0.55, 1.23 |
| 2009 0 | .1067 | 0.68, 8.77 |
| 2010 | 0.5318 | 0.27, 41.03 |
| 2011 | 0.09054 | 0.74, 5.17 |
| ALL YRS | 1[1]10-4 | 1.12, 1.88 |
| t avg mean | | |
| 2003 | 0.06439 | 0.81, 3.65 |
| 2004 | 1 | 0.27, 10.62 |
| 2005 | 0.003339 | 1.17, 123.0 |
| 2006 | 0.8172 | 0.29, 4.0 |
| 2007 | 0.537 | 0.47, 7.42 |
| 2008 | 0.05985 | 0.59, Inf |
| 2009 | 0.0006337 | 1.37, 51.68 |
| 2010 | 0.2853 | 0.50, 9.28 |
| 2011 | 0.05729 | 0.85, 3.49 |
| ALL YRS 5.87 | [1]10-9 | 1.36, 2.23 |
| d hus 0 | | |
| 2003 | 0.5374 | 0.55, 3.41 |
| 2004 | 1 | 0.27, 11.01 |
| 2005 | 0.04401 | 0.81, 7.13 |
| 2006 | 0.001708 | 1.31, Inf |
| 2007 | 0.009199 | 1.0, 37.34 |
| 2008 | 0.3051 | 0.60, 5.92 |
| 2009 | 0.02726 | 0.82, 90.16 |
| 2010 | 1 | 0.41, 2.78 |
| 2011 | 0.577 | 0.57, 2.77 |
| ALL YRS | 1.48[1]10-5 | 1.12, 1.64 |
| d t avg mean 0 | | |
| 2003 | 0.004956 | 1.0, 24.12 |
| 2004 | 0.445 | 0.14, 6.0 |
| 2005 | 0.001164 | 1.23, 12.03 |
| 2006 | 0.01198 | 0.97, 11.08 |
| 2007 | 0.01147 | 0.96, 11.05 |
| 2008 | 0.08552 | 0.74, 11.18 |
| 2009 | 0.06847 | 0.73, 17.69 |
| 2010 | 0.08251 | 0.15, 1.63 |
| 2011 | 0.6031 | 0.47, 3.55 |
| ALL | YRS 4.98[1]10-11 | 1.35, 2.06 |

**(b)**

| YR p- | value | 99% Conf. Bnd. |
|---|---|---|
| hus 1 | | |
| 2003 | 0.1652 | 0.10, 2.43 |
| 2004 | 1 | 0.22, 24.42 |
| 2005 | 0.002004 | 0.09, 0.87 |
| 2006 | 1 | 0.32, 7.65 |
| 2007 | 0.389 | 0.33, 1.90 |
| 2008 | 0.02142 | 0.9, 8.48 |
| 2009 | 0.1822 | 0.67, 6.06 |
| 2010 | 0.6005 | 0.23, 4.22 |
| 2011 | 0.9166 | 0.6, 1.88 |
| ALL YRS | 0.07 | 0.72, 1.06 |
| d hus 2 | | |
| 2003 | 0.0083 | 1.01, 5.77 |
| 2004 | 0.79 | 0.36, 13.33 |
| 2005 | 0.275 | 0.71, 2.54 |
| 2006 | 0.24 | 0.66, 4.36 |
| 2007 | 0.19 | 0.62, 9.33 |
| 2008 | 0.18 | 0.65, 7.52 |
| 2009 | 0.53 | 0.44, 6.25 |
| 2010 | 0.08 | 0.21, 1.51 |
| 2011 | 0.59 | 0.69, 1.87 |
| ALL YRS | 0.13 | 0.78, 1.07 |
| urbanity | | |
| 2003 | 0.0083 | 1.01, 5.77 |
| 2004 | 0.79 | 0.36, 13.33 |
| 2005 | 0.275 | 0.71, 2.54 |
| 2006 | 0.24 | 0.66, 4.36 |
| 2007 | 0.19 | 0.62, 9.33 |
| 2008 | 0.18 | 0.65, 7.52 |
| 2009 | 0.53 | 0.44, 6.25 |
| 2010 | 0.08 | 0.21, 1.51 |
| 2011 | 0.59 | 0.69, 1.87 |
| ALL YRS | 2.2_10-16 | 3.67, 5.06 |
| airport proximity | | |
| 2003 | 0.004956 | 1.0, 24.12 |
| 2004 | 0.445 | 0.14, 6.0 |
| 2005 | 0.001164 | 1.23, 12.03 |
| 2006 | 0.01198 | 0.97, 11.08 |
| 2007 | 0.01147 | 0.96, 11.05 |
| 2008 | 0.08552 | 0.74, 11.18 |
| 2009 | 0.06847 | 0.73, 17.69 |
| 2010 | 0.08251 | 0.15, 1.63 |
| 2011 | 0.6031 | 0.47, 3.55 |
| ALL YRS | 1_10-16 | 1.73, 2.93 |

DOI: https://doi.org/10.7554/eLife.30756.013

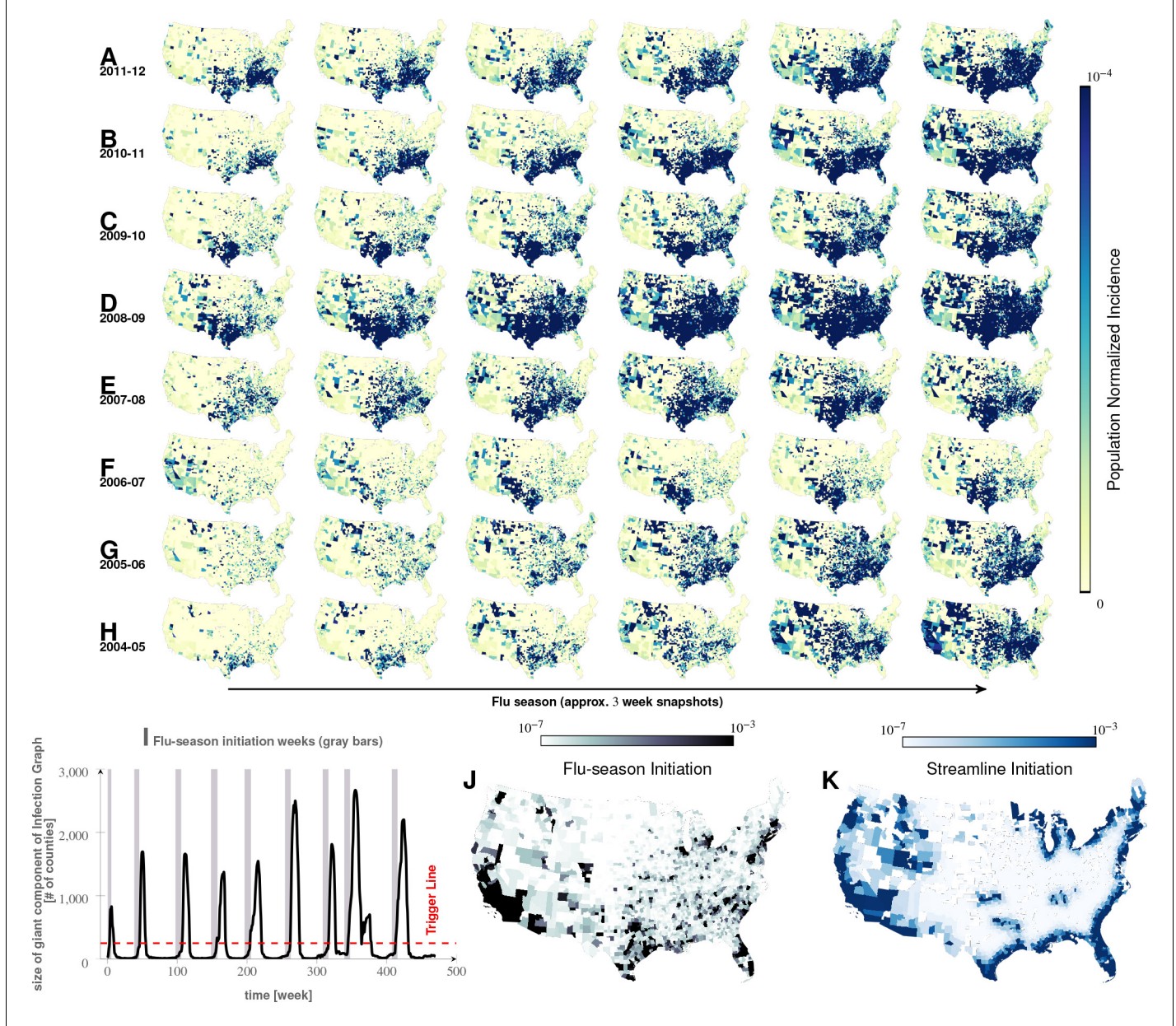

**Figure 2.** Characteristics of seasonal influenza in the continental US An analysis of county-specific, weekly reports on the number of influenza cases for a period of 471 weeks spanning January 2003 to December 2013 (Plates A-H) for recurrent patterns of disease propagation. In particular, the weeks leading up to that in which an epidemic season peaks (determined by significant infection reports from the maximum number of counties for that season) demonstrate an apparent flow of disease from south to north, which cannot be explained by population density alone (also see movie in Supplement). Plate I illustrates the near-perfect time table for a seasonal epidemic. Plates J and K compare the county-specific initiation probabilities of an influenza season, and the causality streamlines.

DOI: https://doi.org/10.7554/eLife.30756.010

The following video is available for figure 2:

**Figure 2—video 1.** Movement of seasonal influenza waves across USA.

DOI: https://doi.org/10.7554/eLife.30756.014

Vaccination coverage failed to reach predictive significance. The variables corresponding to spatio-temporal indicators of the cumulative fraction of the inoculated population are included in our best model (see the last entry in *Table 5*), but their effect fails to be significant. However, if we drop those variables from the model, the DIC increases. We suggest that the strong dependence

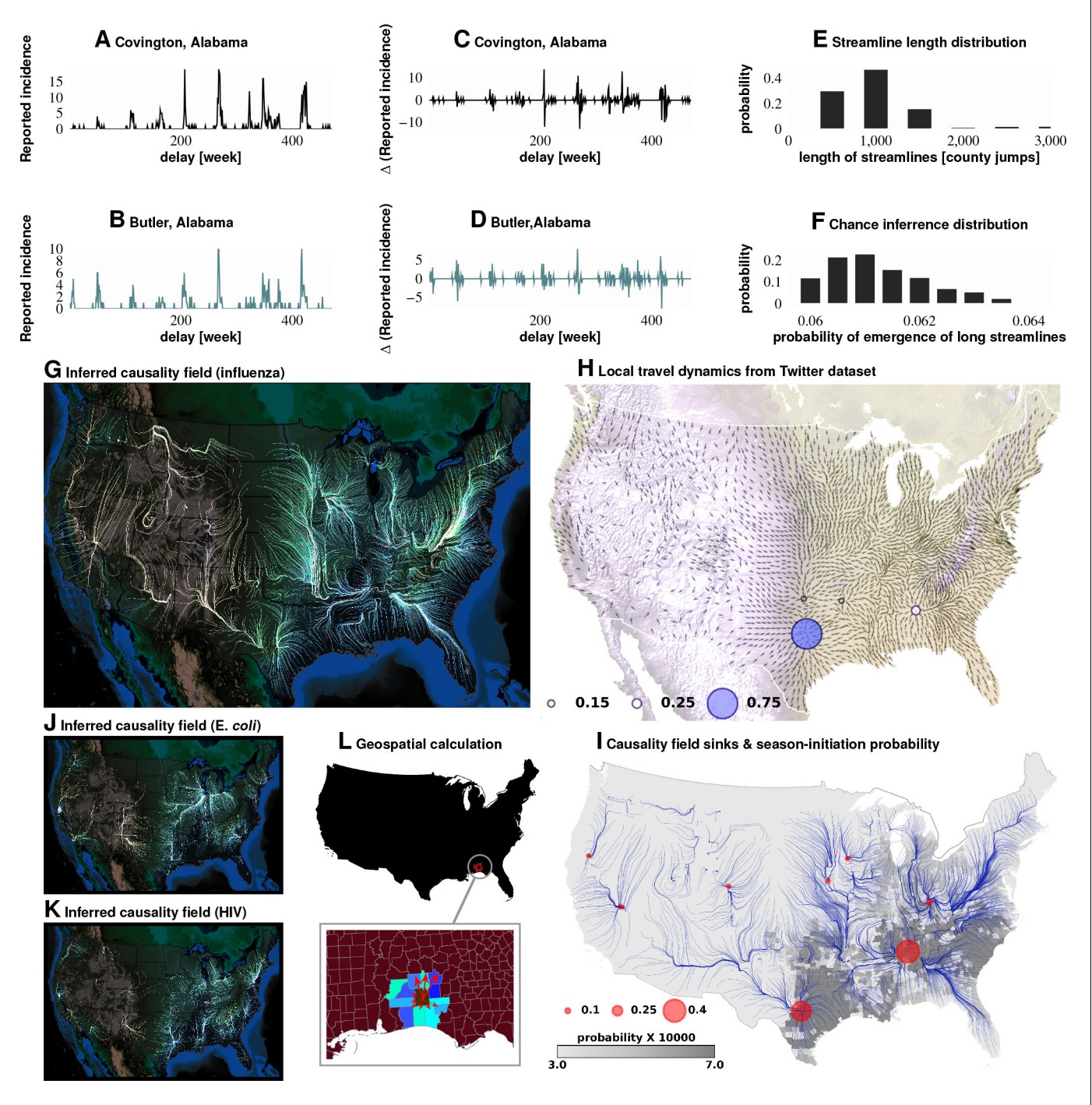

**Figure 3.** Computation of causality field, Approach 1 Plates A and B: Incidence data from neighboring counties in Alabama, US. Plates C and D: Transformation to difference-series, *i.e.*, change in the number of reported cases between weeks. We imposed a binary quantization, with positive changes mapping to '1,' and negative changes mapping to '0.' From a pair of such symbol streams, we computed the direction-specific coefficients of Granger causality (see Supplement). For each county, we obtained a coefficient for each of its neighbors, which captured the degree of influence flowing outward to its respective neighbors (Plate L). We computed the expected outgoing influence by considering these coefficients as representative of the vector lengths from the centroid of the originating county to centroids of its neighbors. Viewed across the continental US, we then observed the emergence of clearly discernible paths outlining the 'causality field' (Plate G). The long streamlines shown are highly significant, with the probability of chance occurrence due to accidental alignment of component stitched vectors less than $10^{-185}$; while each individual relationship has a chance occurrence probability of $\sim 6\%$ (Plates E and F). Plate H: Spatially-averaged travel patterns (see text in Materials and methods) and the sink distribution between expected travel patterns. These patterns (Plate H), along with the inferred causality field (Plate I), match up closely, with sinks

*Figure 3 continued on next page*

*Figure 3 continued*

showing up largely in the Southern US, explaining the central role played there. In Plate H, the size of the blue circles indicate the percentage of movement streamlines (computed by interpreting the locally averaged movement directions as a vector field) that sink to those locations. In Plate I, the size of the red circles indicate the percentage of causality streamlines that sink to the indicated locations. We note that ~75% of the movement streamlines sink in counties belonging to the Southern states, which matches up well with the sinks of the causality streamlines. In Plates J and K show spatial analysis results for two different infections (HIV and *E. coli*, respectively) and which exhibit very different causality fields, negating the possibility of boundary effects.

DOI: https://doi.org/10.7554/eLife.30756.015

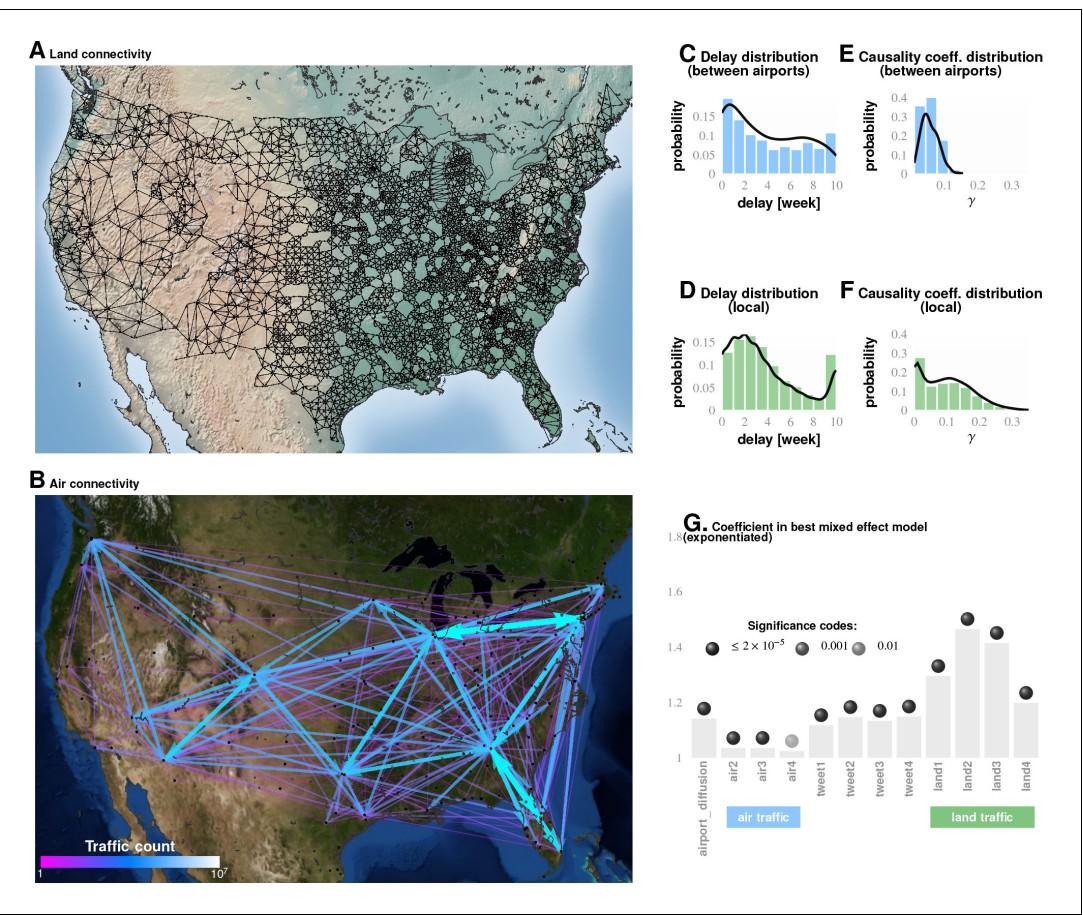

**Figure 4.** Comparing influence of short- and long-distance travel on infection propagation Plate A shows land connectivity visualized as a graph with edges between neighboring counties. Plate B shows air connectivity as links between airports, with edge thickness proportional to traffic volume. Plate C shows the delay in weeks for the propagation of Granger-causal influence between counties in which major airports are located, and Plate E shows the distribution of the inferred causality coefficient between those same counties. Plates D and F show the delay and the causality coefficient distribution respectively, which we computed by considering spatially neighboring counties. The results show that local connectivity is more important. We reached a similar conclusion using mixed-effect Poisson regression, as shown in Plate G: The inferred coefficients for land connectivity are significantly larger than those for air connectivity, tweet-based connectivity, or exponential diffusion from the top 30 largest airports. The coefficients shown in Plate G are exponentiated, allowing us to visualize probability magnitudes (see 'Model Definition').

DOI: https://doi.org/10.7554/eLife.30756.016

The following figure supplement is available for figure 4:

**Figure supplement 1.** Our analysis of the Twitter movement matrix indicates that people most frequently travel between neighboring counties, preferentially towards higher-population-density areas, which shows that the maximum-probability movement patterns follow the local gradient of increasing population density.

DOI: https://doi.org/10.7554/eLife.30756.017

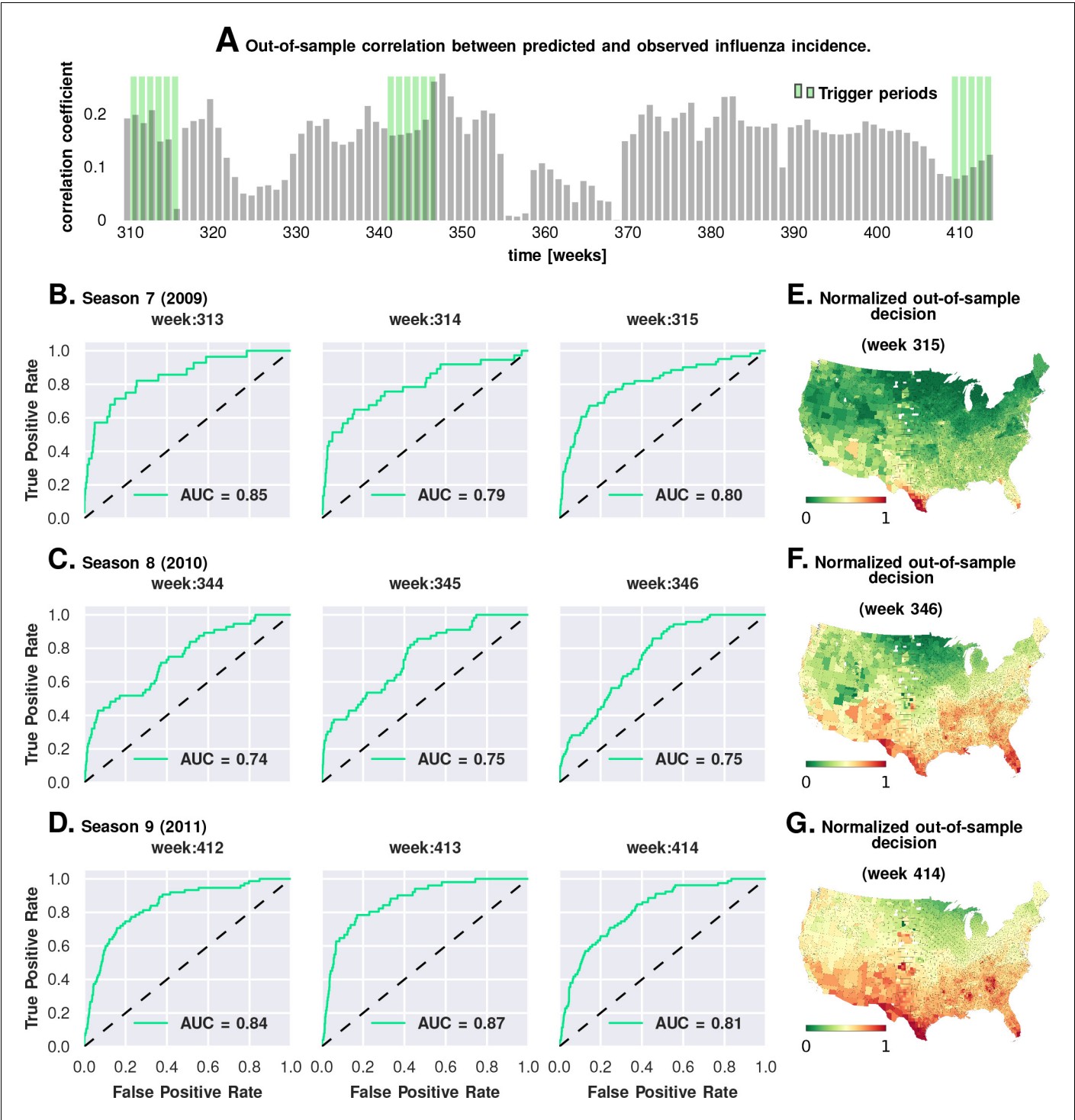

**Figure 5.** Prediction performance with training data from the first six seasons and validation on the last three. Plate A shows the correlation between the observed incidence and the model-predicted response. We show significant positive correlation, particularly within the trigger periods, between the model predictions and the actual held-out data. This gives us confidence to construct ROC curves for each week. Plates B-D show the ROC curves for the last three weeks of each of the three seasons in the out-of-sample period (potentially, these computations can be repeated for all possible partitions of study weeks into training and test samples). Plates E-G illustrate that the normalized decision variable, which is the normalized response from the model, identifies the South and Southeastern counties as the trigger zones.

DOI: https://doi.org/10.7554/eLife.30756.018

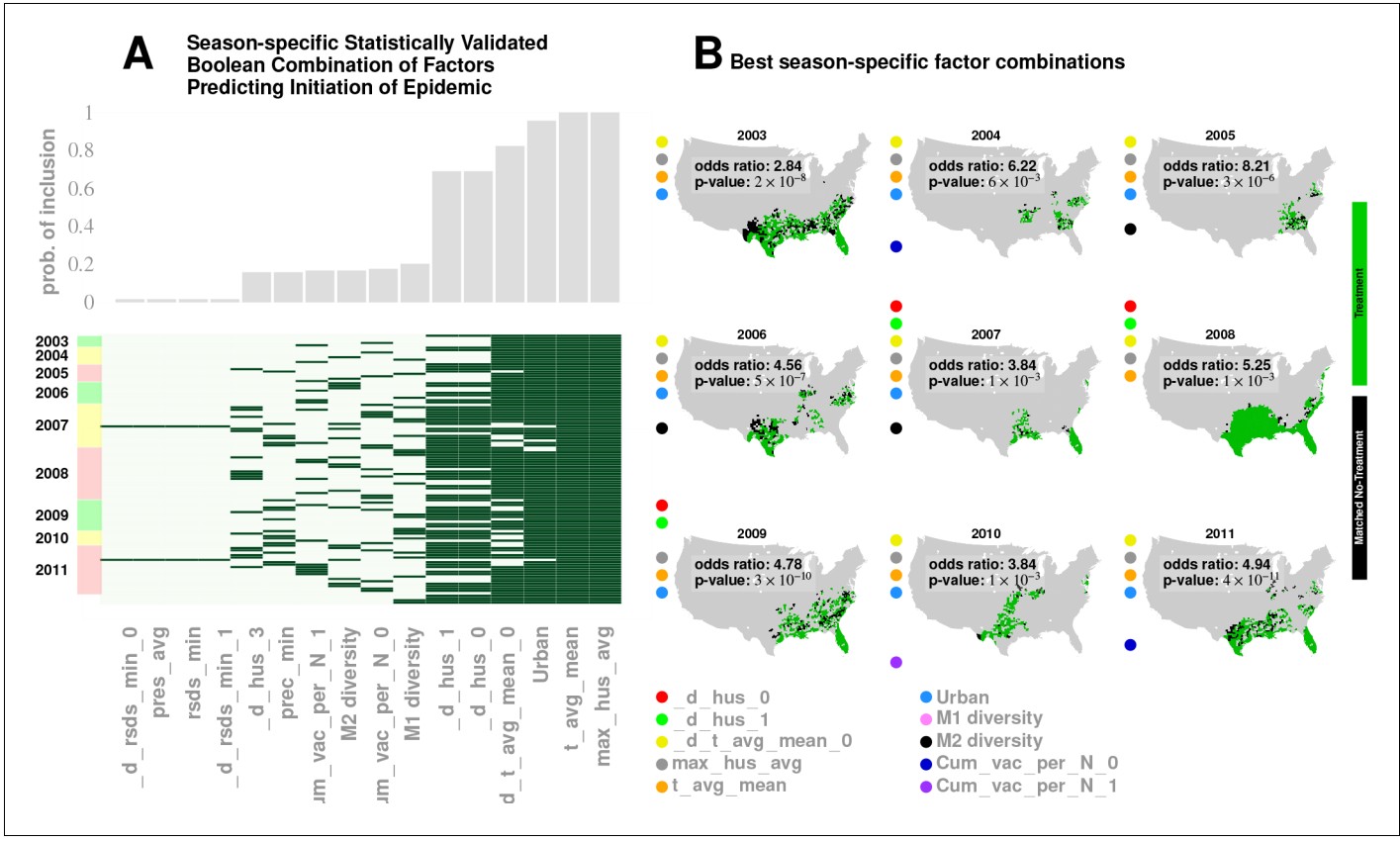

**Figure 6.** Results for our analysis involving county-matching (Approach 3). Plate A illustrates the factor combinations that turn out to be significant over the nine seasons. Notably, for each season, we have multiple, distinct factor sets that turn out to be significant ($p$<0.05) and yield a greater-than-unity odd ratio. Plotting the probability with which different factors are selected when we look at season-specific county matchings (the top panel in Plate A), we see a corroboration of the conclusions drawn in Approach 2. We find that specific humidity and average temperature, along with their variations, are almost always included. We do see some new factors that fail to be significant in the regression analysis, *e.g.*, degree of urbanity and vaccination coverage. While vaccination coverage is indeed included as a factor in our best performing model, in Approach two it failed to achieve significance, perhaps due to its strong dependence on antigenic variation (see *Figure 1J–M*). Degree of urbanity is indeed significant for some of the regression models we considered (see Supplementary Information), but was not significant for the model with the smallest DIC. Note that 'Treatment' here is defined as a logical combination of weather factors. A treatment is typically a conjunction of several weather variables. For example, the treatment shown in top left panel of Plate B involves a conjunction of: (1) a drop in average temperature during the week of infection; (2) a drop in temperature during the week of infection; (3) a higher-than-average specific humidity; (4) a higher-than-average temperature, and; (5) a high degree of urbanity. With respect to the 'treatment,' we can divide counties into three groups: (1) 'treated counties,' shown in green; (2) at least one matching county for each of the treated counties (matching counties are very close to the treated counties in all aspects but in treatment, which we called 'control' counties), shown in black, and; (3) other counties, shown in grey. The counties in the 'treatment' and 'control' groups are further subdivided into those counties that initiated an influenza wave and those that have not, resulting in four counts arranged into a two-by-two contingency table. We then used the Fisher exact test to test for association between treatment and influenza onset. Panels in Plate B show both the treated and control sets for the 9 seasons for a subset of chosen factors. The results are significant, as shown in *Tables 2* and *3*. The variable definitions are given in *Table 4*. Notably, some of the variables found significant in the regression analysis are not included above, and some which are not found to be significant in the best regression model show up here. This is not to imply that they are not predictive or lack causal influence. The matched treatment approach, as described above, is not very effective if we use more than ~$10 - 15$ factors simultaneously to define the treated set (for the amount of data we have); this results in a contingency table populated with zero entries.

DOI: https://doi.org/10.7554/eLife.30756.019

between antigenic diversity and vaccination coverage (see *Figure 1J–M*) is responsible for this effect: Vaccination coverage is important, but its influence is captured by the antigenic variation.

**Table 4.** Variables in mixed-effect Poisson regression analysis (Approach 2)

| Variable name | physical effect |
| --- | --- |
| N | Total number of patient visits given week and county (the offset) |
| max_HUS_mean | Mean county-specific maximum specific humidity over nine years |
| d_max_HUS_i | Normalized and zero-centered deviations of maximum humidity, i = 0–4 weeks before |
| t_avg_mean | Mean county-specific temperature over nine years |
| d_t_avg_i | Normalized and zero-centered deviations of mean temperature, i = 0–4 weeks before |
| RSDS_mean | Mean county-specific solar insolation over nine years |
| d_RSDS_i | Normalized and zero-centered deviations of mean solar insolation, i = 0–4 weeks before |
| AVG_PRESS mean | Mean county-specific solar insolation over nine years |
| d_AVG_PRESS_i | Normalized and zero-centered deviations of mean surface pressure, i = 0–4 weeks before |
| tot_prec_mean | Mean county-specific total precipitation over nine years |
| d_tot_prec_i | Normalized and zero-centered deviations of mean total precipitation, i = 0–4 weeks before |
| Wind_avg_mean | Mean county-specific average wind speed over nine years |
| d_Wind_avg_i | Normalized and zero-centered deviations of average wind speed, i = 0–4 weeks before |
| Income | County-specific mean income |
| airport_diffusion | Influence from proximity to airports, modeled as human traffic-weighted exponential diffusion from the 30 largest US airports |
| Am_Ind | % of American Indians in the county |
| Asian | % of Asians |
| White_Hisp | % of Caucasian/Hispanics |
| W_non_Hisp | % of Caucasian/Non-Hispanics |
| Black_Hisp | % of Black/Hispanics |
| B_non_Hisp | % of Black/Non-Hispanics |
| Pacific | % of Pacific Islanders |
| Insured | % of county population insured |
| Poor | % of county population under poverty line |
| Urban | % of county population classified as urban |
| land_i | influenza velocity change in the land neighbors of the county $i$ weeks before the current week |
| tweet_i | influenza velocity change in the Twitter neighbors of the county $i$ weeks before the current week |
| air_i | influenza velocity change in airport neighbors of the county $i$ weeks before the current week |
| v_i | change in rate of infection in the county itself $i$ weeks from the current |
| M1 | Diversity in M1 protein primary structure |
| M2 | Diversity in M2 protein primary structure |
| NA | Diversity in NA protein primary structure |
| HA | Diversity in HA protein primary structure |
| Cum_vac_per_N_i | vaccination coverage in the county cumulated over past 20 weeks $i$ weeks from the current |

(a) Definition of variables

DOI: https://doi.org/10.7554/eLife.30756.020

## The most important factors

The strongest predictor groups (ranked by importance) are the population's socio-demographic properties, weather, antigenic drift of the virus, land-based travel, and auto-correlation of influenza waves.

*Weather* As far as weather effects are concerned, epidemics tend to originate in places with high mean, maximum specific humidity, high average temperature, and low average air pressure, namely, in counties at the Southern, and, to a lesser extent, Eastern and Western US coastlines. Additionally, the spread of an epidemic is significantly influenced by a drop in specific humidity up to four weeks

**Table 5.** Different Models Considered and DIC Ranking

| Equation used in Poisson regression | DIC |
|---|---|
| flu ~LOGN + 1 + max_HUS_mean + d_max_HUS_0 + d_max_HUS_min_1 + d_max_HUS_min_2 + d_max_HUS_min_3 + t_avg_mean + d_t_avg_0 + d_t_avg_min_1 + d_t_avg_min_2 + d_t_avg_min_3 + max_HUS_mean * t_avg_mean + d_max_HUS_0 * d_t_avg_0 + d_max_HUS_min_1 * d_t_avg_min_1 + d_max_HUS_min_2 * d_t_avg_min_2 + d_max_HUS_min_3 * d_t_avg_min_3 + RSDS_mean + d_RSDS_0 + d_RSDS_min_1 + d_RSDS_min_2 + d_RSDS_min_3 + AVG_PRESS_mean + d_AVG_PRESS_0 + d_AVG_PRESS_min_1 + d_AVG_PRESS_min_2 + d_AVG_PRESS_min_3 + Income + airport_diffusion + Am_Ind + Asian + White_Hisp + W_non_Hisp + Black_Hisp + B_non_Hisp + Pacific + Insured+Poor + Urban+v1+v2+land1+land2+land3+land4+tweet1+tweet2+tweet3+tweet4+air1+air2+air3+air4+HA + M1+M2+NA. | 185942 |
| flu ~LOGN + 1 + max_HUS_mean + d_max_HUS_0 + d_max_HUS_min_1 + d_max_HUS_min_2 + d_max_HUS_min_3 + d_max_HUS_min_4 + t_avg_mean + d_t_avg_0 + d_t_avg_min_1 + d_t_avg_min_2 + d_t_avg_min_3 + d_t_avg_min_4 + RSDS_mean + d_RSDS_0 + d_RSDS_min_1 + d_RSDS_min_2 + d_RSDS_min_3 + d_RSDS_min_4 + AVG_PRESS_mean + d_AVG_PRESS_0 + d_AVG_PRESS_min_1 + d_AVG_PRESS_min_2 + d_AVG_PRESS_min_3 + d_AVG_PRESS_min_4 + tot_prec_mean + d_tot_prec_0 + d_tot_prec_min_1 + d_tot_prec_min_2 + d_tot_prec_min_3 + d_tot_prec_min_4 + Wind_avg_mean + d_Wind_avg_0 + d_Wind_avg_min_1 + d_Wind_avg_min_2 + d_Wind_avg_min_3 + d_Wind_avg_min_4 + Income + airport_diffusion + Am_Ind + Asian + White_Hisp + W_non_Hisp + Black_Hisp + B_non_Hisp + Pacific + Insured+Poor + Urban+v1+v2+land1+land2+land3+land4+tweet1+tweet2+tweet3+tweet4+air1+air2+air3+air4+HA + M1+M2+NA.+HA * NA. | 185940.6 |
| flu ~LOGN + 1 + max_HUS_mean + d_max_HUS_0 + d_max_HUS_min_1 + d_max_HUS_min_2 + d_max_HUS_min_3 + t_avg_mean + d_t_avg_0 + d_t_avg_min_1 + d_t_avg_min_2 + d_t_avg_min_3 + max_HUS_mean * t_avg_mean + d_max_HUS_0 * d_t_avg_0 + d_max_HUS_min_1 * d_t_avg_min_1 + d_max_HUS_min_2 * d_t_avg_min_2 + d_max_HUS_min_3 * d_t_avg_min_3 + RSDS_mean + d_RSDS_0 + d_RSDS_min_1 + d_RSDS_min_2 + d_RSDS_min_3 + AVG_PRESS_mean + d_AVG_PRESS_0 + d_AVG_PRESS_min_1 + d_AVG_PRESS_min_2 + d_AVG_PRESS_min_3 + Income + airport_diffusion + Am_Ind + Asian + White_Hisp + W_non_Hisp + Black_Hisp + B_non_Hisp + Pacific + Insured+Poor + Urban+v1+v2+land1+land2+land3+land4+tweet1+tweet2+tweet3+tweet4 air1+air2+air3+air4+HA + M1+M2+NA.+Cum_vac_per_N_0 | 185938.1 |
| flu ~LOGN + 1 + max_HUS_mean + d_max_HUS_0 + d_max_HUS_min_1 + d_max_HUS_min_2 + d_max_HUS_min_3 + d_max_HUS_min_4 + t_avg_mean + d_t_avg_0 + d_t_avg_min_1 + d_t_avg_min_2 + d_t_avg_min_3 + d_t_avg_min_4 + RSDS_mean + d_RSDS_0 + d_RSDS_min_1 + d_RSDS_min_2 + d_RSDS_min_3 + d_RSDS_min_4 + AVG_PRESS_mean + d_AVG_PRESS_0 + d_AVG_PRESS_min_1 + d_AVG_PRESS_min_2 + d_AVG_PRESS_min_3 + d_AVG_PRESS_min_4 + tot_prec_mean + d_tot_prec_0 + d_tot_prec_min_1 + d_tot_prec_min_2 + d_tot_prec_min_3 + d_tot_prec_min_4 + Wind_avg_mean + d_Wind_avg_0 + d_Wind_avg_min_1 + d_Wind_avg_min_2 + d_Wind_avg_min_3 + d_Wind_avg_min_4 + Income + airport_diffusion + Am_Ind + Asian + White_Hisp + W_non_Hisp + Black_Hisp + B_non_Hisp + Pacific + Insured+Poor + Urban+v1+v2+land1+land2+land3+land4+tweet1+tweet2+tweet3+tweet4+air1+air2+air3+air4+HA + M1+M2+NA.+Cum_vac_per_N_diff_1 + Cum_vac_per_N_diff_2 + Cum_vac_per_N_diff_3 + Cum_vac_per_N_diff_4 | 185935.9 |
| flu ~LOGN + 1 + max_HUS_mean + d_max_HUS_0 + d_max_HUS_min_1 + d_max_HUS_min_2 + d_max_HUS_min_3 + d_max_HUS_min_4 + t_avg_mean + d_t_avg_0 + d_t_avg_min_1 + d_t_avg_min_2 + d_t_avg_min_3 + d_t_avg_min_4 + RSDS_mean + d_RSDS_0 + d_RSDS_min_1 + d_RSDS_min_2 + d_RSDS_min_3 + d_RSDS_min_4 + AVG_PRESS_mean + d_AVG_PRESS_0 + d_AVG_PRESS_min_1 + d_AVG_PRESS_min_2 + d_AVG_PRESS_min_3 + d_AVG_PRESS_min_4 + tot_prec_mean + d_tot_prec_0 + d_tot_prec_min_1 + d_tot_prec_min_2 + d_tot_prec_min_3 + d_tot_prec_min_4 + Wind_avg_mean + d_Wind_avg_0 + d_Wind_avg_min_1 + d_Wind_avg_min_2 + d_Wind_avg_min_3 + d_Wind_avg_min_4 + Income + airport_diffusion + Am_Ind + Asian + White_Hisp + W_non_Hisp + Black_Hisp + B_non_Hisp + Pacific + Insured+Poor + Urban+v1+v2+land1+land2+land3+land4+tweet1+tweet2+tweet3+tweet4+air1+air2+air3+air4+HA + M1+M2+NA. | 185933.7 |

*Table 5 continued on next page*

Table 5 continued

| Equation used in Poisson regression | DIC |
|---|---|
| flu ~LOGN + 1 + max_HUS_mean + d_max_HUS_0 + d_max_HUS_min_1 + d_max_HUS_ min_2 + d_max_HUS_min_3 + d_max_HUS_min_4 + t_avg_mean + d_t_avg_0 + d_t_avg_min_1 + d_t_avg_min_2 + d_t_avg_min_3 + d_t_avg_min_4 + RSDS_mean + d_RSDS_0 + d_RSDS_min_1 + d_RSDS_min_2 + d_RSDS_min_3 + d_RSDS_min_4 + AVG _PRESS_mean + d_AVG_PRESS_0 + d_AVG_PRESS_min_1 + d_AVG_PRESS_min_2 + d_ AVG_PRESS_min_3 + d_AVG_PRESS_min_4 + tot_prec_mean + d_tot_prec_0 + d_tot_ prec_min_1 + d_tot_prec_min_2 + d_tot_prec_min_3 + d_tot_prec_min_4 + Wind_avg _mean + d_Wind_avg_0 + d_Wind_avg_min_1 + d_Wind_avg_min_2 + d_Wind_avg_min _3 + d_Wind_avg_min_4 + Income + airport_diffusion + Am_Ind + Asian + White_Hisp + W _non_Hisp + Black_Hisp + B_non_Hisp + Pacific + Insured+Poor + Urban+v1+v2+land1 +land2+land3+land4+tweet1+tweet2+tweet3+tweet4+air1+air2+air3+air4+HA + M1+M2+NA.+Cum_vac_per_N_0 | 185932.3 |
| flu ~LOGN + 1 + max_HUS_mean + d_max_HUS_0 + d_max_HUS_min_1 + d_max_HUS_min_ 2 + d_max_HUS_min_3 + d_max_HUS_min_4 + t_avg_mean + d_t_avg_0 + d_t_avg_min _1 + d_t_avg_min_2 + d_t_avg_min_3 + d_t_avg_min_4 + RSDS_mean + d_RSDS_0 + d_RSDS_min_1 + d_RSDS_min_2 + d_RSDS_min_3 + d_RSDS_min_4 + AVG_PRESS_mean + d_AVG_PRESS_0 + d_AVG_PRESS_min_1 + d_AVG_PRESS_min_2 + d_AVG_PRESS_ min_3 + d_AVG_PRESS_min_4 + tot_prec_mean + d_tot_prec_0 + d_tot_prec_min_1 + d_tot_prec_min_2 + d_tot_prec_min_3 + d_tot_prec_min_4 + Wind_avg_mean + d_Wind _avg_0 + d_Wind_avg_min_1 + d_Wind_avg_min_2 + d_Wind_avg_min_3 + d_Wind_avg_ min_4 + Income + airport_diffusion + Am_Ind + Asian + White_Hisp + W_non _Hisp + Black_Hisp + B_non_Hisp + Pacific + Insured+Poor + Urban+v1+v2+land1+land2+land3+land4 +tweet1+tweet2+tweet3+tweet4+air1+air2+ air3+air4+HA + M1+M2+NA.+Cum_vac_per_N_0 + Cum_vac_per_N_1 + Cum_vac_ per_N_2 + Cum_vac_per_N_3 | 185926.6 |

DOI: https://doi.org/10.7554/eLife.30756.021

before its onset. However, this effect is weaker than the mean maximum specific humidity effect. Drop in average temperature that dips one to three weeks prior to the epidemic onset is also significantly important (this is consistent with earlier experiments [*Lowen et al., 2008*]), especially when the temperature drop is accompanied by a decrease in specific humidity, average wind speed, and solar flux. However, high levels of solar flux in the week of onset are also important. This complicated set of weather conditions, a signature of the cold air front (*Shaman et al., 2010*), is validated by our out-of sample predictions to increase the risk of triggering the seasonal epidemic. Total precipitation also plays a positive role.

## The weather/humidity paradox

How can colder weather and lower humidity be a predictor of influenza, if influenza epidemic waves tend to start in the South with warmer climates and higher humidity? Our resolution of this seeming controversy is as follows: The stress is on the drop in both humidity and temperature in those areas with high average annual values of these measurements. A blast of colder, lower-humidity weather in these warm-climate areas has two effects: (1) The influenza virus can stay viable in water droplets longer than in hot, sunny weather, and; (2) Humans tend to interact indoors, in more crowded conditions. Both of these factors are favorable for transmission of the virus to the population at large.

*Antigenic variation* Antigenic diversity for HA, NA, M1, and M2 are important predictors. While HA, NA, and M1 inhibit the trigger, M2 diversity enhances it. This peculiar difference in the direction of influence might be a manifestation of the roles played by the individual viral proteins in its life-cycle.

The first three proteins are directly involved in the viral binding to host cell surface receptors, while M2 activity is needed only during HA biosynthesis. Additionally, proteolysis experiments indicated that M2 proton channel activity helped to protect (H1N1)pdm09 HA from premature conformational changes as it traversed low-pH compartments during transport to the cell surface (*Alvarado-Facundo et al., 2015*).

We found that antigenic diversity is a significant predictor in all four of the viral proteins we considered. Interestingly, while the increasing diversity found in HA, NA, and M1 inhibits the epidemic trigger, the higher diversity in M2 enhances it (see Discussion).

*Travel* Land travel intensity one to three weeks before epidemic onset is a strong predictor. Air travel is also predictive, but its strength is an order of magnitude weaker than that of land travel.

*Autocorrelation* The increase in an influenza outbreak's infection weekly rate one and two weeks before an epidemic onset (in the epidemic source county itself) is predictive of epidemic wave origin.

We have used substantially richer datasets than those used by earlier studies (*Shaman et al., 2010*; *Tamerius et al., 2013*), which lends strong statistical support to our conclusions. It also allowed us to disentangle and make precise the contributions from different factors, *e.g.*, mean county humidity vs. drops in humidity before an infection. While we found the former effect to be clearly stronger (in accordance to previously reported results [*Shaman et al., 2010*]), the other diverse set of factors were also found to be significant.

## Validation of predictive capability

The robustness of our results is established in a number of ways:

1. In mixed-effect regression (Approach 2), we compared over 120 chosen model variations (see *Table 5* for an abridged list, and *Supplementary file 3* for the complete enumeration of considered models); the results appear to be qualitatively stable, though the quantitative performance of the models vary somewhat as measured by DIC for different configurations of regression equations.
2. We carried out a direct validation of predictive performance by estimating model parameters using the first six seasons and predicting the epidemic trigger locations using the last three (see *Figure 5* and Materials and methods). The out-of-sample predictions of influenza incidence are always positively correlated with observed incidence (Plate A). Perhaps more importantly, we obtained good predictability as measured by the area under the curve (AUC $\approx 80\%$) for the receiver operating characteristics (ROC, See Plates B-D). Plates E-G show that our out-of-sample predictions correctly identified epidemic initiation in the Southern and Southeastern counties of the continental US.
3. As we discuss in the next section, in Approach 3, we conducted a corroborating matched effect analysis on the counties, using combinations of county-specific factors as a 'treatment,' not unlike clinical trials in which patients on a drug regimen are matched to patients receiving a placebo (*Morgan and Winship, 2015*).

## Approach 3: Matching counties and factor combinations

In Approach 3, we investigated combinations of factors presented as 'treatment' via a non-parametric, exact-matching analysis of US counties during the weeks of epidemic onset on a season-by-season basis.

First, we collected the list of all counties with a drop in maximum specific humidity during the weeks leading up to an influenza season in a particular year. This is the 'treated set': the set of counties that may be thought of as subjected to the positive 'treatment' of a drop in specific humidity. We split this set into two, considering counties that also experience increased influenza prevalence during the epidemic onset, and ones that do not (counties with two different values of the outcome variable). The number of counties in these two sets define the first row of a $2 \times 2$ contingency table. In the second row (the 'control set'), we focused on counties that do not experience a drop in the maximum specific humidity. However, we only considered counties that have a matching counterpart in the treated set in the following sense: For each county in the control set, we found at least one in the treated set such that the rest of the significant variables (other than specific humidity) had similar variation patterns in both counties. Once we defined the control set, we split it in the manner described for the treated set: We counted the number of control counties that experienced an increased influenza prevalence during epidemic onset, and those which did not. This defined the second row of the contingency table. Finally, we used Fisher's exact test to compute an odds ratio (the odds of realizing these numbers by chance), along with the test-derived significance of the association between the 'treatment' and epidemic wave initiation (*p*-value). Furthermore, we defined our treatment to consist of multiple factors simultaneously, *e.g.* specific humidity and its change in the preceding week, along with average temperature and degree of urbanity, see *Figure 6*.

Note that geographic clustering of 'treated' and 'untreated' counties arose automatically as a result of similar weather patterns being imposed via the constraint of multiple climate variables.

Unlike the mixed-effect regression approach (Approach 2), this matching analysis is non-parametric, and intended to reveal whether multiple factors are, indeed, simultaneously necessary.

The results of Approach 3's analysis are presented in *Figure 6* and *Tables 2* and *3*. We found that no single variable was able to consistently yield a statistically significant odds ratio greater than one; multiple factors interacted to shape an epidemic trigger (see *Table 3* for a few examples). With a total of 47 significant variables in our best mixed-effect model, an exhaustive search for all combinations was not feasible. Instead, we performed a standard evolutionary search, looking for combinations that yielded a significant odds ratio for individual seasons. Additionally, we considered all seasons together (by simply adding the contingency tables, element-wise) in order to increase the test's statistical power.

We isolated ten variables (as shown in *Figure 6*, Plate B) in this manner which included maximum specific humidity and average temperature along with their variations, the degree of urbanity, antigenic variation, and vaccination coverage.

The factors that appeared most often in our analysis are illustrated in Plate A: It appears that maximum specific humidity and average temperature, along with their variations, and the degree of urbanity have the most frequent contribution, followed by antigenic variation and vaccination coverage.

We did see some new factors here that failed to be of significance in the regression analysis (Approach 2), *e.g.*, degree of urbanity and vaccination coverage. While vaccination coverage was included as a factor in our best performing model in Approach 2, it failed to achieve significance, perhaps due to its strong dependence on antigenic variation (see *Figure 1J–M*). Degree of urbanity was indeed significant for some of the regression models we considered (see Supplementary Information), but failed to be so for the model with the least DIC.

Thus, Approach three corroborates and strengthens key claims of Approach 2.

The exact set of factors varied somewhat over the seasons; nevertheless, together, they yield significant results when all seasons are considered together. The matching analysis corroborates our results from both the mixed-effect regression and the geographic streamline analyses: The sets of counties initiating the wave are near coasts on the Southern region of the continental US (see Plates A - I in *Figure 6*).

## Local travel vs. long-distance travel and influenza

Our conclusion that local travel is predominantly responsible for disease wave propagation is supported by several lines of analysis.

First, continuous land-movement infection waves are visible in the weekly influenza rate movie; we computed this movie from insurance claim data and made it available with results of this study.

Second, because our all-weeks-included dataset was too large for the R MCMCglmm package to handle, we performed mixed-effect Poisson regression calculations using a 50 percent random sample of all the weeks for which data were available. In this computation, the airport-proximity, fixed-effect coefficient turned out to be statistically significantly negative (see Supplement A, as well as the editable output file 'flu-50-percent-weeks.txt').

Third, the results from our Granger-causality inference showed that:

1. Local county-to-county movements were much more predictive of influenza wave change than airport movements. In comparing Plates E and F in *Figure 4*, we see that the local movement causality coefficient ($\gamma$) is, on average, twice as large as that for long-range movement (*Figure 4E*). *Figure 4E* shows that the mean long-range causality coefficient is approximately 0.05, whereas it is just over 0.1 in the local propagation. As the causality coefficient quantifies the amount of predictability (measured as information in bits) communicated about the target data stream per observed bit in the source data stream, it follows that, on average, every ten bits of sequential incidence data from an influencing location tells us one bit about the unfolding incidence dynamics in the target location. Therefore, in case of the long-range movement, informativeness is twice as low, so we need on average 20 bits to infer one bit about the state of infection. These calculations strongly suggest that local movement is predictively stronger with regards to influenza infection propagation.
2. While the most frequent value of the computed time delay in influence propagation between counties with large airports is zero weeks, this distribution is significantly flatter compared to that for local, county-to-county influence propagation.

## Discussion

A summary of the complex relationship between the driving factors that contribute to the trigger and subsequent development of a seasonal epidemic can be clarified with a forest fire metaphor.

The maturation of a forest fire requires the collusion of multiple factors—namely the presence of flammable media, an initiating spark, and a wind current to help spread the fire. Our conceived mapping of this analogy to influenza infection is as follows:

1. Flammable Media: The Southern US appears to have an unusually high level of social connectivity; it is at least one order of magnitude higher than that in the north of the country (see GSS survey results (*Smith et al., 1972*) and *Table 1*). The number of close friends, close friends who are neighbors, and communities of people who all, or mostly, know each other is much higher in the South than in the country at large. Our conjecture is that a manifestation of this high-connectivity is the highest *apparent* percentage of people infected with influenza (20% as opposed to 4% in other parts of the country).
2. Initiation Spark: An initial spark for the infection wave is generated by a combination of weather and demographic factors. Specifically, warm, humid places are conducive to influenza wave initiation - particularly in weeks where specific humidity drops. Airport proximity is important, as well as demographic and economic makeup and also the degree of urbanization. Note that the first static condition (warm humid places) is highly correlated with areas in the South with greater social connectivity. It is possible that static meteorological variables (warm mean temperature and high mean humidity) serve as proxies for high social connectivity or other correlated socioeconomic factors.
3. Wind: The 'wind' in this analogy is the collective movement of a large number of people, integrated over time, revealing persistent 'currents.' These currents reproducibly point from coastlines and move inwards towards the center of the continent, making them perfect vehicles to transmit the infection inland from the shores.

Each of our three types of computational approaches has their strengths and weaknesses: (1) The Poisson mixed-effect regression allows for the direct comparison of the predictive strength of numerous predictor variables and accounts for spatial and temporal autocorrelation, but relies on strong modeling assumptions; (2) The non-parametric Granger analysis is not limited by restrictive modeling assumptions in our implementation, though it focuses only on trends of infection propagation between counties, and; (3) The county-matching analysis is also model-free, but this freedom comes at the expense of lesser statistical power.

## What is new in this study?

The following aspects make our study of influenza triggers new in the influenza literature: (1) Instead of simulating the plausibility of one particular epidemic trigger model with a dynamic disease transmission model, we used formal model selection tools to compare the goodness of fit of hundreds of plausible models; (2) We explicitly attempted to systematically cross-compare the importance of numerous individual factors typically hypothesized to contribute to epidemic onset; (3) To accomplish this, we collected an unprecedented volume of temporal and spatial data on disease dynamics and the dynamics of putative predisposing factors; (4) We used several orthogonal, computational causality-inference techniques (one of which was developed specifically for this study) to probe associations between disease onset and putative epidemic triggers; (5) We tested our best models for their predictive potential and demonstrated that they are, indeed, suitable for forecasting disease waves, and; (6) We combined, for the first time, numerous candidate factors in a single, integrative study.

Convergent conclusions, culled from these radically different techniques, strengthen our claims and make it statistically unlikely that we are observing analysis artifacts. First, the Granger causality analysis results (Approach 1) provide insights into the details of influenza's epidemiological dynamics. *Figure 3G* traces out the paths most likely followed by the infection, on average, across the continental US. We note that ~75% of the streamlines sink in counties belonging to the Southern states, which matches up well with the streamline-encoded dynamics of weekly disease incidence over nine years (see *Figure 3I*). What drives this particular causality field's geometry? While we cannot

definitively answer this question, a comparison of the global patterns emerges from the local mobility data culled from the aforementioned Twitter database and offers a tentative explanation (see *Figure 3H*). Second, contrary to reported human travel pattern influence on seasonal epidemics (*Viboud et al., 2006*) (but consistent with [*Gog et al., 2014*]), we find that short-distance travel contributes more significantly to disease spread (see *Figure 4*). In particular, we find that long-range air travel is important as an epidemic *trigger*, but once infection waves are triggered, air travel patterns (or proximity to major airports) become less important. Short-range mobility, on the other hand, is apparently important for sustaining infection transmission over each season. Thus, we find short-range travel to be more important for defining the emergent spatio-temporal geometry of infection waves, while proximity to airports is more important for actually triggering an influenza season; the latter loses positive influence once an infection is under way. This conclusion is justified as follows: (1) When we performed regression calculations using all weeks for which data were available (as opposed to wave initiation weeks only) the airport proximity predictor coefficient turned out to be statistically significantly negative (see Supplement). (2) Results from our Granger-causal inference indicate that, on average, the local, putatively causal connections are far stronger compared to the putatively causal connections between counties within which the major airports are located (see *Figure 4C and F*). Additionally, from our best mixed-effect regression model (*Figure 1A*), we find that land connectivity effects are significantly stronger than air connectivity effects. The predictive value of Twitter connectivity, which intuitively captures both local and long-distance travel, lies in-between land and air connectivity coefficients. Note that Twitter connectivity is represented as a directed graph, where for each pair of counties, $i$ and $j$, the $(i,j)$ edge weight represents the conditional probability of ending at county $j$, given that a traveler/Twitter user started her journey in county $i$. Transition probabilities from $i$ to $j$ sum to one over all $j$. Therefore, intuitively, the Twitter connectivity graph should have the features of both a land-connectivity and an air travel graph; which indeed appears to be close to reality. 3) While airport diffusion is a significant factor in our best Poisson model (using data from the initiation period), the causal streamlines (constructed with the complete, all-year incidence data) do not seem to originate from airport-bearing counties.

The role of short-distance travel is particularly crucial in explaining influenza's time-averaged, geo-spatial prevalence. While the mixed-effect regression analysis explains seasonal initiation in the vicinity of the continental US Southern shores, it might not, by itself, adequately explain its average prevalence patterns across the country.

Also not explained solely by our regression models is the occurrence of relatively high infection prevalence in the central parts of the country. These differences cannot be attributed to long-distance air travel, as discussed before. However, the routes taken by the causality streamlines (as computed by the non-parametric Granger analysis), interpreted as paths followed by an infection on average, suggest an explanation: The close match between the Granger-causal flow and the short-range mobility patterns (derived from Twitter analysis) strongly suggest that average disease prevalence is modulated by short-range mobility.

## Rationale for observational analysis

The traditional empirical approach of testing a causal link between a factor and an outcome of an experiment was to vary one factor at a time, while keeping the other factors (experimental conditions) constant. This 'all the rest of the conditions are equal' assumption is often referred to by its Latin form as *ceteris paribus*. R.A. Fisher ([*Fisher, 1935*], p. 18) noted that, in real-life experiments, perfect *ceteris paribus* is not achievable 'because uncontrollable causes which may influence the results are always . . . innumerable.' Fisher's proposed solution to this problem is to design experiments to involve random assignment of treatment (the putative causal factor's states) to individual trials and then use regression analysis to estimate the value and significance of the putative causal effect.

Likewise, hypothesis-driven science, wherein investigators formulate a single, testable hypothesis and design specific experiments to test it, is a core element of the scientific method, and works well in most scientific fields. However, a new challenge emerges in data-rich scientific fields, such as genomics, epidemiology, economics, climate modeling, and astronomy: How do we choose the most promising hypotheses among millions of eligible candidates that potentially fit data? One solution to this challenge is the many-hypotheses approach, a method of automated hypothesis generation in which many hypotheses are systematically produced and simultaneously tested against all

available data. This approach is currently used, for example, in whole-genome association or genetic linkage studies, and often enables truly unexpected discoveries. In contrast to the single-hypothesis approach, the many-hypotheses approach explicitly accounts for the large universe of possible hypotheses through calibrated statistical tests, effectively reducing the likelihood of accidentally accepting a mediocre hypothesis as a proxy for the truth (*Nuzzo, 2014*).

The many-hypotheses approach provides a complement to carefully controlled and highly focused wet laboratory experiments. Running controlled experiments to test a single hypothesis necessarily ignores many of the complexities of a real-world phenomenon; these complexities are necessarily present in large, longitudinal datasets. Of course, the data-driven 'many-hypotheses' approach is only one aspect of the broader scientific process progressing toward the development of verifiable general theories.

## Agreement and disagreement between methods

Intuitively, we expected that all three approaches would produce similar, if not identical results. In practice, while the three approaches agreed in most cases, this agreement was not perfect. For example, the highlighted areas (greater incidence) in the first influenza season snapshot for *Figure 2A–H* each should match relatively well to the maps in *Figure 6B* (or at least some unspecified subset of 'high incidence treatment counties'). While the county-matching results point to initiation at coasts, in *Figure 2*, 2006-2007 initiation seems to spread from the West Coast and, in , 2010 has a scattered pattern across the middle of the US.

The intuitive explanation of perceived discrepancy is that the matching method agrees with other analysis types predominantly, but not in all cases. Each analysis has limitations. In the case of the matching analysis, we have less statistical power than in, for example, Poisson regression; matching by numerous parameters reduces the initial set of thousands of counties to a handful of matching 'treated' counties (which meet a particular combination of weather and sociodemographic conditions) and 'untreated' counties (very similar to 'treated' ones in all respects but treatment). The difficult-to-match, 'weeded out' counties may happen to be in the coastal areas indicated as the most likely places of influenza wave origin by other analyses.

In the case of the 2007 and 2010 results, the matching analyses pick patterns that are different from those produced by the causality streamline analysis and mixed-effect Poisson regression models.

*Figure 6*'s Plate B shows the distribution of the treatment counties and matched-non-treatment counties. Note that here, we are not directly predicting initiation, so while the patterns in *Figure 2* and *Figure 6* should indeed show some similarity, they are not required to match up perfectly. The most similar treated counties do indeed show up in the Southern shores.

## Generalizability

Our analysis uses no prior knowledge specific to influenza epidemiology. As such, these methods are not limited by either the pathogen under consideration (influenza), or the geospatial context (United States). The tools developed here are expected to be equally applicable to analyzing general epidemiological dynamics for pathogens other than influenza, unfolding in arbitrary geographical regions. The specific conclusions we draw about the initiation and propagation of the seasonal influenza in US might not hold true for influenza epidemiology in a different geographical context. However, the analysis tools are still applicable. More broadly, our tools delineate a general approach to modeling complex spatio-temporal dynamics, with applications beyond solely disease epidemiology.

We conclude by highlighting the structure of overlapping conclusions delivered by our three approaches. Approach 1: Granger-causality analysis suggests that an epidemic tends to begin in the South, near water bodies and that short-range, land-based travel is more influential compared to air-travel for infection propagation, providing a map of mean infection flow across the continental US. Approach 2: Poisson regression identifies significant predictive factors, ranks these factors by importance, suggests that Southern shores are where the epidemic begins, and corroborates Approach 1's result on short-range vs. long-range travel. Approach 3 (county-matching): This approach drills down further to the epidemic onset source to the Southeastern shores of continental US, and identifies a smaller validated subset of predictive factors.

## Materials and methods

The methods and putative predictors identified in our study should be directly applicable to data outside the US. While the balance (relative importance) of putative triggers of influenza waves may vary, the factors themselves should be universal globally, because the virus and host biology are universal.

### Candidate factors in influenza initiation

To investigate county-specific variability, we grouped candidate factors into several categories: demographic, relation to human movement, infection state of county neighbors, and county's own recent state, and climatic.

*Major hypotheses* regarding putative causal factors affecting infection dynamics can be traced to a handful of earlier publications:

- Short- and long-range, work-related human movement (*Viboud et al., 2006*), including air travel (*Colizza et al., 2006*; *Balcan et al., 2009*; *Viboud et al., 2006*)
- Demographic confounders (*Gog et al., 2014*; *Charu et al., 2017*)
- Social contact among children in schools, or the 'Return-to-school effect' (*Gog et al., 2014*)
- Absolute humidity (*Shaman and Kohn, 2009*)
- Other climate variables (*Chowell et al., 2012*; *te Beest et al., 2013*)
- Host immunity, as affected by vaccination coverage, previous infections, and antigenic variation (*Centers for Disease Control and Prevention et al., 2009*).

### Human movement

We considered two measurements of human movement: (1) The first measurement reflects the proximity of counties to major airports. We computed an exponentially-diffusing influence from counties with major US airports, weighted by passenger enplanements at their respective locations. This accounted for people moving to and from both major airports and neighboring counties, and; (2) The second measurement is a large-scale movement matrix representing people's week-to-week travels between counties. These data were culled from a complete collection of geo-located Twitter messages, captured over 3.5 years, and constituted a large-scale, longitudinal sample of individual movements. We used only automatically geo-tagged tweets.

### Demographic

Influenza is transmitted through direct contact with infected individuals, via contaminated objects, and virus-laden aerosols. Thus, human population density (how many people happen to be around) and social connectivity (how many people interact with each other and how frequently [*Bedford et al., 2015*]) are factors expected to affect local virus incubation and spread. In addition to population density, we considered socioeconomic factors such as county-mean household income, levels of poverty and urbanity, as well as the prevalence of ethnic and age groups. All these socio-demographic and socio-economic data were derived from reports provided by the US Census ([Web Page]; 2017. Available from: https://www.census.gov/geo/reference/ua/uafacts.html).

### Return-to-school effect

Social contact among children in schools has been extensively investigated as a determinant of the peak incidence rate. This is one of the few factors that might lend itself to intervention relatively easily, and hence the interest is well-justified. While any reduction in social contact should, in theory, directly impact transmission, quantifying the effect of this specific mode of contact on the incidence rate has been difficult to calculate. Predictions of the reduction in the peak incidence associated with reduced social contact were typically 20–60% (*Ferguson et al., 2006*; *Haber et al., 2007*), with some studies predicting much larger reductions of 90% (*Mniszewski et al., 2008*; *Ghosh and Heffernan, 2010*). Reductions in the cumulative attack rate (AR, ratio of the number of new cases to the size of the population at risk) were usually smaller than those in the peak incidence. Several studies predicted small (~10%) or no reduction in the cumulative AR (*Ferguson et al., 2006*; *Haber et al., 2007*; *Yasuda et al., 2005*; *Ciofi degli Atti et al., 2008*; *Yasuda et al., 2008*; *Kelso et al., 2009*; *Davey et al., 2008*; *Rizzo et al., 2008*; *Vynnycky and Edmunds, 2008*; *Glass and Barnes, 2007*;

*Lee et al., 2010*; *Yang et al., 2011*; *Zhang et al., 2012*), whilst a few predicted substantial reductions (e.g. 90%) (*Glass et al., 2006*; *Davey et al., 2008*; *Elveback et al., 1976*; *Davey and Glass, 2008*). Only two studies (*Glass et al., 2006*; *Lee et al., 2010*) predicted that peak incidence might increase markedly under certain circumstances following school closures, for example by 27% if school closures caused a doubling in the number of contacts in the household and community, or by 13% if school systems were closed for two weeks at a prevalence of 1% in the general population. Studies have also investigated the effect of such interventions on children vs. adults; one study predicted an overall reduction in the cumulative AR, but an increase of up to 48% in the cumulative AR for adults in some situations (*Araz et al., 2012*).

While this diverse set of predictions in the literature often pertains to the effect of school closures as an intervention tool, we are more interested in the influence that the current school schedule has, if any, on triggering an epidemic. To answer this specific question, we formulated a simple statistical test to determine whether the timing of return-to-school after summer and winter holidays significantly predicts influenza season initiation. We found insufficient evidence in support of this effect (see Methods and materials).

## Climate variables

Specific humidity and a drop in temperature have been suggested as the key drivers in triggering seasonal influenza epidemics (*Lowen et al., 2008*; *Shaman et al., 2010*). These initial conclusions were drawn from experiments conducted using an animal model (guinea pig), under controlled laboratory conditions (*Lowen et al., 2008*), followed by indirect support from epidemiological modeling (*Shaman et al., 2010*).

## Vaccination coverage

Vaccination is widely regarded as our most promising tool to combat influenza, though antigenic variation between seasons makes it difficult to craft an effective vaccination strategy (*Boni, 2008*). Understanding how the virus will evolve in the short-term is key to finding the correct antigenic match for an upcoming influenza season. Additionally, short-term molecular evolution might rapidly give rise to immune-escape variants that, if detected, might dictate intra-season updates in vaccine composition. More importantly, vaccination itself might exert significant selection pressure to influence antigenic drift. The effect of vaccination on viral evolution has been documented in an avian H5N2 lineage in vaccinated chickens (*Lee et al., 2004*), suggesting that similar processes might be occurring in human counterparts. The diversity of the surface proteins at any point in time between seasons suggests that our current vaccination strategies are limited to confer partial immunity, which can result in a highly immune or vaccinated population selectively pressurizing the viral population to evolve more quickly than usual. Given that influenza moves quickly across geographies and that there are multiple, co-circulating strains that may confer partial cross-protection, changing viral service proteins represent a 'moving target' for the human immune system. There are additional complexities due to early-life immune imprinting in humans.

## Accounting for non-vaccination host immunity

Resistance to influenza infection in human hosts arises via two related mechanisms: immunological memory from a previous infection by an identical or sufficiently similar strain, or vaccination against the current strain (or a sufficiently similar strain). In our analysis, we explicitly account for the degree of vaccination coverage. Accounting for host immunity is more difficult, because resistance to infection is not directly observable; however, our analysis also uses the antigenic variation of influenza virus as a proxy to host resistance: If the relative antigenic variation is large, then the susceptibility of non-vaccinated hosts increases, while low antigenic variation increases the probability of encountering a resistant host. In addition, we use the absolute antigenic deviation of the later-season virus from the first-season virus in our data set (winter of 2003–2004) as a predictor, in order to capture longer-term effects of immunological memory. Thus, if the magnitude of the relative drift between two succeeding years is small, we have a likely decrease in susceptibility. Likewise, if the absolute deviation from a few years back starts decreasing, we would also register a decrease in susceptibility. Incorporating these factors in the diverse set of models that we investigate, guarantees that we are indeed considering host susceptibility contributions from a wide range of possible mechanisms.

## Antigenic Variation

The influenza virus counteracts host immunity via subtle genomic changes over time. The more gradual process, known as antigenic drift, is a manifestation of a gradual accumulation of mutations within viral surface proteins recognized by human antibodies, such as hemagglutinin (HA) and neuraminidase (NA). These mutations are typically introduced during cycles of viral replication (*Boni et al., 2004*). Most of these mutations are neutral, *i.e.* they do not affect the functional conformation of the viral proteins. However, some of these alterations do, in fact, sufficiently change secondary and tertiary protein structures to have a negative impact on the binding of host antibodies raised in response to previously circulating strains (*Webby and Webster, 2001*). (Many such mutations also reduce the virus's viability.) Thus, while a single infection episode is potentially enough to provide long-term host immunity to the invading strain, antigenic variation due to intense selection pressure gives rise to novel viral strains, making re-infections possible within the span of a few years (*Andreasen, 2003*). This kind of perpetual Red-Queen arms race injects influenza dynamics with auto-correlative dependencies over multiple seasons. It has been suggested that substantial antigenic drift might be associated with more severe, early-onset influenza epidemics, resulting in increased mortality (*Treanor, 2004*). In contrast to antigenic drift, *antigenic shift* is an abrupt, major change in virus structure due to gene segment re-assortment that occurs during simultaneous infection of a single host by multiple influenza subtypes (*De Clercq, 2006*). Antigenic shift results in new versions of viral surface proteins. Antigenic shift due to re-assortment gives rise to novel influenza subtypes that, if capable of sustained human-to-human transmission, can have devastating consequences for human populations, *e.g.* the 2009 H1N1 pandemic (*Neumann et al., 2009*).

In our analysis, we factor in the potential effect of antigenic variation by estimating the surface protein's population diversity – hemagglutinin (HA), neuraminidase (NA), matrix protein 1 (M1), and matrix protien M2 – as a function of time and geographical sample collection location. Our rationale for our focus on these proteins is that HA, NA, and to some degree M1, are all present on the viral surface (*Lamb et al., 1985*), contribute to viral assembly, and mediate the release of membrane-enveloped particles (*Chlanda et al., 2015*). M2 has been shown to have enhanced the pandemic 2009 influenza A virus [(H1N1)pdm09] (*Friedman et al., 2017*) HA-pseudovirus infectivity (*Alvarado-Facundo et al., 2015*).

## Data sources

### Social connectivity

Data was obtained from the General Social Surveys, NORC, See *Table 1*, [].

### Clinical data source

The source of the clinical incidence data used in this study is the Truven Health Analytics MarketScan® Commercial Claims and Encounters Database for the years 2003 to 2012 (*Hansen, 2017*). The database consists of approved commercial health insurance claims for between 17.5 and 45.2 million people annually, with linkage across years, for a total of approximately 150 million individuals. This US national database contains data contributed by over 150 insurance carriers and large, self-insuring companies. We scanned 4.6 billion inpatient and outpatient service claims and identified almost six billion diagnosis codes. After un-duplication, we identified approximately 12.8 unique diagnostic codes per individual. We processed the Truven database to obtain the reported weekly number of influenza cases over a period of 471 weeks spanning from January 2003 to December 2013, at the spatial resolution of US counties. To define influenza in insurance claims, we used the following set of ICD9 codes: 487.8, 488.12, 488.1, 488.0, 488.01, 488.02, 487.0, 487.1, 488.19, 488.09, 488, 487, and 488.11. We also considered including non-specific codes representing *unspecified viral infection* (079.99), as suggested in (*Viboud et al., 2014*), and decided against it. (We provide explicit annotation for each code, as well as rationale for choosing narrower ICD-code definition of ILIs, in the Supplement Section S-D).

A peak percentage of ILI-affected people of 20% may seem high–especially given that it is a much greater percentage than the change in seropositivity over most influenza seasons. Note that our *apparent* estimates of influenza infection rate are unavoidably inflated, because what we are measuring is the week- and county-specific prevalence of influenza-like illnesses *among insured patients who contacted their physician for any reason during the week in question*. In other words,

the denominator that we used for computing prevalence was almost certain to be much smaller than the overall population of the corresponding county.

## Data on antigenic drift for influenza A

Sequence data for this computation was obtained from the National Institute of Allergy and Infectious Diseases (NIAID) Influenza Research Database (IRD) (*Zhang et al., 2017*) through their web site at http://www.fludb.org.

## Data on vaccination coverage

Data on vaccinations was extracted from our EHR database corresponding to the procedural codes 90661 and Q2037, which corresponds to the dominant influenza vaccines. (http://flu.seqirus.com/files/billing_and_coding_guide.pdf)

## Data on human air travel

We used a complete, directed graph of passenger air travel for 2010, accounting for the number of passengers transported in each direction (*The United States Bureau of Transportation Statistics, 2010*). For each county, we computed an air neighborhood network: For counties $i$ and $j$, the incoming edge to county $i$ represents the proportion of passengers, and $p_{ji}$ represents the ratio of all passengers who traveled from county $i$ to county $j$ by plane, mto the total number of travelers who left county $i$ by plane during the year, so that $\sum_{j \neq i} p_{ji} = 1$.

## Data on general human travel patterns in the US

Using the complete Twitter dataset, we aggregated a movement matrix to capture people's week-to-week travels between counties from geo-located Twitter messages (captured during the period of January 1, 2011 through June 30, 2014). This dataset includes approximately $1.7 \times 10^9$ messages and represents $3 \times 10^8$ user-days of location information. A small, but significant, percentage of Twitter messages are automatically tagged with the author's current latitude/longitude information, as determined by their mobile device. Each latitude/longitude-annotated tweet was mapped to a FIPS county code based on Tiger/Line shape files from the 2013 Census dataset (http://www.census.gov/geo/maps-data/data/tiger-line.html). In addition, we calculated a variant of our movement matrix to capture seasonality and other temporal dynamics: A set of 52 movement matrices captured weekly, county-to-county movements, observed in each week of the year, aggregating each week based on the corresponding observations, from each year from 2011 through 2014.

This dataset constitutes a large-scale, longitudinal sample of individual movements. We found that the movement patterns are consistent with intuitive patterns and prior studies of large-scale movements; the majority of people in a county remain in the same county week-over-week. Most travel between counties occurs between neighboring counties, and between counties and large metropolitan areas, conditioned on distance and size of the metropolitan area. We represent our movement data as an $n \times n$ matrix $\mathbf{M}$ that captures the likelihood that a person observed in county $i$ during the course of a week will be observed in county $j$ in the following week. We calculate the entries $m_{i,j}$ of matrix $\mathbf{M}$ as follows:

$$m_{i,j} = \sum_{t<T} \frac{1}{Pop_{i,t}} \sum_{u<m} x_{u,i,t} x_{u,j,t+1} \tag{1}$$

For each time interval, we computed the mean movement vector from a given county by summing all county-to-county movement vectors, weighted by the proportion of people moving in each direction.

To investigate the role of proximity to major airports, we modeled influence diffusion as follows: Let $x_i$ be the $i^{th}$ county, and $v_i$ be the total contribution obtained by diffusing influences from the major airport bearing counties. Let $x_k^\star$ be the $k^{th}$ airport-bearing county, and let $N$ be the total number of major airport locations considered, in our case, $N = 27$. Let $g_k$ be the volume of traffic for the $k^{th}$ major airport location. We then compute $v_i$ as follows:

$$v_i = \sum_{k=1}^{N} g_k e^{-C_0 \Theta(x_k^\star, x_i)} \tag{2}$$

where $\Theta(\cdot, \cdot)$ is the distance in miles between two locations, we computed using the Haversine approximation. The value of the constant $C_0$ was chosen to be 0.1. Small variations in the constant do not significantly alter our conclusions. As noted above, proximity to airports had a significant positive influence in sparking seasonal epidemics; the influence is significantly weaker once an outbreak is well under way.

## Estimating antigenic diversity

In this study, we measured antigenic diversity as follows: Let $S_{i,x,t}$ be the set of amino acid sequences for the $i^{th}$ protein (one of HA, NA, M1, or M2), collected in year $t$ ($t$ ranging between 2003 and 2011), in state $x$ of the continental US. The temporal resolution of the sequence data is thus set to years instead of weeks, and the spatial resolution to states instead of counties. These resolutions are coarse compared to our EHR data on infection incidences, and is set in this manner to maintain sufficient statistical power. For each such set of amino acid sequences $S_{i,x,t}$, we compute the set $D(S_{i,x,t})$ of pairwise edit distances:

$$D(S_{i,x,t}) = \{y : y = \mathcal{L}(s_1, s_2), s_1, s_2 \in S_{i,x,t}, s_1 \neq s_2\} \tag{3}$$

where $\mathcal{L}(s_1, s_2)$ is the standard edit distance (also known as the Levenshtein distance) between the sequences $s_1, s_2$. Mathematically, the Levenshtein distance between two strings $a, b$ (of length $|a|, |b|$ respectively) is given by:

$$\mathcal{L}(a, b) = lev_{a,b}(|a|, |b|), where \tag{4}$$

$$lev_{a,b}(i,j) = \begin{cases} \max(i,j) & \text{if } \min(i,j)=0, \\ \min \begin{cases} lev_{a,b}(i-1,j)+1 \\ lev_{a,b}(i,j-1)+1 \\ lev_{a,b}(i-1,j-1)+1_{(a_i \neq b_j)} \end{cases} & \text{otherwise.} \end{cases} \tag{5}$$

where $1_{(a_i \neq b_j)}$ is the indicator function equal to 0 when $a_i = b_j$ and equal to one otherwise, and $lev_{a,b}(i,j)$ is the distance between the first $i$ characters of $a$ and the first $j$ characters of $b$.

The antigenic diversity of the $i^{th}$ protein at time $t$ in state $x$ was then defined as the median of the distribution of the values in the set $D(S_{i,x,t})$. Clearly, as the sequences became more diverse at a point in time and space due to molecular variations brought about by either drifts or shifts, the measure deviated more from zero. Use of the median provided robustness to outliers.

Importantly, here we did not compare temporal changes in viral antigenic makeup directly, but estimated the time-specific diversity in the viral protein primary structure. We expected our diversity measure to be representative of the cumulative changes that occurred within each of the nine influenza seasons.

## Estimating vaccination coverage

We incorporated the effect of vaccination coverage by estimating the cumulative fraction of the population that received the current influenza vaccine within the previous 20 week period. This approach to estimating vaccination coverage does not correct for season-specific antigenic match, or the lack thereof. Nevertheless, because we explicitly included measures of antigenic diversity in addition to vaccination coverage, we expected that effects arising from the degree of antigenic match would indeed be factored in; if there is a significant mismatch, we expected the antigenic diversity in that year to be less and vice verse. This assumes implicitly that vaccination does indeed play a major role in exerting significant selection pressure, the assumption that was reinforced by our observation of a strong dependency between vaccination coverage and normalized antigenic diversity.

We found that antigenic diversity is quite strongly affected by vaccination coverage. This reflects the theoretical predictions in Boni et al. (*Boni et al., 2006*), where it is shown that the amount of

observed antigenic drift increases as immunity in the host population increases and pressures the virus population to evolve.

## Estimating the effect of return-to-school days

The absence of consensus, and the diversity of modeling assumptions pertaining to this effect (described above) makes it difficult to validate conclusions in large scale epidemiological data. We carried out a simple test to determine whether there is statistical evidence that after-holiday, return-to-school periods in August-September and in January predict or trigger the seasonal epidemic.

For this test, we assumed a broad window to cover all such school openings across the continental US including the last week in August, the entire month of September, two weeks in October, and two weeks in January. We next carried out a Fisher's exact test to determine whether the overlap between these weeks and the identified 'trigger period' (See Table LABEL:SI-tabschool) for the seasonal epidemic are sufficiently non-random.

## Weather data

The dataset starts with the week beginning December 31 st, 2002 and includes 522 weeks (which ends exactly on the week ending December 31 st, 2012). Temperature and precipitation data come from the 2.5 arcminute (approximately 4 km) PRISM (*Oregon State University, 2014*) dataset and other variables (wind speed, specific humidity, surface pressure, downward incident, and shortwave solar radiation) come from the 7.5 arcminute (approximately 12 km) Phase 2 North American Land Data Assimilation System (NLDAS-2) dataset (*Mitchell, 2004*; *Cosgrove et al., 2003*). These datasets were selected in large part due to the fact that both are updated in near real-time, making it possible to use these datasets for future monitoring applications. PRISM is released daily, with an approximately 13 hr delay (data for the previous day is released at 1pm EST each day) while NLDAS is released daily, with an approximately 2.5 day delay).

Variables were aggregated to county boundaries based on shapefiles from the GADM database of Global Administrative Areas (*Areas, 2014*). Where appropriate, we considered both the average daily climate variable (for example, the daily maximum temperature averaged over the week) as well as the the maximum and/or minimum of the variable experienced over the week. For precipitation, we considered only the cumulative total precipitation experienced during the week.

## Three approaches

### Approach 1: Non-parametric Ganger analysis of county-specific incidence

#### Analysis of quantized clinical data

For the causality analysis, we started with an integer-valued time series for each US county, and to carry out the causality analysis, we first quantized the series in two steps:

1. 1. Computing the difference series, *i.e.*, the weekly change in the number of reported cases.
2. 2. Mapping positive changes to symbol '1' and negative changes to symbol '0'.

This mapped each data series to a symbol stream over a binary alphabet. The binary quantization is not a restriction imposed by the inference algorithm; while we do require quantized magnitudes, longer data streams can be encoded with finer alphabets to accommodate an arbitrary precision. For this specific dataset, the relatively short length of the county-specific time-series necessitated a coarse quantization in order for the results to have a meaningful statistical significance.

Given a pair of such quantized streams $s_a, s_b$, the algorithm described in the Supplement computes two coefficients of causality, one for each direction. Intuitively, the coefficient $\gamma_b^a$ from $s_a$ to $s_b$ is a non-dimensional number between 0 and 1 that quantifies the amount of information that one may acquire about the second stream $s_b$ from observing the first stream ($s_a$). More specifically, $\gamma_b^a$ is the average reduction in the uncertainty of the next predicted symbol in stream $s_b$ in bits, per bit, acquired from observed symbols in stream $s_a$. It can be shown that the coefficient of causality $\gamma_b^a$ is 0 if and only if there is no causal influence from $s_a$ to $s_b$ in the sense of Granger, and assumes the maximum value 1 if and only if $s_b$ is deterministically predictable from $s_a$. Moreover, $\gamma_b^a = \gamma_a^b = 0$ if and only if $s_a$ and $s_b$ are statistically independent processes (*Chattopadhyay, 2014*). It is trivial to produce examples where we would have $\gamma_b^a = 0, \gamma_a^b > 0$ illustrating the ability of the algorithm to capture the asymmetric flow of causal influence in one preferred direction and not the other.

Additionally, whenever we computed the causality coefficient, it was always associated with a time delay: We calculated the coefficient for predicting the target stream some specified number of steps in the future, and the computed coefficient was thus parameterized by this delay. In our analysis, we computed coefficients up to a maximum delay of 10 weeks, and, for each pair of counties, selected the optimum delay which gave rise to the largest coefficient. For more details on the algorithm, see the Supplement.

## Computation of causality fields and causality streamlines

To analyze the propagation dynamics of disease, we defined the notion of a Granger *causal flow* between two counties. Treating county-specific changes in disease prevalence as a time-stamped data stream, we quantified the directional strength of the causal flow between two counties as the degree of predictability of one stream's future, given the history of the other (see *Figure 3A–D*). Our new algorithm was able to compute a *coefficient of causality*: an information-theoretic measure of the information in bits that one stream communicates about another in a direction-specific manner. A strong coefficient of this type suggests that either the disease itself, or an underlying cause, may be propagating from the former county to the latter.

To explore the country-wide propagation dynamics, we stitched together the properly-aligned, adjacent between-county causal flow vectors across the whole US map into *causality streamlines*, representing the average of multiple spatial infection propagation waves over time. To compute causality streamlines, we needed a precise notion of county neighbors. We considered two counties to be neighbors if either they shared a common border, or if one county was reachable within a line-of-sight distance of 50 miles from any point within another. The latter condition removed ambiguities as to whether counties touching at a point should be still considered as neighbors. The exact distance value (50 miles) does not significantly change our results, as long as it does not vary significantly. With the definition of the neighborhood map in place, we proceeded to compute the direction-specific coefficients of causality between neighboring counties. It follows that we would obtain a set of coefficients for each county and one for each of its neighbors, capturing the degree of causal influence from a given particular county to its respective neighbors. Our algorithm also computed the probability of each coefficient arising by chance alone, and we ignored coefficients that have more than a 6% probability that two independent processes lacking any causal connection gave rise to the particular computed value of the coefficient. Once the coefficients had been calculated for each neighbor, we computed the resultant direction of causal influence outflow from that particular county. This was carried out by visualizing the causality coefficients as weights on the length of the vectors, from the centroid of the considered county to the centroids of its neighbors. We then calculated the resultant vector (see *Figure 3*). Viewed systematically across the continental US, these local vectors formed a discernible pattern; we observed the emergence of a non-random structure with clearly discernible paths outlining the 'causality field' (see *Figure 3*, Plates G, J, K). To interpret the plots, note that streamlines start at their thinnest part, their direction is indicted with thickening line; typically multiple streamlines coalesce into a river-like pattern.

## Inferring statistical 'Granger-causality' from data

Granger attempted to obtain a precise definition of causal influence (*Granger, 1980*) from the following intuitive notion: $Y$ is a cause of $X$, if it has unique information that alters the probabilistic estimate of the immediate future of $X$.

Here, we used a new, non-parametric approach to Granger causal inference (*Chattopadhyay, 2014*) (Approach 1). In contrast to state-of-the-art binary tests (*Baek and Brock, 1992*; *Hiemstra and Jones, 1994*), we computed the degree of causal dependence between quantized data streams from stationary ergodic sources in a non-parametric, non-linear setting.

Our approach was significantly more general to common, regression-based implementations of Granger causal inference, and did not involve any autoregressive moving average (ARMA) modeling. All such commonly used techniques impose an *a priori* linear structure on the data, which then constrains the class of dependency structures we can hope to distill.

True causality, in a Humean sense, cannot be inferred (*Hume, 1993*; *Kant, 1998*). Among other reported approaches to ascertaining causal dependency relationships, the work of J. Pearl (*Pearl, 2009a*) is perhaps most visible, and builds on the paradigm of structural causal models (SCM) (*Pearl, 2009b*). While Pearl's work is often claimed to be able to answer causal queries regarding

the effects of potential interventions, as well as regarding counterfactuals, our objective in this paper is somewhat different. We are interested in delineating whether infection transmission pathways can be distilled from the patterns of infection's spatio-temporal incidence.

## Approach 2: Poisson regression on putative factors

In Approach 2, we investigated the relative importance of putative factors as follows: Specifically, let $N_{ijk}$ denote the total number of patients in county $i$, who are of age $j$, and gender $k$. Denoting the number of individuals diagnosed with influenza in a given county during given week as $y_{ijk}$, we modeled the within-county disease incidence counts for every county (for which data was available) in the US using the following mixed-effect Poisson regression model:

$$P(y_{ijk}|\lambda_{i,j,k}) = \frac{\lambda_{ijk}^{y_{ijk}} e^{-\lambda_{ijk}}}{y_{ijk}!} \qquad (6)$$

$$with \, \lambda_{ijk} = N_{ijk} \exp\left\{\alpha + \mathbf{Xb} + \mathbf{s}_i \mathbf{b}_{1,i} + \mathbf{n}_i \mathbf{b}_{2,i} + \mathbf{Zv}\right\},$$

where, $\alpha$ is the intercept, $\mathbf{X}$ and $\mathbf{b}$ are a fixed-effect design matrix and a vector of fixed effects, respectively, $\mathbf{s}_i$ is $1 \times m$ vector of changes in rate of infection in the $i^{th}$ county $1, 2, .., m$ weeks prior to the current, $\mathbf{b}_{1,i}$ is a $m \times 1$ vector of auto-correlation fixed effects, $\mathbf{n}_i$ is a $1 \times (3m)$ vector of changes in the rate of infection in the neighbors of the $i^{th}$ county $1, 2, .., m$ weeks prior to the current (the neighbors are subdivided into *land* neighbors, *Twitter* neighbors, and *air* neighbors), $\mathbf{b}_{2,i}$ is a $(3m) \times 1$ vector of county-neighborhoods fixed effects, and $\mathbf{v}$ and $\mathbf{Z}$ are random effects and their design matrix, respectively. Variable $m$ represents the depth of 'memory' of auto-regression in weeks, in our case $m = 4$.

In this way, the total number of rows in matrix $\mathbf{X}$ is $510 \times 3,143$ (weeks $\times$ counties), with county-specific socioeconomic covariates and week-specific weather covariates. For disease initiation analysis, we included only a subset of time series covering approximately 50 weeks.

We did not use an explicit spatial smoothing. The county-level random effects were used to account for possible heterogeneity in disease reporting across counties. However, as predictors, we used average infection rate values of immediate neighbors for the county (one, two, three, and four weeks in the past). This prediction structure afforded an implicit spatial smoothing, dampening spurious fluctuations in infection rates.

### Out-of-sample prediction and ROC analysis with mixed-effect poisson regression

We carried our out-of-sample prediction with the models inferred with mixed-effect regression. The steps were as follows:

1. We trained the model parameters with data from the trigger periods corresponding to the first six seasons.
2. Once we identified the coefficient's variables, we used it to predict the response variable (influenza incidence) for the last three seasons.

As expected, the predicted incidence does not exactly match the observed out-of-sample data. Nevertheless, we see positive correlation (*Figure 5*, Plate A). Because we were modeling a necessarily spatio-temporal stochastic process, the predictive ability of the model is difficult to judge simply from the observed positive correlation. To resolve this, we investigated the performance of our model by computing how well it predicted the counties that experience flare-ups during the trigger-periods in the out-of-sample data. This exercise is reduced to a classification problem, by first choosing a threshold on the number of reported cases per week to define what is meant by a 'flare-up' (see description below). To quantify the prediction performance, we constructed ROC curves for each of the three target seasons, for each fixed week, as follows:

1. We first quantized the incidence data to reduce it to a binary stream. In particular, we chose a threshold (ten), such that for each county, and each week, we reported a '1' if the number of reported cases was greater than the threshold, and '0' otherwise. Note this quantization is different to what we used in carrying out our non-parametric Granger analysis in Approach 1.

2. For each county $i$, and week $j$, we then ended up with the binary class variable $X_{j,i} \in \{0, 1\}$ and a decision variable $Y_{j,i} \in \mathbb{R}$, where the former is the quantized incidence described above, and $Y_{j,i}$ is the response predicted by the model, normalized between 0 and 1.

3. For a chosen decision threshold $\theta_D \in \mathbb{R}$, we could determine the predicted class $\widehat{X}_{j,i} \in \{0, 1\}$ as:$\widehat{X}_{j,i} = \begin{cases} 1, & if\, Y_{j,i}\theta_D \\ 0 & otherwise \end{cases}$

4. Comparing the observed and predicted classes, we computed the false positive rate (FPR: defined as the ratio of false positives to the sum of false positives and true negatives), and the true positive rate (TPR: defined as the ratio of true positives to the sum of true positives and false negatives). Finally, we constructed the ROC curve, which shows the relationship of the TPR and the FPR as the decision-threshold $\theta_D$ is varied.

5. We constructed Receiver Operating Characteristic (ROC) curves for each week in the out-of-sample period, and estimated the area under curve (AUC). The AUC measures the performance of the predictor (our model) to correctly classify the counties that would go on to have a disease incidence greater than the initial set threshold (ten cases in our analysis). In the perfect case, we would have an AUC of 100%, which implies that we can achieve zero false positives, while getting a 100% true positive rate. Our best model achieves approximately 80% AUC for the trigger weeks, as shown in *Figure 5*.

## Variable and model selection

Approaches 2 and 3 have different limitations; these affect which approach is most effective at which statistical detection task.

Approach 2 (mixed-effect regression analysis) can use a whole dataset for model selection, that can span individual predictors, their combinations, and even interactions between factors (see *Table 5*; *Figure 1—figure supplement 2*; *Figure 1—figure supplement 3*; *Figure 1—figure supplement 4*; *Source data 2*; *Source data 3*; *Supplementray file 3*; *Source code 1*). Model selection allows us to choose a model with the right balance of complexity and explanatory power, thus enabling this analysis to detect collinear explanatory variables and drop the weaker predictors that increase model complexity but do not add explanatory power.

For example, the best model in our mixed-effect regression model series, with deviance information criterion (DIC) $185,926.6$ (see *Table 5*, the last model), includes a term d_max_HUS_min_3 * d_t_avg_min_3 that denotes an interaction between the weekly change in the maximum specific humidity and the weekly change in the average temperature, both of which are weather parameter changes recorded three weeks before the current week. Because it does not split datasets, Approach two is more powerful at detecting interactions between explanatory variables and rather complicated models. However, this approach is more susceptible to bias (picking non-causal correlations that are inherent in the raw data), if the data are unbalanced, unlike with Approach 3.

We added and removed random effects to the model structure, taking out and adding fixed effects, executing the regression algorithm, and plotting the DIC achieved against model complexity. Here, model complexity is simply proxied by the descriptional complexity in terms of how large the model was (the number of factors in the regression equation). We stopped when the drop in DIC stabilized (See *Figure 1—figure supplement 3* ). Notably, the DIC has been shown to select overfits (*Ando, 2011*; *Plummer, 2008*; *van der Linde, 2012*), and does not properly account for model complexity in practice. Our driving idea was the identification of the Pareto front that trades off accuracy (in terms of DIC) with model complexity in a transparent manner. This is, of course, a 'greedy approach,' in the sense that we do not guarantee that we have indeed found the 'best' possible model. However, because our cross-validation (*Figure 5*) yielded good results, we deemed the stopping rule to be satisfactory.

## Approach 3: County-matching effect analysis

We designed Approach three specifically to balance biased observational data. While Approach three does not allow for explicit model selection, it is good at detecting simpler combinations of predictors putatively 'causally' affecting the outcome variable. However, because this approach involves splitting data into smaller and smaller subsets, as the complexity of predictor function ('treatment') grows, it quickly runs out of statistical power.

These two approaches' properties explain the perceived misalignment of results from the two streams of analysis. For an intuitive understanding of matching approaches, consider specific humidity's effect on influenza prevalence. The county-matching method's goal is to deduce associations putatively interpreted as causality relations. For example, consider testing the question of whether counties with higher-than-average mean maximum specific humidity do, indeed, have higher influenza prevalence, in a statistically significant sense, when compared to counties that do not, provided all other factors are held constant.

Let $Y$ denote the set of all US counties, and let $S$ be the set of all factors we find to be significant in our mixed-effect regression analysis. Now, for any subset of factors $K \subseteq S$, we denote the complement set as $\overline{K} = S \setminus K$. Additionally, we define the Boolean function $\mathcal{T} : S \times Y \rightarrow \{true, false\}$, (the treatment function) where for some factors $s \in S$, and some counties $y \in Y$, the Boolean value $\mathcal{T}(s, y)$ is true if the signal or treatment corresponding to the factor $s$ is present in county $y$. We used the sign for the coefficients we obtained in our best mixed-effect regression model to determine what counts as a positive signal. For example, because maximum specific humidity (denoted by the variable $max\_hus\_avg$) enters with a positive coefficient, if county $y$ experiences a higher-than-average maximum specific humidity, then $\mathcal{T}(max\_hus\_avg, y)$ is true.

To simplify the notation, we used $\mathcal{T}_y^s$ to denote a true signal, and $\neg \mathcal{T}_y^s$ to denote a false one.

Finally, for any $K \subseteq S$, we define the following three sets, which we refer to as the $W$-sets:

$$W_{treated}^K = \left\{ y \in Y : \bigwedge_{k \in K} \mathcal{T}_y^k \right\} \tag{8}$$

$$W_{matched-control}^K =$$

$$\left\{ y \in Y : \left( \neg \bigwedge_{k \in K} \mathcal{T}_y^k \right) \bigwedge \left( \exists y' \in W_{true}^K \left( \forall r \in \overline{K} \left( (\mathcal{T}_{y'}^r \wedge \mathcal{T}_y^r) \vee (\neg \mathcal{T}_{y'}^r \wedge \neg \mathcal{T}_y^r) \right) \right) \right) \right\} \tag{9}$$

$$W_{other}^K = \left\{ y \in Y : \left( \neg \bigwedge_{k \in K} \mathcal{T}_y^k \right) \right\} \setminus W_{matched-control}^K \tag{10}$$

Clearly, $W_{treated}^K$ is the treated set, that is, the set of counties which exhibit the signal encoded by the set $K$ is then the matched control set of counties, which lack the signal, but each county in this set has a matching counterpart in $W_{treated}^K$.

The $W$-sets allow us to set up a $2 \times 2$ contingency table for any chosen subset of factors $K \subseteq S$ as described before. Specifically, we split the sets $W_{treated}^K$ and $W_{matched-control}^K$ into two subsets each, representing those counties which experience a spike in influenza prevalence and which do not. The $2 \times 2$ contingency table is then subjected to Fisher's exact test.

## Acknowledgements

We thank Erin Gannon, Mark Jit, Prabhat Jha, Elizabeth C Lee, and Margarita Rzhetsky for numerous comments on earlier versions of the manuscript. This work was supported by NIH grants 1P50MH094267, U01HL108634-01, R01HL122712-02, R01GM100467, and U01GM110748, in addition to DARPA contract W911NF1410333, and a gift from Liz and Kent Dauten.

## Additional information

### Funding

| Funder | Grant reference number | Author |
| --- | --- | --- |
| National Institutes of Health | U01HL108634-01 | Jeffrey L Shaman |
| National Institutes of Health | R01GM100467 | Jeffrey L Shaman |
| National Institutes of Health | U01GM110748 | Jeffrey L Shaman |

| National Institutes of Health | 1P50MH094267 | Andrey Rzhetsky |
| Defense Sciences Office, DAR-PA | W911NF1410333 | Andrey Rzhetsky |
| National Institutes of Health | R01HL122712 | Andrey Rzhetsky |

The funders had no role in study design, data collection and interpretation, or the decision to submit the work for publication.

## Author contributions

Ishanu Chattopadhyay, Conceptualization, Resources, Data curation, Software, Formal analysis, Validation, Investigation, Visualization, Methodology, Writing—original draft, Writing—review and editing; Emre Kiciman, Formal analysis, Methodology, Writing—original draft, Writing—review and editing; Joshua W Elliott, Resources, Data curation, Formal analysis, Methodology, Writing—review and editing; Jeffrey L Shaman, Methodology, Writing—review and editing; Andrey Rzhetsky, Conceptualization, Resources, Software, Formal analysis, Supervision, Funding acquisition, Validation, Investigation, Visualization, Methodology, Writing—original draft, Project administration, Writing—review and editing

## Author ORCIDs

Ishanu Chattopadhyay http://orcid.org/0000-0001-8339-8162
Emre Kiciman http://orcid.org/0000-0001-5429-468X
Andrey Rzhetsky http://orcid.org/0000-0001-6959-7405

## Decision letter and Author response

Decision letter https://doi.org/10.7554/eLife.30756.035
Author response https://doi.org/10.7554/eLife.30756.036

# Additional files

## Supplementary files

• Source code 1. R code for analyses in Approach 2: mcmc_flu.r Raw outputs of the regression experiments in Approach 2: R Output Logs: models computed using only weeks associated with initiation of influenza wave.
DOI: https://doi.org/10.7554/eLife.30756.022

• Source code 2. County-matching code for Approach 3: counterfactual.cc
DOI: https://doi.org/10.7554/eLife.30756.023

• Source data 1. Streamline raw data for *Figure 4G*: streamline_data.zip Each streamline is a a series of physical coordinates (latitude and longitude in degrees), with a linebreak after each streamline.
DOI: https://doi.org/10.7554/eLife.30756.024

• Source data 2. Analysis of 50% of all weeks in study with mixed-effect Poisson regression: flu-50-percent-weeks.txt
DOI: https://doi.org/10.7554/eLife.30756.025

• Source data 3. Raw outputs of all mixed-effect regression analyses in this study.
DOI: https://doi.org/10.7554/eLife.30756.026

• Supplementary file 1. Summarized Algorithms for Non-parametric Granger Causal Inference: SI2.docx
DOI: https://doi.org/10.7554/eLife.30756.027

• Supplementary file 2. Back to School Effect Analysis.
DOI: https://doi.org/10.7554/eLife.30756.028

• Supplementary file 3. Regression Equations and Corresponding DIC values.
DOI: https://doi.org/10.7554/eLife.30756.029

• Supplementary file 4. Regression Equations for all models tested in our regression analysis along with corresponding DIC values. Balancing Precision and Sensitivity when Identifying Influenza-like Illnesses from Electronic Medical Records.

DOI: https://doi.org/10.7554/eLife.30756.030

• Transparent reporting form
DOI: https://doi.org/10.7554/eLife.30756.031

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

## Appendix 1

DOI: https://doi.org/10.7554/eLife.30756.032

# Non-parametric Granger Causal Inference

## Standard Implementations of Granger Causal Inference

Suppose that one is interested in the possibility that a vector series $Y_t$ causes another vector $X_t$. Let $J_n$ be an information set available at time $n$, consisting of terms of the vector series $Z_t$, that is,

$$J_n = \{Z_t : tn\} \tag{11}$$

$J_n$ is said to be a proper information set with respect to $X_t$, if $X_t$ is included within $Z_t$. Further, suppose that $Z_t$ does not include any component of $Y_t$, and define

$$J'_n = \{(Z_t, Y_t) : tn\} \tag{12}$$

Inferring causality in the mean is easier, and if one is satisfied with using minimum mean square prediction error as the criterion to evaluate incremental predictive power, then one may use linear, one-step-ahead least squares predictors to obtain an operational procedure: If $VAR(X|J_n)$ is the variance of one-step forecast error of $X_{n+1}$ given $J_n$, then $Y$ is a prima facie cause of $X$ with respect to $J'_n$ if:

$$VAR(X|J'_n) < VAR(X|J_n) \tag{13}$$

Testing for bivariate Granger causality *in the mean* involves estimating a linear, reduced-form vector autoregression:

$$X_t = A(L)X_t + B(L)Y_t + U_{X,t} \tag{14}$$

$$Y_t = C(L)X_t + D(L)Y_t + V_{Y,t} \tag{15}$$

where $A(L)$, $B(L)$, $C(L)$, and $D(L)$ are one-sided lag polynomials in the lag operator $L$ with roots all-distinct, and outside the unit circle. The regression errors $U_{X,t}$, $V_{Y,t}$ are assumed to be mutually independent and individually i.i.d. with zero mean and constant variance. We used a standard joint test ($F$ or $\chi^2$-test) to determine whether lagged $Y$ has significant linear predictive power for current $X$.

The null hypothesis that $Y$ does not strictly Granger cause $X$ is rejected if the coefficients of the elements in $B(L)$ are jointly significantly different from zero.

Linear tests presuppose restrictive and often unrealistic (**Darnell and Evans, 1990**; **Epstein, 1987**) structure on data. Brock (**Brock, 1991**) presents a simple bivariate model to analytically demonstrate the limitations of linear tests in uncovering nonlinear influence. To address this issue, a number of nonlinear tests have been suggested, *e.g.*, with generalized, autoregressive, conditional heteroskedasticity (GARCH) models (**Asimakopoulos et al., 2000**), using wavelet transforms (**Papadimitriou et al., 2003**), or heuristic, additive relationships (**Chu et al., 2005**). However, these approaches often assume a class of allowable non-linearities; thus not quite alleviating the problem of presupposed structure. This is not just an academic issue; Granger causality has been shown to be significantly sensitive to non-linear transformations (**Roberts and Nord, 1985**).

Non-parametric approaches, *e.g.* the Hiemstra-Jones (HJ) test (**Hiemstra and Jones, 1994**) on the other hand, attempt to completely dispense with presuppositions regarding causality structure. Given two series, $X_t$ and $Y_t$, the HJ test (which is a modification of the Baek-Brock test [**Baek and Brock, 1992**]) uses correlation integrals to test if the probability of similar futures for $X_t$ given similar pasts, change significantly if we condition instead on similar pasts for both $X_t$ and $Y_t$ simultaneously.

Nevertheless, in order to achieve consistent estimation of the correlation integrals, the data series are required to be ergodic, stationary, and absolutely regular $i.e.$-mixing, with an upper bound on the rate at which the $\beta$-coefficients approach zero (**Denker and Keller, 1983**). The additional assumptions beyond ergodicity and stationarity serve to guarantee that sufficiently separated fragments of the data series are nearly independent. The HJ test and its variants (**Diks and Panchenko, 2006**; **Seth and Principe, 2010**) have been quite successful in econometrics; uncovering nonlinear causal relations between money and income (**Baek and Brock, 1992**), aggregate stock returns and macroeconomic factors (**Hiemstra and Kramer, 1995**), currency future returns (**Asimakopoulos et al., 2000**) and stock price and trading volume (**Hiemstra and Jones, 1994**). Surprisingly, despite clear evidence that linear tests typically have low power in uncovering nonlinear causation (**Hiemstra and Jones, 1994**; **Asimakopoulos et al., 2000**), the application of non-parametric tests has been limited in areas beyond financial and macroeconomic interests.

## 1.1 Our Approach: Non-parametric Approach Based on Probabilistic Automata

We use a new, non-parametric Granger causality-based test for quantized processes (**Chattopadhyay, 2014**).

Going beyond binary hypothesis testing, we quantify the notion of the degree of causal influence between observed data streams, without presupposing any particular model structure. Generative models of causal influence are inferred with no a priori, imposed dynamical structure beyond ergodicity, stationarity, and a form of weak dependence. The explicit generative models may be used for prediction. The proposed inference algorithms are PAC-efficient (**Chattopadhyay, 2014**), $i.e$, we are guaranteed to find good models with high probability, and with small sample complexity.

## Modeling Framework: Probabilistic Automata and Crossed Probabilistic Automata

In this section, we briefly describe the notion of probabilistic automata, the inference problem, the notion of causal states, and the decision fusion strategy for multiple models.

### Self Models

The event catalogue's quantized data streams may be viewed as sample paths from hidden, quantized stochastic processes that drive the observed dynamics. A self-model for such a stream is a generative model that captures statistically significant symbol patterns that causally determine (in a probabilistic sense) future symbols. A good modeling framework for such self-models are probabilistic finite state automata (PFSA) (**Chattopadhyay and Lipson, 2013**; **Chattopadhyay, 2014**).

One such self model, shown in **Figure 1** (Plate A(i)), is an example of a simple PFSA with two states generated from a binary symbol sequence. Note that, for this particular example, a sequence 11 localizes or synchronizes the machine to state $q_2$, which then implies that the next symbol is a 0 with a probability of 0.1 and a 1 with a probability of 0.9. Thus, given this stream's self-model, along with the short history 11, we can predict the symbol distribution in the next time step. Notably, the states in a PFSA are not the alphabet symbols themselves, but are equivalence classes of symbol sequences (histories) that lead to statistically equivalent futures. This particular example has two states–not because of the fact that the alphabet is binary, but because there are only two distinct distributions that dictate the next symbol from any point in the dynamical evolution. The symbol value is distributed as either [0.1 0.9] or [0.3 0.7] over the binary event alphabet, and the current context or 'state' dictates which distribution is in effect chosen. These contexts are simply historical sequence equivalence classes that could have potentially lead to the present state; the specific history belonging to whichever current equivalence class actually transpired makes no difference in the future. Hence, these classes are dynamical states for the discrete stochastic evolution of the system at hand. The transition structure of the PFSA (represented in the graph as the labeled edges with

probabilities) specifies how we move between these classes as new symbols are generated, or as new data is acquired.

The inference problem here is to determine both the number of states and the transition structure given a sufficiently long, quantized data stream. In our unsupervised approach, there are no restrictions on admissible graphs for the inferred PFSA, and no a priori constraints on the number of states that may appear (**Chattopadhyay and Lipson, 2013**).

Thus, the number of states inferred by our algorithm is a direct measure of the statistical complexity of the underlying process (**Crutchfield, 1994**). An inferred single-state, self model implies that the stream is a sequence of independently generated symbols (white noise or a Bernoulli process), and is therefore uninformative.

The inference algorithm identifies the number of causal states by searching for distinct 'contexts'–sequences which, once transpired, lead to a distinct probability distribution of the next symbol. The computation proceeds as follows (**Chattopadhyay and Lipson, 2013**):

1. Let the set of all possible sequences up to a length $L$ be denoted as $S_L$.
2. Compute the probability of a future symbol (at a specified time shift) being $\sigma_0$ or $\sigma_1$ after a specific string $\omega$ from the set $S_L$ is encountered; call this distribution $\phi_\omega$ for the string $\omega$.
3. Call the set of probability distributions obtained by this method as $\Phi$.
4. Find clusters in the set $\Phi$, such that individual clusters are separated by some pre-specified distance $\epsilon > 0$. These clusters represent the causal states, as they are classes of histories (sequences) that lead to identical, immediate future.
5. Suppose string $\omega$ is in cluster $q_i$, and sequence $\omega\sigma_0$ is in cluster $q_j$; it then follows that, in the inferred PFSA, there is a transition labeled $\sigma_0$ from the corresponding state $q_i$ to state $q_j$. Carrying out this procedure for each symbol of the alphabet for each inferred cluster or state identifies the complete transition structure of the model.
6. Once the transition structure is identified, we choose an arbitrary initial state and step through the model as dictated by the input data stream. We count the number of times each edge or transition is traversed, and, by normalizing the count, we arrive at an estimate of the edge probabilities.

This completes the PFSA inference.

## Formal Definition of PFSAs and Future Prediction

Mathematically, a PFSA is a 4-tuple $G = (Q, \Sigma, \delta, \widetilde{\pi})$, where $Q$ is the set of states, $\Sigma$ is the symbol alphabet, $\delta : Q \times \Sigma \to Q$ is the transition function specifying the graph structure such that for any state $q \in Q$, $\delta(q, \sigma) \in Q$ is the end state of the transition from state $q$ via symbol $\sigma$, and $\widetilde{\pi} : \Sigma \times Q \to [0, 1]$ is the symbol probability function such that for any state $q \in Q$, $\widetilde{\pi}(q, \sigma) \in [0, 1]$ is the probability of generating the symbol $\sigma \in \Sigma$ from the state $q$, with the constraint:

$$\forall q \in Q, \sum_{\sigma \in \Sigma} \widetilde{\pi}(q, \sigma) = 1 \qquad (16)$$

We specify a time shift $\Delta$ when we infer a PFSA $G$ from a given input stream; this time shift is the delay with which the inferred model makes predictions. Specifically, each transition step in $G$ is translated to $\Delta$ steps in the quantized stream. Thus, if we know that the current state in the model is $q_0$, then the predicted symbol $\Delta$ steps for the future are $\sigma$ with probability $\widetilde{\pi}(q_0, \sigma)$.

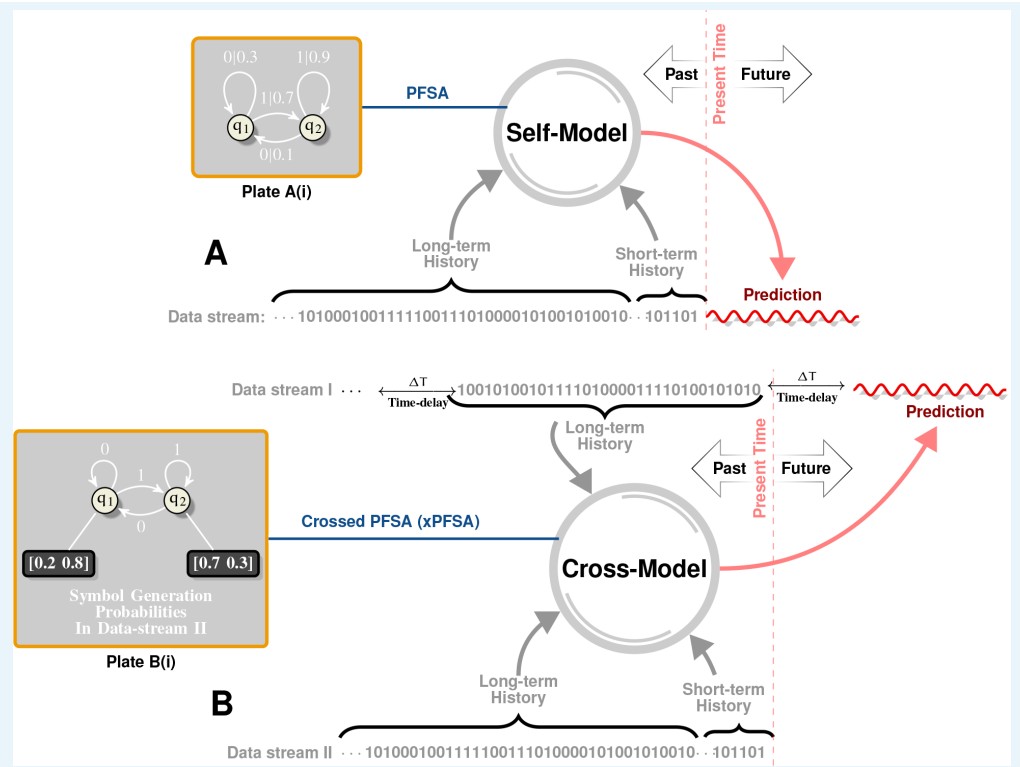

**Appendix 1—figure 1.** Intuitive Description of Self and Cross Models. *Plate A* illustrates the notion of self-models. Historical data is first represented as a symbol sequence (denoted as 'Data stream' in Plate A) using space-time discretization and magnitude quantization. For example, we may use a spatial discretization of $\pm 3°$ in both latitudes and longitudes, a temporal discretization of 1 week, and a binary magnitude quantization that maps all magnitudes below $4.0$ to symbol $0$, and all higher magnitudes to symbol $1$. This symbol stream then represents a sample path from a hidden, quantized stochastic process. A self-model is a generative model of this data stream, which captures symbol patterns that causally determine (in a probabilistic sense) future symbols. Specifically, our inferred self-model (see Plate A(i) for an example) is a probabilistic, finite state automata (PFSA). *Plate B* illustrates the notion of cross-models. Instead of inferring a model from a given stream to predict future symbols in the same stream, we now have two symbol streams (Data Stream I and Data Stream II), and the cross-model is essentially a generative model that attempts to predict symbols in one stream by reading historical data in another. Notably, as shown in Plate B(i), the cross-model is syntactically not exactly a PFSA (arcs have no probabilities in the cross-model, but each state has an output distribution). We call such models "crossed probabilistic finite state automata,' or XPFSA. Once these models are inferred, they may be used to predict the future evolution of the data streams. Thus, the self-model in Plate A may be initialized with its unique stationary distribution, after which a relatively short observed history would dictate the current distribution on the model states. This, in turn, would yield a distribution over the symbol alphabet in the next time step. For a cross-model, we would be able to obtain future symbol distribution in the second stream, given a short history in the first stream. Note that the cross-model from I→ II is not necessarily the same as the cross-model in the other direction.

DOI: https://doi.org/10.7554/eLife.30756.033

Note that, if we have two models $G_1 = (Q_1, \Sigma, \delta_1, \tilde{\pi}_1)$ and $G_2 = (Q_2, \Sigma, \delta_2, \tilde{\pi}_2)$ with time shifts $\Delta_1$ and $\Delta_2$ respectively, it follows that with each step the models make predictions at different points in time in the future.

Note that to make a prediction using a PFSA, we must know its current state. This is non-trivial in general due to the synchronization problem (*Chattopadhyay and Lipson, 2013*; *Chattopadhyay, 2014*). We find an approximation of the current state as follows:

1. Compute the he PFSA states' stationary distribution using standard Markov chain tools.
2. Treat the stationary distribution row vector $\wp$ as the initial state distribution. The rationale here is that if we assume that the model has been operating for a sufficient length of time, the current distribution must be very close to the stationary distribution in the expected sense.
3. Compute the square matrices, $\Gamma_\sigma \in \mathbb{R}^{|Q| \times |Q|}$, for each symbol $\sigma \in \Sigma$, such that the $ij^{th}$ element of $\Gamma_\sigma$, denoted as $\Gamma_\sigma|_{ij}$, which signifies the probability of going from state $q_i$ to state $q_j$ via the symbol $\sigma$.
4. Use a relatively short history $h = \sigma_1 \cdots \sigma_\ell$ of past symbols (before the current time) to update the state distribution as follows:

$$\wp^{[k]} = normalize\left(\wp^{[k-1]}\Gamma_{h_k}\right), k = 1, \cdots, \ell \qquad (17)$$

where $h_k$ is the $k^{th}$ entry of the short history $h$, and $\wp^{[k]}$ is the state distribution at step $k$. It follows from ergodicity that the updated distribution $\wp^\ell$ is an estimate of the current state distrbution.

The predicted distribution over the symbol alphabet $\Delta$ steps in the future (if the time shift for the model is $\Delta$), is then simply:

$$Pr(\sigma_{\text{predicted}} = \sigma) = \sum_{q \in Q} \wp_q^\ell \widetilde{\pi}(q, \sigma) \qquad \text{(After } \Delta \text{ time steps)}$$

where $\wp_q^\ell$ is the current probability for state $q$, i.e., the entry corresponding to state $q$ in $\wp^\ell$.

## Cross Models

Unlike a self model, which attempts to predict future symbols in the same stream, a cross model attempts to predict symbols in a target stream after reading a short history in the source stream. Plate B(i) in **Figure 1** illustrates a simple cross model with two states. Note that the model syntax is different from a PFSA; in particular, there are no probabilities on the arcs, though each state has a specified output distribution. Thus, if we see sequence 101101 in data stream $s_B$ (See **Figure 1**, Plate B), then we are in state $q_2$ in the cross-model, and hence the next symbol in the data stream $s_A$ can be predicted to be 0 with a probability of 0.7 and 1 with a probability of 0.3. We call such models crossed Probabilistic Finite State Automata (XPFSA) (**Chattopadhyay, 2014**). A cross model state or an XPFSA has a slightly different interpretation from PFSA; while PFSA states are equivalence classes of histories that lead to identical futures in the same stream, the XPFSA are equivalence classes of histories in the source stream that lead to identical futures in the target stream (the future evolution in the target stream does not matter).

XPFSA inference ˜ (**Chattopadhyay, 2014**) is similar in principle to PFSA inference. Here, we have two input data streams, the source stream $s_A$ over the source alphabet $\Sigma_A$, and the target stream $s_B$ over the source alphabet $\Sigma_B$. The broad steps are as follows:

1. Let the set of all possible sequences up to a length $L$ in source stream $s_A$ be denoted as $S_L^A$.
2. Compute the probability of a future symbol (at a specified time shift) being $\sigma \in \Sigma_B$ in the target stream $s_B$ after a specific string $\omega$ from the set $S_L^A$ is encountered in the source stream; call this distribution $\phi_\omega^{s_A, s_B}$ for the string $\omega$.
3. Call the set of probability distributions obtained in this manner as $\Phi^{s_A, s_B}$.
4. Find clusters in the set $\Phi^{s_A, s_B}$, such that the clusters are separated by some pre-specified distance $\epsilon > 0$. These clusters now represent causal states of the cross-dependence between the stream (from $s_A$ to $s_B$), because they are classes of histories (sequences) in the source stream that leads to identical, immediate futures in the target stream.
5. Suppose string $\omega$ is in a cluster corresponding to state $q_i$, and sequence $\omega\sigma_0$ is in a cluster corresponding to $q_j$; it then follows that, in the inferred XPFSA, there is a transition labeled $\sigma_0$ from the corresponding state $q_i$ to state $q_j$. Carrying out this procedure for each symbol of the source alphabet for each inferred cluster or state identifies the complete transition

structure of the cross model. Note that the cross-model transitions are only labelled with symbols from the source alphabet.

6. Once we have identified the transition structure, we choose an arbitrary initial state in the XPFSA and step through the model as dictated by the source data stream. Each time we reach a particular state in the cross-model, we note which symbol from target alphabet $\Sigma_B$ transpires in the target stream (at the specified time shift). Once we reach the end of the input source stream, we normalize the symbol count vectors corresponding to the cross-model states, and this determines the output distribution at each state. Note that each output distribution is a probability distribution over the target alphabet. This completes the XPFSA inference.

An XPFSA may be thought of as a direct generalization of a PFSA, and any PFSA can be represented as an XPFSA. To illustrate this, note that predicting future symbols in the same stream can be thought of as predicting symbols is a second, but identical, copy of the first stream.

If all histories in a source stream are equivalent in this sense, then we have a single state XPFSA, which implies that the source cannot provide any new information on what is going to happen in the target stream based on its own history. Hence, that source lacks any causal influence on the target. Thus, as before, the complexity of the inferred models is directly related to the statistical complexity (**Crutchfield, 1994**) of the learnable, dynamical relationships underlying data streams themselves.

XPFSAs are asymmetric in general; the model for how a source influences a target does not need to be identical when the roles played by the streams are reversed. However, two streams are statistically independent if and only if the XPFSAs in both directions are single state machines (**Chattopadhyay, 2014**).

This leads us to the notion of the causality coefficient, $\gamma_B^A$, from data stream $s_A$ to data stream $s_B$. True to Granger's notion of statistical causality, $\gamma_B^A$ rigorously quantifies the amount of *additional* information we can obtain about the immediate future in stream $s_B$ through observing stream $s_A$.

Explicitly, the coefficient is defined as the ratio of the expected change in the entropy of the next-symbol distribution in stream $s_B$ conditioned over observations in the stream $s_A$ to the entropy of the next-symbol distribution in stream $s_B$, conditioned on the fact that no observations are made on stream $s_A$. We show that causality coefficient $\gamma_B^A$ takes values on the closed unit interval and that higher values indicate a stronger predictability of $s_B$ from $s_A$, *i.e.*, therefore, a higher degree of causal influence. We have $\gamma = 0$ if and only if the inferred machine has a single state, and streams $s_A, s_B$ are statistically independent if and only if $\gamma_B^A = \gamma_A^B = 0$.

Thus, the interpretation of the causality coefficient which is central to our development is as follows:

$\gamma_B^A = 0.3$ *(say) from $s_A$ to $s_B$ means that we can acquire, on average, $0.3$ bits of additional information about $s_B$ from each bit read from $s_A$, over what is already available from the past history of $s_B$.*

This is very different from computing correlations, and it assumes no model structure a priori for the hidden dynamics driving the data streams.

Importantly, it turns out that inferring self models is simply a special case of cross model inference, in which the target stream is simply a time-shifted version of the first stream.

## Formal Definition of XPFSAs and Future Prediction

Formally, an XPFSA is also a 4-tuple $G_{A \to B} = (Q, \Sigma_A, \Delta, \widetilde{\pi})$, where $Q$ is a set of states, $\Sigma_A$ is the input alphabet. The transition function $\delta : Q \times \Sigma_A \to Q$ is defined as before (in the case of the PFSA), but the symbol probability function is defined as $\widetilde{\pi} : Q \times \Sigma_B \to [0, 1]$, with

$$\forall q \in Q, \sum_{\sigma \in \Sigma_B} \widetilde{\pi}(q, \sigma) = 1 \tag{18}$$

where $\Sigma_B$ is the target alphabet, which is distinct from $\Sigma_A$ in general. Note that the symbol distribution specified by $\widetilde{\pi}$ is over the target alphabet $\Sigma_B$.

As before, there is a time shift $\Delta$ associated with inferred XPFSA, such that each symbol transition in the model maps to $\Delta$ steps in the target stream. If we know the current state of the XPFSA to be $q_0$, and we observe symbol $\sigma \in \Sigma_A$ in the source stream, then the predicted symbol in the target stream ($\Delta$ steps in the future) is $\sigma' \in \Sigma_B$, with a probability of $\widetilde{\pi}(\delta(q_0, \sigma), \sigma')$.

As before, determining the current state in the XPFSA is non-trivial. We estimate the current state distribution in a manner similar to that used for self models:

1. Note that the XPFSA graph does not have probabilities on it edges. However, the corresponding source stream's self model does have symbol probabilities on its own edges, and it models the source stream. The problem is that the graph for the self model, and that of the cross model, might not be identical. We solve this problem using projective composition of probabilistic automata (**Chattopadhyay and Ray, 2008**). This operation takes the self model for the source stream $G_A$, and projects it on the inferred XPFSA graph $G_{A \to B}$, and the result is a PFSA $G_A 3.0pt\overrightarrow{-3.0pt\otimes}G_{A \to B}$ with the same structure as that of $G_{A \to B}$.

2. Once we obtain $G_A 3.0pt\overrightarrow{-3.0pt\otimes}G_{A \to B}$, we can estimate its current state as described in the case for self models (using observed symbols in the source stream).

3. Let the current state distribution be $\overline{\wp}^\ell$. Then, the predicted future symbol (at time shift $\Delta$) in the target stream is $\sigma' \in \Sigma_B$ with a probability of:

$$Pr(\sigma_{\text{predicted}} = \sigma') = \sum_{q \in Q} \overline{\wp}_q^\ell \widetilde{\pi}(q, \sigma')$$

where $G_A 3.0pt\overrightarrow{-3.0pt\otimes}G_{A \to B}$ (note we are using the $\widetilde{\pi}$ from the XPFSA, and not from the projected PFSA).

## Computation of the Coefficient of Causality

Let $\mathcal{H}_A, \mathcal{H}_B$ be stationary ergodic processes over finite alphabets $\Sigma_A, \Sigma_B$ respectively. Then, the causal dependence coefficient of $\mathcal{H}_B$ on $\mathcal{H}_A$, denoted as $\gamma_B^A$, is formally defined as the ratio of the expected change in the entropy of the next symbol distribution in $\mathcal{H}_B$. This is due to observations in $\mathcal{H}_A$, which show the entropy of the next symbol distribution in $\mathcal{H}_B$. In the absence of data in $\mathcal{H}_A$, i.e., we have (**Chattopadhyay, 2014**):

$$\gamma_B^A = 1 - \frac{\mathbf{E}\left(\boldsymbol{h}\left(\phi_x^{\mathcal{H}_A, \mathcal{H}_B}\right)\right)}{\boldsymbol{h}\left(\phi_\lambda^{\mathcal{H}_A, \mathcal{H}_B}\right)} \tag{19}$$

where the entropy $\boldsymbol{h}(u)$ of discrete probability distribution $u$ is given by $\sum_i u_i \log_2 u_i$, $\lambda$ is the empty string, and $\mathbf{E}(\cdot)$ is the expectation operator over all possible sequences in the source stream. Once we infer the cross model, computing $\gamma_B^A$ is straightforward.

## Choice of Quantization Schemes

The weekly time series of county-specific records for the number of reported influenza cases is quantized to a binary stream. If $y_k$ is the number of reported cases for a given fixed county at the week index $k$, then the quantized binary stream $\overline{y}_k$ is obtained as follows:

$$\overline{y}_k = \begin{cases} 1 & if\, y_k > y_{k-1} \\ 0 & otherwise \end{cases} \tag{20}$$

