## [Decision Letter]

[Editors’ note: a previous version of this study was rejected after peer review, but the authors submitted for reconsideration. The first decision letter after peer review is shown below.]

Thank you for submitting your work entitled "Conjunctions of factors triggering waves of seasonal influenza" for consideration by *eLife*. Your article has been evaluated by Prabhat Jha (Senior Editor) and four reviewers, one of whom, Mark Jit, is a member of our Board of Reviewing Editors.

All the reviewers and I agreed that this is an interesting, innovative and sophisticated attempt to understand the drivers of seasonal influenza spread throughout continental United States. However, we had wide-ranging concerns about multiple areas of the paper.

Based on these discussions and the individual reviews below, we have decided to reject the paper as we think that substantial time and work will be needed to address all the concerns. However, if you would like to substantially revise the paper in a way that takes into account the concerns, we would be interested to consider a revised version of the manuscript as a new submission. Please note that we cannot guarantee review of the revised version until we have seen it.

The full reviews are included, but the most important concerns that would need to be addressed in a new submission are the following:

1) Presentation: While *eLife* does not have strict format requirements, this paper in particular would benefit from a more classic presentation (e.g. Introduction, Materials and methods, Results, Discussion) that provides a fuller description of the background, purpose and how the paper interacts with and adds to the existing (vast) literature on this topic.

2) The data sources and the way they are used need a more detailed description and justification. The definition of ILI was challenged by one of the reviewers and needs to be justified.

3) While the environmental and demographic factors driving seasonality are extensively examined, there is a need also to take into account viral and host immunity factors such as antigenic drift, seasonal vaccine uptake and antigenic match, host susceptibility and between-host transmissibility (with the latter heavily influenced by the school calendar).

4) The conclusions (e.g. the "Southern hypothesis" and bodies of water being sources) are interesting but need stronger explanation and justification. The different approaches taken did not all give exactly the same results. Claims of causality in epidemiology require an extremely high level of evidence so in this case more moderate language may be appropriate (e.g. "the association suggests.[…]").

5) It would have been useful to make some attempt to validate the methods used, such as by holding back a validation dataset.

*Reviewer #1:*

This is an innovative and well-illustrated attempt to understand the drivers of seasonal influenza spread each season throughout continental United States. The techniques used are interesting and the array of datasets that the investigators have compiled are impressive. Nevertheless, I do think that their conclusions place far too much certainty than can be justified from statistical analysis of observational data.

1) The claims about causality in the paper should be tempered. While Approach 1 may historically have been called Granger causal inference, there is a limit to the extent to which causality can be claimed in the absence of (and sometimes even with) experimental data. At best, the manuscript should claim that the associations in the data are consistent with their hypotheses.

2) The concept of streamlines is an interesting one but it's not clear what the underlying mechanisms behind them are. They seem to imply that there is a continuous line linking infection in a particular location with a source in another, usually distant, location. Is it not more consistent with our understanding of infection spread to assume that infection spread is a diffusion process, even though environmental and host factors may enhance diffusion in particular directions?

3) All three approaches used are essentially data mining exercises, followed by post-hoc generation of hypotheses to explain the associations found. This does not mean they are invalid – plenty of science works in this way, and some care has been taken to avoid type I errors. Nevertheless, the possibility of bias, confounding, alternative explanations and simply random error should not be discounted. This is particularly the case because there is strong geographic clustering of many of the variables identified as key drivers (e.g. densely populated areas, coastal areas, places with airports, places at the geographical and climactic extremes of the USA tend to be the same places). Hence the language used should be substantially tempered to be more tentative. While the authors have used *E. coli* and HIV spread as controls to ensure that these display different patterns of spatial spread, the problem is that the seasonality of these pathogens is completely different or (in the case of HIV) largely absent. Looking at a non-respiratory but still highly seasonal pathogen e.g. norovirus might be interesting.

4) One shortcoming of the data analysis is that although a large number of variables related to environmental and demographic factors are considered, there is little consideration of viral and host characteristics which are key to driving influenza spread. For instance, over the nine years the model is run, the United States has experienced waves of different influenza types, a pandemic (which led to a shift in the influenza A/H1 subtype), different levels of vaccine coverage with varying levels of match to the prevailing strains, as well as changes in immune profiles of individuals over the population. There is no attempt to understand these drivers and capture their spatiotemporal heterogeneity even though there are datasets (e.g. vaccine coverage, serology, virological testing) that could be used to inform this.

Note about presentation:

The presentation of the paper is rather confusing. There is hardly any introductory material; the first section is essential a summary of methods and results. This is followed by a more detailed description of the methods and finally a discussion. While *eLife* does not have strict rules about manuscript formatting, the paper should at least follow the general conventions of medical/life sciences journals i.e. the Introduction, Materials and methods, Results, Discussion division which will make finding information easier for most readers. A fuller Introduction section setting the background is also needed e.g. why this is an important problem from a clinical/public health point of view, what other attempts have been made to address it and what their findings and shortcomings have been?

*Reviewer #2:*

This is an interesting Big Data study that aims to uncover the causal factors driving the onset of epidemic influenza at the county level in the United States. Authors rely on multiple datasets including medical claim records, demographic characteristics, human movement inferred from Twitter data, and climate data. Authors employed three different methodological approaches and found a complex relationship with influenza epidemic onset. Although the study is of significant interest, I do have several concerns indicated below.

1) The definition of epidemic onset is loosely defined and is unclear whether the same definition was employed across approaches. They discuss a "trigger period" of several weeks to exponential expansion. Is exponential expansion a necessary condition? This is the response variable in their study and little is discussed about its variation across counties and studied seasons.

2) Patterns from different approaches were not necessarily in agreement as authors discussed. In particular, authors did not attempt to reconstruct season-to-season variation in influenza susceptibility. A mechanistic approach that accounts for temporal variation in influenza susceptibility is missing (perhaps departing from previous work by Shaman and his colleagues). Authors did not attempt to capture variation in reproduction numbers, vaccination rates, vaccine efficacy, and likely others across seasons.

3) Authors found strong support for a southern origin of influenza in the US. It would have been interesting to predict flu onset across the vast range of southern counties.

4) A validation phase was not attempted. For instance, authors could use the first few years of data for model calibration and the remainder of the data for forecasting purposes. It would have been interesting to quantify the predictive power of their approaches.

*Reviewer #3:*

This is a methodologically sophisticated paper which cleverly uses medical claims and other large datasets, and three different computational approaches, to explore whether three factors (weather, X and Y) might explain the timing of seasonal influenza in the US. They find that influenza epidemics tend to start in Southern states.

My impression is that the paper is highly sophisticated in terms of analytics, but that the data sources that are used to describe influenza and the possible drivers (such as weather, mobility) are not always well developed and ill described (Truwen) or even possibly inadequate (Twitter). The findings of short-distance traveling being important driver corroborates findings by Gog et al., another pioneer study using medical claims data to study influenza dissemination, which is nice.

Trusting other reviewers will take on an evaluation of the analytic approaches, I will focus instead on the validity of the data (outcomes, explanatory) side of the matter.

It is my impression that the paper needs substantial revision in order to be ready for publication. For example, data sources and approaches must be far better described and more sensitivity analyses done. The lack of consideration of genetic drift variants and school schedule patterns, along with the likely consequences of regional differences in Truwen data coverage and ICD coding, if any, should be argued.

On the topic of drivers not considered:…

1) One obvious driver of seasonal influenza timing is not mentioned in this paper: the degree of genetic drift of the influenza virus relative to those that circulated in earlier years. That influenza occurs early in big "drift" years is evident from the unusually early onset of the 2003 Fujian H3N2 epidemic, and also of most pandemics in history including the 2009 pandemic. This is not discussed and that is a caveat that should be discussed. If such data are too much trouble, then at the very least consideration of which influenza subtype is dominant in each season would be useful; for example influenza B tends to be epidemic later than influenza A.

a) Could authors do a sensitivity sub-analysis of seasons of major drift variants (2003 Fujian and 2009 pandemic flu) to see if their identified drivers also are found for those?

2) Another driver that is not discussed is the school session patterns across the US during the autumn and winter. As it was picked up as a major driver of pandemic spread by Gog et al., 2014, why would it not also be important for seasonal influenza? This should be listed as a shortcoming of the analysis, or authors could explain in the text why they did not think this was important.

On the topic of validity and validation of the observational data underlying the analyses:

3) I am concerned about the robustness of the analysis due to possible low counts of ILI as the authors defined it narrowly as ICD codes 487-488 – which could mean much noise in ILI time series at the local level. The authors may benefit from a very similar study of influenza dissemination in the 2009 pandemic in the US, which was based on IMS CMS1500 claims data (Gog et al., 2014); the underlying paper preparing the medical claims data and validating against CDC influenza data (Viboud et al., Plos One 2013). Viboud et al., established that adding 079.9 (ICD9) to the case definition dramatically (by 90%) increased the sensitivity of the case definition, without much cost on the specificity.

a) Yet in the "Flammable media" (Discussion section) the authors discuss how 20% of Southeners have ILI each season. This seems way too high, as most people do not seek care for ILI and 20% (and what would the rate have been if the ILI definition was not so narrow?) is already higher than population-based sero-epidemiological assessments of attack rates and it is well known that most of people with ILI do not seek medical care and that at least half of infections are asymptomatic. So 20% ILI population prevalence in Truwen data is unbelievable.

i) Furthermore, it is difficult to believe that Southern US is better connected than, for example, dense US Northern cities like New York with large scale public transportation where influenza may be easily spread. See my later point about ICD coding preferences –… another worry that seasonal initial levels in southern US is a data artefact somehow, due to for example higher Truwen coverage and different ICD coding practices.

ii) Moreover, the authors comment at some point that the ILI prevalence is higher in the mid-west. However, local differences in ICD coding preferences by doctors are likely a better explanation, and so these differences could just be artefacts. For that reason it would be better to not include prevalence in the modeling, but rather base the analysis solely on the initial epidemic phase and/or the peak timing.

4) The Truwen data source and extraction of the ILI and other time series should be described in much more detail. Is it CMS^-1^500 data? Were influenza codes sought among all codes listed or only the first-listed? Etc. Also, it would be helpful to show graphically the time series data, to allow the reader to evaluate data stability and volume – over time and between locales. Or else, add some summary table that summarizes such observational findings. The Truwen claims database is likely a convenience sample, so it is likely that the coverage increases dramatically over time since 2003, and also that there are substantial differences in coverage by US geographical areas depending on Truwen levels of foot holds. Did the authors see such intrinsic variability in the data (for example, consider another code in summer months? How would such variability in coverage influence the analyses presented here – or is it somehow taken into account?

a) The authors offer a comparison with HIV patterns – but is that appropriate as HIV is a chronic disease – how exactly was this done? Did authors extract only ICD codes for new onset HIV or using longitudinal patient-level data (nothing in the Materials and methods section about this)? A better comparison would perhaps be something else epidemic but with another mode of transmission such as rotavirus or norovirus diarrhea epidemics?

5) Validity of Twitter data to describe mobility? The authors conclude that short-distance land travel is an important determinant of the spread of influenza, using Twitter data to determine mobility. But is this defensible, as only older kids and adults have Twitter accounts – yet we know that children are important drivers of influenza (see fx Gog et al., two major factors determining spread of influenza was school schedules and short distance traveling). Please comment on this possibility that the Twitter data are not determinants of children's movements.

6) On the "Initiation spark" (DDiscussion section). The authors conclude that cold weather and low humidity are important drivers. Put together with their finding that most epidemics originate in the Southern United states, one must ask: How could cold weather/low humidity be a predictor of influenza *and* influenza starts in the South – when Southern states have warmer climate and higher humidity in early winter than Northern states? Would be great if the authors would discuss this possible paradox.

*Reviewer #4:*

This manuscript presents an analysis of a large dataset of ILI from insurance records to identify spatial patterns in influenza outbreak initiation and spread. The authors present 3 classes of analysis to identify variables correlated with outbreak initiation. Overall, I found this work very interesting but in dire need of restructuring to clarify the main questions and methods. At present, this reads as an explanation of several very innovative approaches to mining these large datasets for patterns, but doesn't lay out a clear goal or hypothesis. Similarly, the Discussion lists an array of patterns that were discovered, but fails to identify a single (or even just a few) key insight that was gained through this approach.

I'm usually not a stickler for presentation of papers in the classic Introduction/Materials and methods/Results/Discussion format – but given that the authors are presenting several novel methods here, I think the manuscript would really benefit from being more classic. At present, some aspects of the methods are described in the "Approach #" sections, but others are presented in the "Materials and methods". I think it would be much clearer to group all the methods together so that the results could themselves be presented in a more concise format that would highlight the key insights.

The figures are excellent, but they are referenced out of order in the text; e.g. Figure 2 gives results from Approach 2 which is presented after the results from Approach 1, which are in Figure 4 and 5.

Overall, I would like to see a more concise statement of the major innovation from this approach and this analysis. At present, this is a list of variables that were identified as significant (albeit using some really interesting methods and excellent data), many of which have already been identified as predictive of flu in the past. What would be really interesting, if possible, would be to illustrate whether this method was in some way more predictive of flu dynamics than conventional methods. For example, could these analyses be carried out on the first 8 years of ILI data and the resulting model used to predict the 9th; if so, does the addition of Approach 1/3 do better than the standard Poisson regression?

[Editors’ note: what now follows is the decision letter after the authors submitted for further consideration.]

Thank you for submitting your article "Conjunction of Factors Triggering Waves of Seasonal Influenza" for consideration by *eLife*. Your resubmission has been evaluated by Prabhat Jha (Senior Editor) and three reviewers, one of whom, Mark Jit (Reviewer #1), is a member of our Board of Reviewing Editors The following individual involved in review of your submission has agreed to reveal their identity: Elizabeth C Lee (Reviewer #3).

The reviewers have discussed the reviews with one another. We consider that you have undertaken a major effort in putting their paper in an appropriate format and addressing the reviewers' comments and are, in general, satisfied with the revisions made. However, we still have a few remaining concerns that should be addressed by the authors to make their paper suitable for potential publication in *eLife*.

The Reviewing Editor has drafted this decision to help you prepare a revised submission. If you are able to address at least all the essential points to our satisfaction, we should be able to make a quick decision about the manuscript.

Essential revisions:

1) The literature review is still not adequate. Reviewing the literature is not simply a matter of having a string of references. Ideally a couple of paragraphs would be added to the Introduction or Discussion discussing the history of related statistical models for understanding the timing, spread and burden of seasonal influenza, and how this paper builds on the existing literature.

Also several of the previous reviewers found it difficult to determine the actual aim of the data-driven exercise. While we understand and are sympathetic to the point that this is not a traditional hypothesis-driven study, it still needs an aim if not a hypothesis. Could the last line of the first paragraph be reworked into what the aim of this was? (e.g. is it to determine the most important factors that are associated with initiating influenza transmission in the continental USA at the beginning of each season?)

2) You state that "our more precisely defined ILI data defines better against the CDC dataset" – could details (including quantitative measures of better definition) be given? Both narrower and broader definitions of ILI are imperfect proxies of influenza activity, although in different ways, and it is not a given that a more specific measure is necessarily less biased.

3) The incorporation of antigenic drift, vaccine coverage and school data are important and innovative additions to the paper. However, one missing factor is host immunity not due to vaccination i.e. if there is a strong influenza season in one year with no substantial antigenic drift after that then the population immune to influenza is likely to be greater the following year. Ideally, this could be captured mechanistically by incorporating SIR-type epidemiological models, or by more statistical approaches, ideally informed by measures of prior immunity such as serology. At the very least, the implications of this factor need to be adequately discussed.

4) We still think (as reviewer 3 of the previous round of reviews pointed out) that a 20% influenza infection rate is far too high, even if it incorporates asymptomatic/subclinical infection. This is much greater than the change in seropositivity over most influenza seasons other than pandemic years.

5) Given the spatiotemporal nature of the findings, the authors should include more information about the Truven database. For instance, are the authors able to provide additional details about the spatial distribution of reporting rates (e.g., number of physicians contributing claims in a given county) within the Truven database? While insurance claims do provide high resolution spatial information, the use of raw counts for influenza-related diagnosis codes could be problematic if there is great spatial or temporal variation in reporting rates.

6) While the methods and results are clearly described, the organization of the paper can be improved to facilitate easier reading. We found ourselves flipping back and forth a lot between the text because details that appeared to be missing earlier on would then appear later in different parts of the text. Figures were also cited out of order across the methods descriptions, and in some cases we don't believe all of the figures were discussed. As a few examples, the methods used to generate Figure 3 were not discussed until the Results (subsection “Approach 1: Non-parametric Granger analysis”, fourth and fifth paragraphs) but Figure 3 results are first discussed in the Materials and methods (subsection “Analysis of quantized clinical data (Approach 1)”, first paragraph).

For more concrete advice, we would remove references to results-related figures from the Materials and methods section so that readers examine the figures only after all of the terms and analyses have been explained, and, having a cleaner separation between Materials and methods and Results will improve readability. There were many times where the authors alluded to some small part of the results in the Materials and methods before the description of the analysis had been completed. Conversely, the Results section had relatively detailed methods descriptions, which seemed redundant since the Results followed the Materials and methods. The authors may also consider a greater overhaul of the manuscript structure that groups the primary results in the text and figures (as opposed to organizing the manuscript by methodological approach) and moves many of the supporting or diagnostic figures to the supplement.

7) Beyond improving the organization of the paper, we think the reporting of results for Approach 1 could be better explained and supported in the text. For instance, it was not clear how the claims in the sixth and eighth paragraphs of the subsection “Approach 1: Non-parametric Granger analysis” were demonstrated with the cited figures.

8) Finally, given the emphasis on the novelty of comparing results across three methodological approaches, the authors should discuss the generalizability of their findings to influenza epidemiology beyond the United States and the generalizability of their framework to other contexts with high spatiotemporal resolution data.

[Editors' note: further revisions were requested prior to acceptance, as described below.]

Thank you for submitting your article "Conjunction of Factors Triggering Waves of Seasonal Influenza" for consideration by *eLife*. Your article has been reviewed by one of the original peer reviewers, and the evaluation has been overseen by a Reviewing Editor and Prabhat Jha as the Senior Editor.

The reviewers have discussed the reviews with one another and the Reviewing Editor has drafted this decision to help you prepare a revised submission. We appreciate your revisions to the organization of the text and interpretation of the results, and think that many of the changes improve the readability and flow of the paper and clarify the goals of the analysis. Nevertheless, we still have remaining concerns about some parts of the analyses and about the sparsity of information for others. Hence we think that the article is still not ready to be published in its current state.

We would like to avoid multiple further rounds of revision, so we can give you one more opportunity to address the remaining issues before a final decision about the manuscript is made.

The remaining issues with the manuscript can be divided into two categories: (i) substantial scientific issues that need to be addressed before the manuscript is published, (ii) recommendations about the description of the work that we strongly encourage you to take but which are optional.

Essential revisions about scientific content:i) (Discussion, second paragraph) It is still difficult to interpret the results related to short-range and long-range travel. We were unable to find the regression results for the full set of data weeks in the supplement. Also, there is not enough detail in the analysis description to see how Figure 5 Plates C-F demonstrate that local connections are more explanatory than airport connections.

ii) The highlighted areas (greater incidence) in the first flu season snapshot for Figure 3 each should match relatively well to the maps in Figure 7B (or at least some unspecified subset of "high incidence treatment counties"). While the county-matching results to point to initiation at coasts, 2006-2007 in Figure 3 seems to initiate from the West Coast and 2010 in Figure 7 has a scattered pattern across the middle of the US. Please check this or justify the differences.

iii) Also related to Figure 7, was there also a geographic constraint on the choice of the matched control counties? It is surprising that everything is so geographically clustered.

Recommendations about presentation of content:

i) It would be useful to provide a quantitative example of how to interpret one of the results in Figure 2.

ii) (Subsection “The weather/humidity paradox”) This humidity paradox has come up quite a bit recently and it may be cleaner to have a fuller treatment of this in the Discussion.

iii) The subsection on Variable and model selection at the end of the Discussion seems out of place. Was this meant to be moved to the Materials and methods?

iv) Please point out the Discussion section describing the generalizability of the findings (as mentioned in the response).

v) The Conclusion section does much more of the storytelling that would be expected from a Discussion. It may be more compelling to kick off the Discussion with the analogy describing the epidemiological findings, and then elaborate on the strengths of using multiple methods to corroborate statistical findings.

vi) You may consider condensing the figures in the main text to those that present primary findings (removing some plates and potentially reducing the total number of figures). The figures are all visually appealing, but every single one is densely packed with primary findings, descriptive information, data processing details, and diagnostics/sensitivity. Some of the main takeaways may get lost.

vii) Figure 2 might be more aptly named "Putative determinants of seasonal influenza onset[…]"

viii) Figure 4, Plate I: Please add an explanation of the red circles to the legend title or caption.

ix) Please provide some explanation of how you developed the 126 models examined in Approach 2. Based on the models listed in Table 5 and in the supplement, it seems that there is substantial variation in the number of predictors included in the model and all of the models appear to include multiple autoregressive terms for the same environmental and travel predictors. What is the motivation for this approach? In addition, it would be good to better understand the random effects in the model (subsection “Approach 2: Poisson Regression on Putative Factors”, first paragraph) and the motivation for this model structure. Does this term include a spatial smoothing component?

---

## [Author Response]

[Editors’ note: the author responses to the first round of peer review follow.]

The full reviews are included, but the most important concerns that would need to be addressed in a new submission are the following:1) Presentation: While eLife does not have strict format requirements, this paper in particular would benefit from a more classic presentation (e.g. Introduction, Materials and methods, Results, Discussion) that provides a fuller description of the background, purpose and how the paper interacts with and adds to the existing (vast) literature on this topic.

As requested, we introduced classic paper structure.

We also increased coverage of literature, for example see new references Alvarado-Facundo et al., 2009; Andreasen, 2003; Araz et al., 2012; Bedford et al., 2015; Boni, 2008; Boni et al., 2004; Chlanda et al., 2015; Rizzo et al., 2008; Shaman and Kohn, 2009; Sham et al., 2010; Tamerius et al., 2013; Treanor, 2004; Vynnycky and Edmunds, 2008; Yasuda et al., 2008; Yasuda, Yoshizawa and Suzuki, 2005.

2) The data sources and the way they are used need a more detailed description and justification. The definition of ILI was challenged by one of the reviewers and needs to be justified.

We expanded descriptions and added a special comment on ILI definition, see subsection “Clinical data source”.

3) While the environmental and demographic factors driving seasonality are extensively examined, there is a need also to take into account viral and host immunity factors such as antigenic drift, seasonal vaccine uptake and antigenic match, host susceptibility and between-host transmissibility (with the latter heavily influenced by the school calendar).

As requested, we included into analysis, as requested, viral antigen diversity(subsection “Vaccination Coverage”), human immunization rates(subsections “Data on vaccination coverage” and “Estimating vaccination coverage”), and schools schedule(subsection “Estimating the effect of return-to-school days”).

4) The conclusions (e.g. the "Southern hypothesis" and bodies of water being sources) are interesting but need stronger explanation and justification. The different approaches taken did not all give exactly the same results. Claims of causality in epidemiology require an extremely high level of evidence so in this case more moderate language may be appropriate (e.g. "the association suggests.…").

We reworded our clams to sound more moderately (no causality statement or “putative causality” at most) throughout the text.

5) It would have been useful to make some attempt to validate the methods used, such as by holding back a validation dataset.

As suggested, we added to our analysis experiments with estimating parameters with test data and then testing the model with a held-out dataset. The predictive power of the best models appears reasonably high (subsection “Validation of Predictive Capability”).

Reviewer #1:[…] 1) The claims about causality in the paper should be tempered. While Approach 1 may historically have been called Granger causal inference, there is a limit to the extent to which causality can be claimed in the absence of (and sometimes even with) experimental data. At best, the manuscript should claim that the associations in the data are consistent with their hypotheses.

Agreed, carefully changed throughout.

2) The concept of streamlines is an interesting one but it's not clear what the underlying mechanisms behind them are. They seem to imply that there is a continuous line linking infection in a particular location with a source in another, usually distant, location. Is it not more consistent with our understanding of infection spread to assume that infection spread is a diffusion process, even though environmental and host factors may enhance diffusion in particular directions?

We absolutely agree, the streamlines show the patterns of dominant flow that are realized via diffusion-like process.

3) All three approaches used are essentially data mining exercises, followed by post-hoc generation of hypotheses to explain the associations found. This does not mean they are invalid – plenty of science works in this way, and some care has been taken to avoid type I errors. Nevertheless, the possibility of bias, confounding, alternative explanations and simply random error should not be discounted. This is particularly the case because there is strong geographic clustering of many of the variables identified as key drivers (e.g. densely populated areas, coastal areas, places with airports, places at the geographical and climactic extremes of the USA tend to be the same places). Hence the language used should be substantially tempered to be more tentative.

Agreed, changed.

While the authors have used E. coli and HIV spread as controls to ensure that these display different patterns of spatial spread, the problem is that the seasonality of these pathogens is completely different or (in the case of HIV) largely absent. Looking at a non-respiratory but still highly seasonal pathogen e.g. norovirus might be interesting.4) One shortcoming of the data analysis is that although a large number of variables related to environmental and demographic factors are considered, there is little consideration of viral and host characteristics which are key to driving influenza spread. For instance, over the nine years the model is run, the United States has experienced waves of different influenza types, a pandemic (which led to a shift in the influenza A/H1 subtype), different levels of vaccine coverage with varying levels of match to the prevailing strains, as well as changes in immune profiles of individuals over the population. There is no attempt to understand these drivers and capture their spatiotemporal heterogeneity even though there are datasets (e.g. vaccine coverage, serology, virological testing) that could be used to inform this.

We now repeated our analysis, incorporating additional data types, as suggested. (Please see our reply to point #3 raised by the Editor.)

Note about presentation:The presentation of the paper is rather confusing. There is hardly any introductory material; the first section is essential a summary of methods and results. This is followed by a more detailed description of the methods and finally a discussion. While eLife does not have strict rules about manuscript formatting, the paper should at least follow the general conventions of medical/life sciences journals i.e. the Introduction, Materials and methods, Results, Discussion division which will make finding information easier for most readers. A fuller Introduction section setting the background is also needed e.g. why this is an important problem from a clinical/public health point of view, what other attempts have been made to address it and what their findings and shortcomings have been?

Agreed, the required changes were made, now we have the standard section titles (please see our reply to point #1 raised by the Editor).

Reviewer #2:This is an interesting Big Data study that aims to uncover the causal factors driving the onset of epidemic influenza at the county level in the United States. Authors rely on multiple datasets including medical claim records, demographic characteristics, human movement inferred from Twitter data, and climate data. Authors employed three different methodological approaches and found a complex relationship with influenza epidemic onset. Although the study is of significant interest, I do have several concerns indicated below.1) The definition of epidemic onset is loosely defined and is unclear whether the same definition was employed across approaches. They discuss a "trigger period" of several weeks to exponential expansion. Is exponential expansion a necessary condition? This is the response variable in their study and little is discussed about its variation across counties and studied seasons.

We made it clear that the same definition was used across all approaches and clarified the definition of onset, see subsection “Analysis of quantized clinical data (Approach 1)”, first paragraph.

2) Patterns from different approaches were not necessarily in agreement as authors discussed. In particular, authors did not attempt to reconstruct season-to-season variation in influenza susceptibility. A mechanistic approach that accounts for temporal variation in influenza susceptibility is missing (perhaps departing from previous work by Shaman and his colleagues). Authors did not attempt to capture variation in reproduction numbers, vaccination rates, vaccine efficacy, and likely others across seasons.

In the revised version, we have additional predictors (such as immunization and viral antigenic drift) that bring us closer to the mechanistic interpretation, please see our response to the Editor’s point #3.

3) Authors found strong support for a southern origin of influenza in the US. It would have been interesting to predict flu onset across the vast range of southern counties.

Following the reviewer’s suggestion, we did run a cross-validation analysis, estimating and testing models on disjoint subsets of data (see subsection “Validation of Predictive Capability”).

4) A validation phase was not attempted. For instance, authors could use the first few years of data for model calibration and the remainder of the data for forecasting purposes. It would have been interesting to quantify the predictive power of their approaches.

Agreed. In the revised version, we attempted splitting data into training and testing (see subsection “Validation of Predictive Capability”).

Reviewer #3:This is a methodologically sophisticated paper which cleverly uses medical claims and other large datasets, and three different computational approaches, to explore whether three factors (weather, X and Y) might explain the timing of seasonal influenza in the US. They find that influenza epidemics tend to start in Southern states.My impression is that the paper is highly sophisticated in terms of analytics, but that the data sources that are used to describe influenza and the possible drivers (such as weather, mobility) are not always well developed and ill described (Truwen) or even possibly inadequate (Twitter). The findings of short-distance traveling being important driver corroborates findings by Gog et al., another pioneer study using medical claims data to study influenza dissemination, which is nice.Trusting other reviewers will take on an evaluation of the analytic approaches, I will focus instead on the validity of the data (outcomes, explanatory) side of the matter.It is my impression that the paper needs substantial revision in order to be ready for publication. For example, data sources and approaches must be far better described and more sensitivity analyses done. The lack of consideration of genetic drift variants and school schedule patterns, along with the likely consequences of regional differences in Truwen data coverage and ICD coding, if any, should be argued.On the topic of drivers not considered:1) One obvious driver of seasonal influenza timing is not mentioned in this paper: the degree of genetic drift of the influenza virus relative to those that circulated in earlier years. That influenza occurs early in big "drift" years is evident from the unusually early onset of the 2003 Fujian H3N2 epidemic, and also of most pandemics in history including the 2009 pandemic. This is not discussed and that is a caveat that should be discussed. If such data are too much trouble, then at the very least consideration of which influenza subtype is dominant in each season would be useful; for example influenza B tends to be epidemic later than influenza A.a) Could authors do a sensitivity sub-analysis of seasons of major drift variants (2003 Fujian and 2009 pandemic flu) to see if their identified drivers also are found for those?

We now included genetic drift into our modeling as suggested by the reviewer, please see our response to the Editor’s point #3.

2) Another driver that is not discussed is the school session patterns across the US during the autumn and winter. As it was picked up as a major driver of pandemic spread by Gog et al., 2014, why would it not also be important for seasonal influenza? This should be listed as a shortcoming of the analysis, or authors could explain in the text why they did not think this was important.

We included the school seasons in our analysis, please see our response to the Editor’s point #3.

On the topic of validity and validation of the observational data underlying the analyses:3) I am concerned about the robustness of the analysis due to possible low counts of ILI as the authors defined it narrowly as ICD codes 487-488 – which could mean much noise in ILI time series at the local level. The authors may benefit from a very similar study of influenza dissemination in the 2009 pandemic in the US, which was based on IMS CMS1500 claims data (Gog et al., 2014); the underlying paper preparing the medical claims data and validating against CDC influenza data (Viboud et al., Plos One 2013). Viboud et al., established that adding 079.9 (ICD9) to the case definition dramatically (by 90%) increased the sensitivity of the case definition, without much cost on the specificity.

The addition of codes 079.98 and 079.99 does substantially increase the number of unique people with viral infections cases (over all years) from 4,009,982 to 10,682,458. However, after carefully considering description of codes 079.99 and 079.98, we were uncomfortable using them. For example, code 079.99 is associated with the following list of conditions.

- Acute retinal necrosis

- Acute viral bronchiolitis

- Acute viral disease

- Acute viral laryngotracheitis

- Acute viral otitis externa

- Acute viral thyroiditis

- Amantadine resistant virus present

- Arthritis due to viral infection

- Arthritis of hand due to viral infection

- Arthritis of knee due to viral infection

- Arthropathy associated with viral disease

- Boid inclusion body disease

- Cardiomyopathy due to viral infection

- Cardiomyopathy, due to viral infection

- Congenital viral disease

- Congenital viral infection

- Disease of possible viral origin

- Encephalitis due to influenza-specific virus not identified

- Maternal viral disease complicating pregnancy

- Maternal viral disease in pregnancy

- Neonatal viral infection of skin

- Nonspecific syndrome suggestive of viral illness (finding)

- Oral mucosal viral disease

- Postpartum (after childbirth) viral disease

- Postpartum viral disease

- Postviral depression

- Postviral excessive daytime sleepiness

- Postviral infection debility

- Viral acute pancreatitis

- Viral bronchitis

- Viral carditis

- Viral dermatitis of eyelid

- Viral disease

- Viral disease in childbirth

- Viral disease in pregnancy

- Viral ear infection

- Viral esophagitis

- Viral eye infection

- Viral infection

- Viral infection by site

- Viral lower respiratory infection

- Viral musculoskeletal infection

- Viral myositis

- Viral pleurisy

- Viral respiratory infection

- Viral retinitis

- Viral syndrome

- Viral ulcer of esophagus

- Virus present

- Zanamivir resistant virus present

While we are not disputing Viboud et al.'s conclusions in the paper cited by the reviewer, given the large volume of specific data in our sample (4,009,982 unique people with ILIs), we do not see a compelling reason to add cases with non-specific codes. We directly evaluated “our” stricter-defined ILIs against the CDC data (temporal prevalence of influenza) and observed a very good correspondence (that deteriorates with addition of non-specific viral infection codes).

a) Yet in the "Flammable media" (Discussion section) the authors discuss how 20% of Southeners have ILI each season. This seems way too high, as most people do not seek care for ILI and 20% (and what would the rate have been if the ILI definition was not so narrow?) is already higher than population-based sero-epidemiological assessments of attack rates and it is well known that most of people with ILI do not seek medical care and that at least half of infections are asymptomatic. So 20% ILI population prevalence in Truwen data is unbelievable.

We introduced more careful wording, mentioning unreported cases, see page 22, lines 625—626.

i) Furthermore, it is difficult to believe that Southern US is better connected than, for example, dense US Northern cities like New York with large scale public transportation where influenza may be easily spread. See my later point about ICD coding preferences – another worry that seasonal initial levels in southern US is a data artefact somehow, due to for example higher Truwen coverage and different ICD coding practices.

We are talking about empirically ascertained social networks, measuring actual number of contacts among people, see Table 1.

ii) Moreover, the authors comment at some point that the ILI prevalence is higher in the mid-west. However, local differences in ICD coding preferences by doctors are likely a better explanation, and so these differences could just be artefacts. For that reason it would be better to not include prevalence in the modeling, but rather base the analysis solely on the initial epidemic phase and/or the peak timing.

In predictors we use a derivative of prevalence, which should be invariant to coding biases. Our cross-validation results suggest that, even if biases are present in coding, the modeling does capture predictive information.

4) The Truwen data source and extraction of the ILI and other time series should be described in much more detail. Is it CMS^-1^500 data? Were influenza codes sought among all codes listed or only the first-listed? Etc. Also, it would be helpful to show graphically the time series data, to allow the reader to evaluate data stability and volume – over time and between locales. Or else, add some summary table that summarizes such observational findings. The Truwen claims database is likely a convenience sample, so it is likely that the coverage increases dramatically over time since 2003, and also that there are substantial differences in coverage by US geographical areas depending on Truwen levels of foot holds. Did the authors see such intrinsic variability in the data (for example, consider another code in summer months? How would such variability in coverage influence the analyses presented here – or is it somehow taken into account?

Agreed: we used percent of insured per county to adjust for insurance coverage variation. Also, we used the total number of people described each week for given county as an offset in the Poisson regression. To answer the reviewer’s question about geographic distribution of cases, we have a movie showing ILI spread over nearly a decade-period.

a) The authors offer a comparison with HIV patterns – but is that appropriate as HIV is a chronic disease – how exactly was this done? Did authors extract only ICD codes for new onset HIV or using longitudinal patient-level data (nothing in the Materials and methods section about this)? A better comparison would perhaps be something else epidemic but with another mode of transmission such as rotavirus or norovirus diarrhea epidemics?

The HIV figure is shown only to illustrate that different diseases have distinct patterns of flow. As norovirus and rotavirus infections are much rarer, we have concerns about consistency of coding and testing (i.e., specificity) for these infections.

5) Validity of Twitter data to describe mobility? The authors conclude that short-distance land travel is an important determinant of the spread of influenza, using Twitter data to determine mobility. But is this defensible, as only older kids and adults have Twitter accounts – yet we know that children are important drivers of influenza (see fx Gog et al., two major factors determining spread of influenza was school schedules and short distance traveling). Please comment on this possibility that the Twitter data are not determinants of children's movements.

We also used a measure of direct proximity of counties to account for land movements, and also explicitly tested the school schedule hypothesis.

6) On the "Initiation spark" (Discussion section). The authors conclude that cold weather and low humidity are important drivers. Put together with their finding that most epidemics originate in the Southern United states, one must ask: How could cold weather/low humidity be a predictor of influenza and influenza starts in the South – when Southern states have warmer climate and higher humidity in early winter than Northern states? Would be great if the authors would discuss this possible paradox.

We added a new paragraph explicitly discussing this “paradox,” see subsection “Is there a paradox here?”.

Reviewer #4:This manuscript presents an analysis of a large dataset of ILI from insurance records to identify spatial patterns in influenza outbreak initiation and spread. The authors present 3 classes of analysis to identify variables correlated with outbreak initiation. Overall, I found this work very interesting but in dire need of restructuring to clarify the main questions and methods. At present, this reads as an explanation of several very innovative approaches to mining these large datasets for patterns, but doesn't lay out a clear goal or hypothesis. Similarly, the Discussion lists an array of patterns that were discovered, but fails to identify a single (or even just a few) key insight that was gained through this approach.

Agreed, restructured. The first paragraph in Discussion now explains the innovative claims of the manuscript.

I'm usually not a stickler for presentation of papers in the classic Introduction/Materials and methods/Results/Discussion format – but given that the authors are presenting several novel methods here, I think the manuscript would really benefit from being more classic. At present, some aspects of the methods are described in the "Approach #" sections, but others are presented in the "Materials and methods". I think it would be much clearer to group all the methods together so that the results could themselves be presented in a more concise format that would highlight the key insights.

Agreed, done as requested: descriptions of all methods are combined in “Materials and methods,” while “Results” summarize conclusions. Please see reply to the Editor’s comment #1.

The figures are excellent, but they are referenced out of order in the text; e.g. Figure 2 gives results from Approach 2 which is presented after the results from Approach 1, which are in Figure 4 and 5.

Corrected, figures are reordered.

Overall, I would like to see a more concise statement of the major innovation from this approach and this analysis. At present, this is a list of variables that were identified as significant (albeit using some really interesting methods and excellent data), many of which have already been identified as predictive of flu in the past. What would be really interesting, if possible, would be to illustrate whether this method was in some way more predictive of flu dynamics than conventional methods. For example, could these analyses be carried out on the first 8 years of ILI data and the resulting model used to predict the 9th; if so, does the addition of Approach 1/3 do better than the standard Poisson regression?

We added explicit highlights of innovation in our work: see the first paragraph in Discussion, which now explains the innovative claims of the manuscript.

[Editors' note: the author responses to the re-review follow.]

Essential revisions:1) The literature review is still not adequate. Reviewing the literature is not simply a matter of having a string of references. Ideally a couple of paragraphs would be added to the Introduction or Discussion discussing the history of related statistical models for understanding the timing, spread and burden of seasonal influenza, and how this paper builds on the existing literature.

As requested, we have augmented the manuscript with a short review of prior mathematical modeling approaches to analyzing influenza epidemics (see subsection “Recent computational studies of influenza”, fourth paragraph).

Also several of the previous reviewers found it difficult to determine the actual aim of the data-driven exercise. While we understand and are sympathetic to the point that this is not a traditional hypothesis-driven study, it still needs an aim if not a hypothesis. Could the last line of the first paragraph be reworked into what the aim of this was? (e.g. is it to determine the most important factors that are associated with initiating influenza transmission in the continental USA at the beginning of each season?)

Done as requested, see Introduction, first paragraph.

2) You state that "our more precisely defined ILI data defines better against the CDC dataset" – could details (including quantitative measures of better definition) be given? Both narrower and broader definitions of ILI are imperfect proxies of influenza activity, although in different ways, and it is not a given that a more specific measure is necessarily less biased.

In the revised manuscript, we provide a whole new section dedicated to this issue. We provide definitions of ICD-9-CM codes, and argue that the narrow definition we chose is most appropriate for our purposes. We argue this while directly accepting that, similar to what is outlined in Viboud et al.’s PLOS ONE article, we have no perfect gold standard to evaluate the quality of this choice; precision appears to be more important for our analysis than sensitivity.

3) The incorporation of antigenic drift, vaccine coverage and school data are important and innovative additions to the paper. However, one missing factor is host immunity not due to vaccination i.e. if there is a strong influenza season in one year with no substantial antigenic drift after that then the population immune to influenza is likely to be greater the following year. Ideally, this could be captured mechanistically by incorporating SIR-type epidemiological models, or by more statistical approaches, ideally informed by measures of prior immunity such as serology. At the very least, the implications of this factor need to be adequately discussed.

We added the following statement to the Materials and methods to address this issue:

“Accounting for non-vaccination host immunity: Resistance to influenza infection in human hosts arises via two related mechanisms: immunological memory from a previous infection by an identical or sufficiently similar strain, or vaccination against the current strain (or a sufficiently similar strain). […] Likewise, if the absolute deviation from years previous starts decreasing, we would also register a decrease in susceptibility. Incorporating these factors in the diverse set of models that we investigate guarantees that we are indeed considering host susceptibility contributions from a wide range of possible mechanisms.

4) We still think (as reviewer 3 of the previous round of reviews pointed out) that a 20% influenza infection rate is far too high, even if it incorporates asymptomatic/subclinical infection. This is much greater than the change in seropositivity over most influenza seasons other than pandemic years.

Agreed.

Our estimates of influenza infection rate are unavoidably inflated. This is because what we are measuring is the week- and county-specific prevalence of influenza-like illnesses *among insured patients who contacted their physician for any reason during the week in question*. In other words, the denominator that we use for computing prevalence is almost certain to be much smaller than the overall population of the corresponding county.

We introduced the corresponding disclaimer into the manuscript; see the “Flammable Media” paragraph in the Discussion.

5) Given the spatiotemporal nature of the findings, the authors should include more information about the Truven database. For instance, are the authors able to provide additional details about the spatial distribution of reporting rates (e.g., number of physicians contributing claims in a given county) within the Truven database? While insurance claims do provide high resolution spatial information, the use of raw counts for influenza-related diagnosis codes could be problematic if there is great spatial or temporal variation in reporting rates.

Note that we are using proportions (of people affected with ILIs out of total number of visible people in the MarketScan) rather than raw counts of affected individuals.

We provide a movie that represents influenza prevalence across space and time (each frame is one week), which is the best summary of data we can think of. We supplemented the description of the database with reference to 2017 White Paper, and additional statistics on ILI spatial and temporal distribution in the Supplement (see Figure 1—figure supplement 4).

6) While the methods and results are clearly described, the organization of the paper can be improved to facilitate easier reading. We found ourselves flipping back and forth a lot between the text because details that appeared to be missing earlier on would then appear later in different parts of the text. Figures were also cited out of order across the methods descriptions, and in some cases we don't believe all of the figures were discussed.

We made a significant effort to streamline and harmonize our exposition of methods and results.

As a few examples, the methods used to generate Figure 3 were not discussed until the Results (subsection “Approach 1: Non-parametric Granger analysis”, fourth and fifth paragraphs) but Figure 3 results are first discussed in the Materials and methods (subsection “Analysis of quantized clinical data (Approach 1)”, first paragraph).

Changed as suggested.

For more concrete advice, we would remove references to results-related figures from the Materials and methods section so that readers examine the figures only after all of the terms and analyses have been explained, and, having a cleaner separation between Materials and methods and Results will improve readability. There were many times where the authors alluded to some small part of the results in the Materials and methods before the description of the analysis had been completed.

Done as requested.

Conversely, the Results section had relatively detailed methods descriptions, which seemed redundant since the Results followed the Materials and methods. The authors may also consider a greater overhaul of the manuscript structure that groups the primary results in the text and figures (as opposed to organizing the manuscript by methodological approach) and moves many of the supporting or diagnostic figures to the supplement.

We removed the redundancy.

7) Beyond improving the organization of the paper, we think the reporting of results for Approach 1 could be better explained and supported in the text. For instance, it was not clear how the claims in the sixth and eighth paragraphs of the subsection “Approach 1: Non-parametric Granger analysis” were demonstrated with the cited figures.

We improved the description of Approach 1, providing more extended description.

8) Finally, given the emphasis on the novelty of comparing results across three methodological approaches, the authors should discuss the generalizability of their findings to influenza epidemiology beyond the United States and the generalizability of their framework to other contexts with high spatiotemporal resolution data.

We added a discussion of this generalizability (see subsection *Generalizability* in Discussion).

[Editors' note: further revisions were requested prior to acceptance, as described below.]

The remaining issues with the manuscript can be divided into two categories: (i) substantial scientific issues that need to be addressed before the manuscript is published, (ii) recommendations about the description of the work that we strongly encourage you to take but which are optional.Essential revisions about scientific content:i) (Discussion, second paragraph) It is still difficult to interpret the results related to short-range and long-range travel. We were unable to find the regression results for the full set of data weeks in the supplement. Also, there is not enough detail in the analysis description to see how Figure 5 Plates C-F demonstrate that local connections are more explanatory than airport connections.

Our conclusion that local travel is predominantly responsible for disease wave propagation is supported by several lines of analysis.

First, continuous land-movement infection waves are *visible* in the weekly influenza rate movie; we computed this movie from insurance claim data and made it available with results of this study.

Second, because our all-weeks-included dataset was too large for the R MCMCglmm package to handle, we performed mixed-effect Poisson regression calculations using a 50 percent random sample of all the weeks for which data were available. In this computation, the airport proximity fixed-effect coefficient turned out to be statistically significantly *negative* (see Source data 2, as well as the editable output file “flu-50-percent-weeks.txt”).

Third, the results from our Granger-causality inference showed that:

1) Local county-to-county movements were much more predictive of influenza wave change than airport movements. In comparing Plates E and F in Figure 5’s local movement causality coefficient (γ) is, on average, twice as large as that for long-range movement (Figure 5). Figure 5 shows that the mean long-range causality coefficient is approximately 0.05, whereas it is just over 0.1 in the local propagation. As the causality coefficient quantifies the amount of predictability (measured as information in bits) communicated about the target data stream per observed bit in the source data stream, it follows that, on average, every ten bits of sequential incidence data from an influencing location tells us one bit about the unfolding incidence dynamics in the target location. Therefore, in the long-range movement case, informativeness is twice as low, so we need on average 20 bits to infer one bit about the state of infection. These calculations strongly suggest that local movement is predictively stronger with regards to influenza infection propagation.

2) While the most frequent value of the computed time delay in influence propagation between counties with large airports is zero weeks, this distribution is significantly flatter compared to that for local, county-to-county influence propagation.

We updated the main text of the manuscript with the above prose.

ii) The highlighted areas (greater incidence) in the first flu season snapshot for Figure 3 each should match relatively well to the maps in Figure 7B (or at least some unspecified subset of "high incidence treatment counties"). While the county-matching results to point to initiation at coasts, 2006-2007 in Figure 3 seems to initiate from the West Coast and 2010 in Figure 7 has a scattered pattern across the middle of the US. Please check this or justify the differences.

There is no mistake; the resulting patterns are what they turn out to be. The intuitive explanation of perceived discrepancy is that the matching method agrees with other analysis types predominantly, but not in all cases. Each analysis has limitations. In the case of the matching analysis, we have less statistical power than in, for example, Poisson regression; matching by numerous parameters reduces the initial set of thousands of counties to a handful of matching “treated” counties (which meet a particular combination of weather and sociodemographic conditions) and “untreated” counties (very similar to “treated” ones in all respects but treatment). The difficult-to-match, “weeded out” counties may happen to be in the coastal areas indicated as the most likely places of influenza wave origin by other analyses.

In the case of the 2007 and 2010 results, the matching analyses pick patterns that are different from those produced by the causality streamline analysis and mixed-effect Poisson regression models.

Figure 6’s Plate B shows the distribution of the treatment counties and matched-non-treatment counties. Note that here, we are not directly predicting initiation, so while the patterns in Figure 3 and Figure 6 should indeed have some similarity, they are not required to match up perfectly. The most similar treated counties do indeed show up in the southern shores.

iii) Also related to Figure 7, was there also a geographic constraint on the choice of the matched control counties? It is surprising that everything is so geographically clustered.

The geographic clustering arose from imposing similar weather patterns from multiple climate variables; we added explanation of this observation to the main text. (See subsection “Approach 3: Matching counties and factor combinations” in Results.)

Recommendations about presentation of content:i) It would be useful to provide a quantitative example of how to interpret one of the results in Figure 2.

The fixed-effect regression coefficients plotted in Plate A are shown on logarithmic scale, meaning that the absolute magnitude of predictor-specific effect is obtained by exponentiating the parameter value. A negative coefficient for a predictor variable suggests that influenza rate falls as this factor increases, while a positive coefficient predicts growing rate of infection as the parameter value grows. The integrated influence of individual predictors, under this model, is additive with respect to the county-specific rate of infection.

For example, a coefficient of -0.6for parameter AVG_PRESS_mean tells us that the average atmospheric pressure has a negative association with the influenza rate. As the mean atmospheric pressure for the county grows, the probability that the county would participate in an infection initiation wave falls. As exp(-0.6)=0.54, the rate of infection *drops* by 46 percent when atmospheric pressure *increases* by one unit of zero-centered and standard-deviation-normalized atmospheric pressure. Similarly, an increase in the share of a white Hispanic population predicts an increase in influenza rate: A coefficient of 1.3 translates into a exp(1.3) ⋅ 100% – 100% = 267% rate increase, possibly, because of the higher social network connectivity associated with this segment of population.

We put this information to legend of Figure 1.

ii) (Subsection “The weather/humidity paradox”) This humidity paradox has come up quite a bit recently and it may be cleaner to have a fuller treatment of this in the Discussion.

We considered this suggestion; we do not have much to add on this topic, beyond what is already discussed. Apologies.

iii) The subsection on Variable and model selection at the end of the Discussion seems out of place. Was this meant to be moved to the Materials and methods?

Yes, it has been moved.

iv) Please point out the Discussion section describing the generalizability of the findings (as mentioned in the response).

We have added the section: our apologies, it was added, then deleted by mistake.

v) The Conclusion section does much more of the storytelling that would be expected from a Discussion. It may be more compelling to kick off the Discussion with the analogy describing the epidemiological findings, and then elaborate on the strengths of using multiple methods to corroborate statistical findings.

Agreed; it has been changed as suggested.

vi) You may consider condensing the figures in the main text to those that present primary findings (removing some plates and potentially reducing the total number of figures). The figures are all visually appealing, but every single one is densely packed with primary findings, descriptive information, data processing details, and diagnostics/sensitivity. Some of the main takeaways may get lost.

We made an effort to streamline figures, moving whole Figure 1 to the supplement.

vii) Figure 2 might be more aptly named "Putative determinants of seasonal influenza onset[…]"

Agreed. Corrected.

viii) Figure 4, Plate I: Please add an explanation of the red circles to the legend title or caption.

Done.

ix) Please provide some explanation of how you developed the 126 models examined in Approach 2. Based on the models listed in Table 5 and in the supplement, it seems that there is substantial variation in the number of predictors included in the model and all of the models appear to include multiple autoregressive terms for the same environmental and travel predictors. What is the motivation for this approach?

We added and removed random effects to the model structure, taking out and adding fixed effects, running the regression, and plotting the DIC. Model complexity was simply proxied by the descriptional complexity in terms of how large the model was (the number of factors in the regression equation). We stopped when the drop in DIC stabilized.

The driving idea here is the identification of the Pareto front that trades off accuracy (in terms of DIC) with model complexity. This is, of course, a “greedy approach,” in the sense that we do not guarantee that we have indeed found the “best” possible model. However, since our cross-validation (Figure 5 in main text) yielded good results, we deemed the stopping rule to be satisfactory.

We put this information into the main text (see subsection “Variable and Model Selection” in Materials and methods).

In addition, it would be good to better understand the random effects in the model (subsection “Approach 2: Poisson Regression on Putative Factors”, first paragraph) and the motivation for this model structure. Does this term include a spatial smoothing component?

No, we did not use an explicit spatial smoothing. The county-level random effects were used to account for possible heterogeneity in disease reporting across counties. However, we used as predictors average values of infection rates of immediate neighbors for the county (1, 2, 3 and 4 weeks in the past), this prediction structure afforded an implicit spatial smoothing, dampening spurious fluctuations in infection rates.

We clarified this point in the main text, see Approach 2 in Materials and methods.